# Exploring Landscapes for Better Minima along Valleys

**Tong Zhao**[1,2]  **Jiacheng Li**[1,2]  **Yuanchang Zhou**[1,2]  **Guangming Tan**[1,2,*]  **Weile Jia**[1,2,*]

[1]State Key Lab of Processors, Institute of Computing Technology, Chinese Academy of Sciences
[2]University of Chinese Academy of Sciences.

{zhaotong, lijiacheng22s, zhouyuanchang23s, tgm, jiaweile}@ict.ac.cn

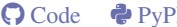  Code   PyPI

## Abstract

Finding lower and better-generalizing minima is crucial for deep learning. However, most existing optimizers stop searching the parameter space once they reach a local minimum. Given the complex geometric properties of the loss landscape, it is difficult to guarantee that such a point is the lowest or provides the best generalization. To address this, we propose an adaptor "E" for gradient-based optimizers. The adapted optimizer tends to continue exploring along landscape valleys (areas with low and nearly identical losses) in order to search for potentially better local minima even after reaching a local minimum. This approach increases the likelihood of finding a lower and flatter local minimum, which is often associated with better generalization. We also provide a proof of convergence for the adapted optimizers in both convex and non-convex scenarios for completeness. Finally, we demonstrate their effectiveness in an important but notoriously difficult training scenario, large-batch training, where Lamb is the benchmark optimizer. Our testing results show that the adapted Lamb, ALTO, increases the test accuracy (generalization) of the current state-of-the-art optimizer by an average of 2.5% across a variety of large-batch training tasks. This work potentially opens a new research direction in the design of optimization algorithms.

## 1 Introduction

Almost all gradient-based optimizers aim to converge to a local minimum [13, 31] after being trapped by a certain local minimum. Figure 1 illustrates this phenomenon of two most widely used optimizers (e.g. SGD [37] and Adam [22]) on typical optimization test functions. However, if an optimizer only relies on local information (e.g. loss function values and their gradients), it can not ensure that the point it finds is the lowest or the one with the best generalization.

What happens if we modify the optimizer to ensure continued exploration along the valley for potentially lower and flatter minima? We propose a

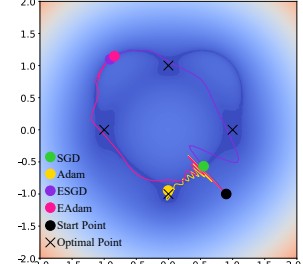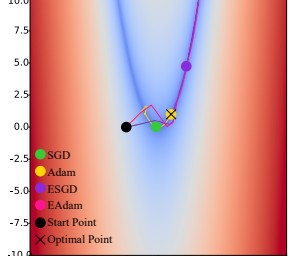

Figure 1: Comparison between SGD/Adam and ESGD/EAdam on 2D polynomial test functions, which are typically representative of landscapes. *Left (the square of the cardioid):* The valley forms a cardioid shape, where loss is 0, and special points are marked by cross. Some of these points have flat neighborhoods, which means better generalization. *Right (Rosenbrock function):* The valley is parabolic, and the optimum is marked with a cross.

---

[*]Corresponding authors

39th Conference on Neural Information Processing Systems (NeurIPS 2025).

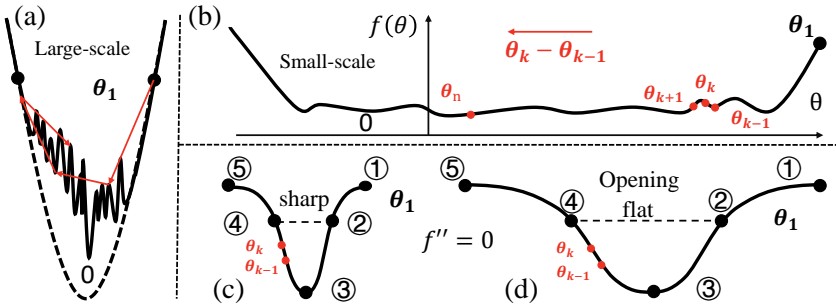

Figure 2: (a) is an intersection of valley (large-scale minimum), which captures optimizers. (b) is the enlarged solid line in (a) between two dots and shows the optimizer escaping from small-scale sharp minimum. $\mathbf{a}_k$ accelerates the training and remembers the direction of the right arrow. Zooming in on point $\boldsymbol{\theta}_k$, we obtain (c) and (d), which show how ALTO handles minimum by analyzing of directions of $\boldsymbol{\theta}_k - \boldsymbol{\theta}_{k-1}$ (or $-\bar{\mathbf{g}}_k$) and $\bar{\mathbf{g}}_k - \bar{\mathbf{g}}_{k-1}$.

gradient-based optimizer adaptor, "E" (exploration and exploitation), for this purpose. As Figure 1 shows, the adapted SGD (ESGD) and Adam (Eadam) explore along the valley (points with loss near 0). The longer the valley an optimizer explores, the flatter and lower the best minimum in the explored valley is, which is chosen as the solution to optimization.

Before implementing the aforementioned ideas, Section 2 introduces necessary preliminaries and notations and then identifies that modifying traditional gradient-based optimizers for persistent exploration requires simultaneously addressing two fundamental issues: (i) being trapped by large-scale local minima (to ensure valley-following behavior), while (ii) escaping sharp small-scale minima (to maintain exploration capability). Beyond this exploration property, the adapted optimizer demonstrates two additional advantages: accelerated training (fast loss decay) and preferential convergence to flatter local minima. For theoretical completeness, Section 3 provides convergence analysis for both convex and non-convex scenarios. The remaining sections apply the adapted optimizer ALTO to resolve an important and challenging training problem, for which the adapted optimizer is very well-suited.

Our experimental results demonstrate the superior performance of ALTO across various datasets and tasks, such as CV [20, 50, 42] and NLP [32, 33] training, with 3-5 times hyperparameter tuning per task for all optimizers in large batch training. Compared to the current state-of-the-art, ALTO achieves better accuracy in all our 17 CV and NLP experimental tasks and can save 29.68% of computation time on a typical CV task while reaching the same accuracy. In particular, ALTO can achieve better accuracy (70.83%) on ImageNet with batch size 4086 compared to SGD with a batch size of 256 (70.64%), achieve better test perplexity (78.37) than that of Lamb [51] (83.13, SOTA) in GPT-2 [34] with batch size 4096, and outperform Lamb in image classication task (ResNet50, ImageNet with the same setting as its original paper [51]) with batch sizes (1K, 2K, 4K, 8K, 16K, 32K).

## 2 Method

**Preliminaries and notations.** Consider the non-convex stochastic optimization problem

$$f^* := \min_{\boldsymbol{\theta} \in \mathcal{D}} f(\boldsymbol{\theta}), \quad f(\boldsymbol{\theta}) := \mathbb{E}_{\boldsymbol{\zeta} \sim \mathbb{P}}[\ell(\boldsymbol{\theta}, \boldsymbol{\zeta})],$$

where we define $f(\boldsymbol{\theta})$ as the landscape [43] or the entire landscape. To find the best parameter $\boldsymbol{\theta}^* = \arg\min_{\boldsymbol{\theta} \in \mathcal{D}} f(\boldsymbol{\theta})$ within a domain $\mathcal{D} \subset \mathbb{R}^d$, optimizers aim to minimize the expectation of the loss function $\ell(\boldsymbol{\theta}, \boldsymbol{\zeta})$. This function measures the suitability of the parameter $\boldsymbol{\theta}$ for a sample $\boldsymbol{\zeta}$ drawn from a dataset $\mathcal{Z} \subset \mathbb{R}^s$, subject to a probability distribution $\mathbb{P}$. To navigate the parameter $\boldsymbol{\theta}_k$ towards $\boldsymbol{\theta}^*$ at time step $k$, optimizers usually iteratively update it. For simplicity, consider SGD, which contains the core idea of all gradient-based optimizer for neural network. It guides the parameter to move along the negative gradient direction $\mathbf{g}_k := \frac{1}{|\mathcal{Z}_k|} \sum_{\boldsymbol{\zeta}_i \in \mathcal{Z}_k} \nabla \ell(\boldsymbol{\theta}_k, \boldsymbol{\zeta}_i)$ of the landscape

$\ell_k(\boldsymbol{\theta}) := \frac{1}{|\mathcal{Z}_k|} \sum_{\boldsymbol{\zeta}_i \in \mathcal{Z}_k} \ell(\boldsymbol{\theta}_k, \boldsymbol{\zeta}_i)$ of the batch $\mathcal{Z}_k$ a $\eta_k$-length step

$$\boldsymbol{\theta}_k = \boldsymbol{\theta}_{k-1} - \eta_k \mathbf{g}_k, \tag{1}$$

where $\mathcal{Z}_k$ is a batch sampled from $\mathcal{Z}$, $\mathbf{g}_k$ is an empirical estimator of $\nabla f(\boldsymbol{\theta})$, and $\nabla$ denotes computing the gradient with respect to $\theta$. In the rest of the paper, operations like $\mathbf{x}^2$ and $\mathbf{x}/\mathbf{y}$ involving any vectors $\mathbf{x}$ and $\mathbf{y}$ are elementwise, while we denote inner product, $l_2$-norm, and $l_\infty$-norm as $\langle \cdot, \cdot \rangle$, $\| \cdot \|$, and $\| \cdot \|_\infty$, respectively. If the activation function is Lipschitz, the value of $f(\boldsymbol{\theta})$ at infinity is bound by a power function. Further, if the activation function is ReLU (due to its positive homogeneity), we obtain the slice of landscape $\tilde{f}(x) := f(\boldsymbol{\theta} + x\boldsymbol{\theta}') \to Cx^c$, where $\boldsymbol{\theta} \in \mathcal{D}$ and $\boldsymbol{\theta}' \in \mathbb{S}^{d-1}$ as $x \to \infty$ for some $C, c \geq 0$. For some directions, $c = 0$ and $\tilde{f}(x)$ tends to $C$. Therefore, we assume that $\tilde{f}(x)$ is a power function at infinity.

**Design.** Designing an optimizer capable of continuing exploration along valleys rather than stagnating in local minima requires simultaneously addressing two key aspects:

1. Macroscopically, the optimizer should be captured by large-scale local minima (to ensure valley-following behavior) as shown in Figure 2 (a).

2. Microscopically, it must be able to escape from small-scale local minima (to maintain exploration capability) Figure 2 (b).

Table 1: Comparison of the directions of $-\mathbf{g}_k$, $-\nabla \|\nabla f(\boldsymbol{\theta}_k)\|^2$, and $\bar{\mathbf{g}}_k - \bar{\mathbf{g}}_{k-1}$ in different stages of Figure 2 (c) and (d). "+" and "-" represent positive and negative, respectively. The sign of $\langle \boldsymbol{\theta}_k - \boldsymbol{\theta}_{k-1}, \bar{\mathbf{g}}_k - \bar{\mathbf{g}}_{k-1} \rangle$ indicates whether the optimizer is accelerated (+) or decelerated (-).

| Direction/Sign | ①② | ②③ | ③④ | ④⑤ |
|---|---|---|---|---|
| $\boldsymbol{\theta}_k - \boldsymbol{\theta}_{k-1}$ | - | - | - | - |
| $-\bar{\mathbf{g}}_k$ | - | - | + | + |
| $-\nabla \|\nabla f(\boldsymbol{\theta}_k)\|^2$ | + | - | + | - |
| $\bar{\mathbf{g}}_k - \bar{\mathbf{g}}_{k-1}$ | + | - | - | + |
| $\langle \boldsymbol{\theta}_k - \boldsymbol{\theta}_{k-1}, \bar{\mathbf{g}}_k - \bar{\mathbf{g}}_{k-1} \rangle$ | - | + | + | - |

Almost all traditional gradient-based optimizers address the first aspect, but none of them address the second. Considering $-\nabla \|\nabla f(\boldsymbol{\theta}_k)\|^2$, we find it is similar to $-\bar{\mathbf{g}}_k := -\mathbb{E}\mathbf{g}_k$ for optimization but with repulsion from sharp minima. Moving along $-\nabla \|\nabla f(\boldsymbol{\theta}_k)\|^2$ converges to a stable point $\bar{\mathbf{g}}_k = \mathbf{0}$ and tends to escape the sharp local minimum. The stable point may not be a local minimum, but it is usually flat. The reason hides in

$$-\nabla \|\nabla f(\boldsymbol{\theta}_k)\|^2 = -2\mathbf{H}_k \bar{\mathbf{g}}_k,$$

where $\mathbf{H}_k$ is the Hessian matrix of $f$ at parameter $\boldsymbol{\theta}_k$. The only difference between $-\bar{\mathbf{g}}_k$ and $-\nabla \|\nabla f(\boldsymbol{\theta}_k)\|^2$ is the Hessian matrix $\mathbf{H}_k$. The larger $\|\mathbf{H}_k\|$ is, the sharper the minimum becomes. $-\mathbf{H}_k \bar{\mathbf{g}}_k$ enlarges the stepsize around a sharp minimum, which is helpful for escaping sharp minimum. In contrast, small $-\mathbf{H}_k$ means flat minimum, and $-\mathbf{H}_k \bar{\mathbf{g}}_k$ is helpful for convergence to the flat minimum. Therefore, combining $-\bar{\mathbf{g}}_k$ with $-\mathbf{H}_k \bar{\mathbf{g}}_k$ in an optimizer should converge to a flatter minimum than using only $-\bar{\mathbf{g}}_k$. However, directly calculating this direction is not affordable. Fortunately, we have

$$-\bar{\mathbf{g}}_k \approx \boldsymbol{\theta}_k - \boldsymbol{\theta}_{k-1},$$

which holds with the positive constant omitted, and then we have

$$-\nabla \|\nabla f(\boldsymbol{\theta}_k)\|^2 \approx \mathbf{H}_k(\boldsymbol{\theta}_k - \boldsymbol{\theta}_{k-1}) \approx \bar{\mathbf{g}}_k - \bar{\mathbf{g}}_{k-1}. \tag{2}$$

In fact, $\bar{\mathbf{g}}_k - \bar{\mathbf{g}}_{k-1}$ is a better choice for escaping sharp minima than $-\nabla \|\nabla f(\boldsymbol{\theta}_k)\|^2 \approx \mathbf{H}_k(\boldsymbol{\theta}_k - \boldsymbol{\theta}_{k-1})$. Figure 2 (c) and (d) shows the 4 different stages when an optimizer approximates a sharp minimum and flat minimum, respectively. Without loss of generality, we assume that all directions of $\boldsymbol{\theta}_k - \boldsymbol{\theta}_{k-1}$ are negative in the different stages. From Table 1, we find that the inner product between $\boldsymbol{\theta}_k - \boldsymbol{\theta}_{k-1}$ and some direction should be positive in stage ②③ and ③④ for accelerating the optimizer to escape the sharp minimum. $\bar{\mathbf{g}}_k - \bar{\mathbf{g}}_{k-1}$ is the only direction satisfying the condition. Considering gradient-based optimizers updating parameters along $-\mathbf{g}_k$, we should correct this to $-\mathbf{g}_k + \alpha'(\mathbf{g}_k - \mathbf{g}_{k-1})$ for escaping local minima and then exploring landscape where $\alpha' > 0$. The adapted optimizer is more likely to be captured by a flat minimum. If the minimum is sharp, the optimizer only needs to pass through a small opening with large $\bar{\mathbf{g}}_k$ and $\bar{\mathbf{g}}_k - \bar{\mathbf{g}}_{k-1}$ which means a large step size and thus makes it easier easy to escape. Conversely, if the minimum is flat, meaning a large opening but small $\bar{\mathbf{g}}_k$ and $\bar{\mathbf{g}}_k - \bar{\mathbf{g}}_{k-1}$, the optimizer will be captured. For stability and noticing more informative gradients at early stage of training, we replace $\mathbf{g}_k - \mathbf{g}_{k-1}$ with their exponential moving averages (EMA) [19].

---

**Main Idea (just for emphasis, connected to the context)**

Therefore, we propose a gradient-based optimizer adaptor that can adapt any gradient-based optimizer for exploring better minima along valleys via a simple replacement:

$$\mathbf{g}_k + \alpha\mathbf{a}_k \to \mathbf{g}_k, \text{ where } \mathbf{a}_k = \beta_1\mathbf{a}_{k-1} + (1 - \beta_1)\left(\mathbf{g}_k - \mathbf{g}_{k-1}\right),\ \alpha = -\alpha'. \tag{3}$$

The term $\mathbf{g}_k$ decides whether the optimizer is captured by large-scale local minima. For $\alpha\mathbf{a}_k$, if $\alpha < 0$, it helps with escaping small-scale local minima and thus exploring the landscape; if $\alpha > 0$, it helps with exploiting (accelerating convergence to) the aforementioned large-scale local minima. $|\alpha|$ represents the intensity of exploration and exploitation. $\beta_1$ reflects the persistence of the memory of the gradients-decaying-most directions. Based on (2), (3), and the preceding analysis, the advantages and limitations of the two cases can be inferred as follows (Remark 2.1):

    1. $\alpha < 0$ converges slowly (exploring large parameter space) but tends to flat minima.

    2. $\alpha > 0$ converges fast (exploring small parameter space) but tends to sharp minima.

Since our motivation is to adapt an optimizer for exploring large parameter spaces and finding flat minima, we mainly focus on $\alpha < 0$, which is very suitable for large-batch training. In fact, choosing $\alpha > 0$ usually, but not consistently, leads to some marginal advantages (accuracy, fast convergence) in the small-batch training case, but the corresponding minimum is consistently sharp.

---

Taking SGD, Adam, biased-Lamb, and Lamb [51] as examples, we adapt them into ESGD (Algorithm 3), EAdam (Algorithm 4), ALTO Vanilla (Algorithm 1), and ALTO (Algorithm 2), respectively. Similar to Lamb, we also employ layerwise regularization [51] based on the integration with Adam, and obtain Algorithm 2, where $\boldsymbol{\theta}^{(i)} \in \mathbb{R}^{d_i}$ is the parameter corresponding to the $i$-th layer, with $i \in [h] := \{1, 2, \cdots, h\}$, $\sum_{i=1}^{h} d_i = d$. $\phi$ is a zero-proof function such that $\phi(x) \sim x \geq \varepsilon_3$, usually taken as $\phi(x) := x + \varepsilon_3$. All $\varepsilon$s are very small zero-proof term, and $\lambda_k$ is a weight decay term for parameter regularization.

---

**Algorithm 1: ALTO Vanilla**

**Input:** initialize $\boldsymbol{\theta}_1$, learning rate $\eta_k$, EMA factors $(\beta_1, \beta_2, \beta_3) \in [0, 1)^3$, stable parameter $\varepsilon_1, \varepsilon_2 > 0$, weight decay $\lambda_k > 0$, acceleration factor $|\alpha| < 1/(1 - \beta_1)$, $\mathbf{a}_0 = \mathbf{0}$, $\mathbf{m}_0 = \mathbf{0}$, $\mathbf{v}_0 = 0$, and $\mathbf{g}_0 = \mathbf{0}$.

**Output:** $\{\boldsymbol{\theta}_k\}_{k=1}^{T}$.

1: **while** $k < T$ **do**
2:     $\mathbf{g}_k = \frac{1}{|\mathcal{Z}_k|}\sum_{\boldsymbol{\zeta}_i \in \mathcal{Z}_k} \nabla\ell\left(\boldsymbol{\theta}_k, \boldsymbol{\zeta}_i\right)$;
3:     $\mathbf{a}_k = \beta_1\mathbf{a}_{k-1} + (1 - \beta_1)\left(\mathbf{g}_k - \mathbf{g}_{k-1}\right)$
4:     $\mathbf{m}_k = \beta_2\mathbf{m}_{k-1} + (1 - \beta_2)\left(\mathbf{g}_k + \alpha\mathbf{a}_k\right)$;
5:     $\mathbf{v}_k = \beta_3\mathbf{v}_{k-1} + (1 - \beta_3)\left[\mathbf{g}_k + \alpha\mathbf{a}_k\right]^2$;
6:     $\boldsymbol{r}_k = \mathbf{m}_k/\left(\sqrt{\mathbf{v}_k} + \varepsilon_1\right) + \lambda\boldsymbol{\theta}_k$

7:     $\boldsymbol{\theta}_{k+1}^{(i)} = \boldsymbol{\theta}_k^{(i)} - \frac{\eta_k\mathbf{r}_k^{(i)}\phi\left(\|\boldsymbol{\theta}_k^{(i)}\|\right)}{\left(\|\mathbf{r}_k^{(i)}\| + \varepsilon_2\phi\left(\|\boldsymbol{\theta}_k^{(i)}\|\right)\right)}$

8: **end while**

**Algorithm 2: ALTO**

**Input:** the same as that of Algorithm 1

**Output:** $\{\boldsymbol{\theta}_k\}_{k=1}^{T}$.

1: **while** $k < T$ **do**
2:     $\mathbf{g}_k = \frac{1}{|\mathcal{Z}_k|}\sum_{\boldsymbol{\zeta}_i \in \mathcal{Z}_k} \nabla\ell\left(\boldsymbol{\theta}_k, \boldsymbol{\zeta}_i\right)$;
3:     $\mathbf{a}_k = \beta_1\mathbf{a}_{k-1} + (1 - \beta_1)\left(\mathbf{g}_k - \mathbf{g}_{k-1}\right)$;
4:     $\mathbf{m}_k = \beta_2\mathbf{m}_{k-1} + (1 - \beta_2)\left(\mathbf{g}_k + \alpha\mathbf{a}_k\right)$;
5:     $\mathbf{v}_k = \beta_3\mathbf{v}_{k-1} + (1 - \beta_3)\left[\mathbf{g}_k + \alpha\mathbf{a}_k\right]^2$;
6:     $\hat{\mathbf{m}}_k = \mathbf{m}_k/\left(1 - \beta_2^k\right)$;
7:     $\hat{\mathbf{v}}_k = \mathbf{v}_k/\left(1 - \beta_3^k\right)$;
8:     $\boldsymbol{r}_k = \hat{\mathbf{m}}_k/\left(\sqrt{\hat{\mathbf{v}}_k} + \varepsilon_1\right) + \lambda_k\boldsymbol{\theta}_k$
9:     $\boldsymbol{\theta}_{k+1}^{(i)} = \boldsymbol{\theta}_k^{(i)} - \frac{\eta_k\mathbf{r}_k^{(i)}\phi\left(\|\boldsymbol{\theta}_k^{(i)}\|\right)}{\left(\|\mathbf{r}_k^{(i)}\| + \varepsilon_2\phi\left(\|\boldsymbol{\theta}_k^{(i)}\|\right)\right)}$

10: **end while**

---

**Why is large-batch training important, what challenges does it pose, and how can our adaptor help?** As data scales up and GPU computing power increases, enlarging the batch size is the most direct way to fully utilize as many GPUs as possible (data parallelism) and then to accelerate pretraining. However, this acceleration occurs under the condition that the total number of training epochs remains almost unchanged. This condition means large-batch training involves far fewer parameter updates than small-batch training but expects a nearly the same test accuracy(for more details, see Section A). To overcome this challenge, Krizhevsky [23], Bottou et al. [5] proposed the linear scaling rule ($\eta_k \propto |\mathcal{Z}_k|$) and the square root scaling rule ($\eta_k \propto \sqrt{|\mathcal{Z}_k|}$). These rules are effective when batch-size is not large enough. As the batch size increases to infinity, learning rate becomes bounded [30] by a task-specific critical value determined by the geometry of the landscape. Otherwise, the training will explode. With the same learning rate, the exploratory nature of the

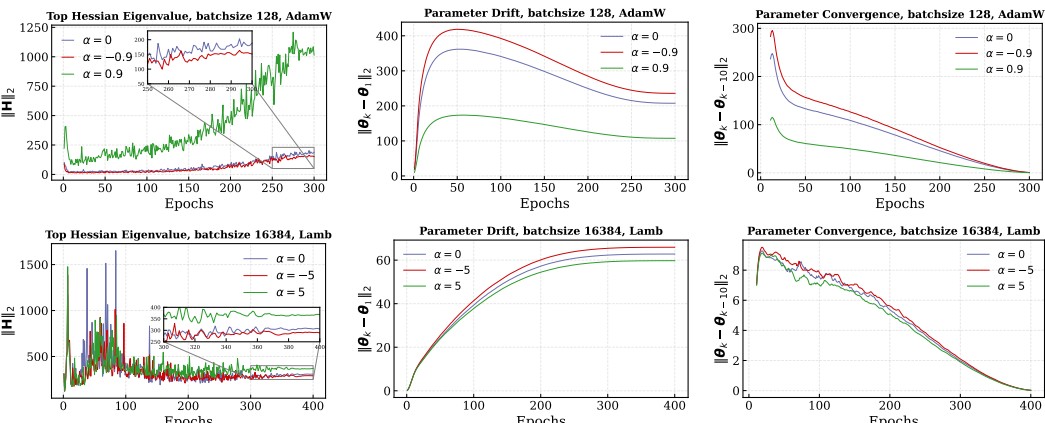

Figure 3: The variances of $\|\mathbf{H}\|_2$ (the top eigenvalue of the Hessian matrix), $\|\boldsymbol{\theta}_k - \boldsymbol{\theta}_0\|$ (parameter drift), and $\|\boldsymbol{\theta}_k - \boldsymbol{\theta}_{k-10}\|$ (parameter convergence) during the pretraining ResNet20 on CIFAR100 with AdamW ($|\mathcal{Z}_k| = 128$), Lamb ($|\mathcal{Z}_k| = 16384$) and their corresponding adapted optimizers with different values of $\alpha$.

adapted optimizer provides it a larger effective learning rate (faster training). Its underlying principle is using remembered informative gradient information at early training stage (the larger the $\beta_1$, the longer the memory) as a guidance during later training stage where gradient information is flooded with noise. Meanwhile, the adapted optimizer is more likely to be captured by flatter minima than the original optimizer. Since the fluctuations at the bottom of the landscape are relatively small, the improvement in generalization may not be substantial, but it is stable.

**Why is $\mathbf{a}_k$ incorporated into $\mathbf{g}_k$ rather than $\mathbf{m}_k$?** Incorporating $\mathbf{a}_k$ into $\mathbf{m}_k$ suggests that $\mathbf{g}_k - \mathbf{g}_{k-1}$ is on the same scale as $-\mathbf{g}_k$, which causes the optimizer to either vibrate violently or have no effect. Incorporating it into $\mathbf{g}_k$ means the EMA of the EMA of $\mathbf{g}_k - \mathbf{g}_{k-1}$ in the final momentum, whereas incorporating into $\mathbf{g}_k$ means a EMA of $\mathbf{g}_k - \mathbf{g}_{k-1}$ in final momentum. These two are intrinsically different, we can not identify the EMA with the EMA of EMA just by adjusting $\beta$. In fact, if we define $\text{EMA}_k$ as the EMA of $\text{EMA}_{k-1}$ with common coefficient $\beta$ for any $k$, the weight of $\mathbf{g}_i - \mathbf{g}_{i-1}$ in $\text{EMA}_n$ at time step $k$ is

$$(1-\beta)^n \beta^{k-i} \binom{k-i+n-1}{n-1}.$$

The larger the n is, the more stable the $\text{EMA}_n$ becomes. We use $n = 2$ for affordable computational cost. This means Algorithm 2 moves along the $\text{EMA}_2$ of $\mathbf{g}_k - \mathbf{g}_{k-1}$ near the bottom of valley. The $\text{EMA}_2$ of $\mathbf{g}_k - \mathbf{g}_{k-1}$ heavily depends on the early stage of training where gradients decay very fast.

*Remark* 2.1. In order to verify the advantages and limitations of positive and negative $\alpha$, we adapt AdamW and Lamb ($\alpha = 0$ Lamb, $\alpha = -5$ ALTO) as examples for the small and large batch cases respectively. Figure 3 illustrates that, compared to adapted optimizers with $\alpha > 0$, those with $\alpha < 0$ are more likely to find flatter minima in larger areas but converge slowly.

*Remark* 2.2. On the constraint $|\alpha| < 1/(1-\beta_1)$. If we replace $k$ with $t$, set $\mathbb{D} = \frac{d}{dt}, \eta_t = \eta \approx \Delta t$, and treat $\boldsymbol{\theta}$ as a 1-dimensional dynamical system updating via $\mathbf{m}$ without considering $\mathbf{v}$ and layer regularization, we have

$$\dot{\theta} = \frac{\theta_t - \theta_{t-1}}{\Delta t} = -m, \quad a = \frac{1}{\alpha}\left(\frac{\eta}{1-\beta_2}\dot{m} + m - g\right), \quad \dot{g} = \frac{\dot{a}}{1-\beta_1} + \frac{a}{\eta}. \qquad (4)$$

According to $g = f', \dot{g} = f''\dot{\theta}$ and Equation (4), we get the differential equation that $\theta$ satisfies:

$$\left[(\eta\mathbb{D}+1-\beta_1)(\eta\mathbb{D}+1-\beta_2) + (1-\beta_2)\eta f''(1+(1-\beta_1)\alpha)\right]\dot{\theta}$$
$$+ (1-\beta_1)(1-\beta_2)f' = 0. \qquad (5)$$

If we regard $f'$ and $f''$ as constant in Equation (5), it becomes a linear ordinary differential equation. For stability and convergence, the real parts of the system's eigenvalues must be negative. Therefore,

$$-\left(\frac{1}{1-\beta_1} + \frac{1}{\eta f_1}\right) < \alpha < -\left(\frac{1}{1-\beta_1} + \frac{1}{\eta f_2}\right),$$

where $f_1 = \max\{\max_\theta f'', 0_+\}, f_2 = \min\{\min_\theta f'', 0_-\}$. For simplicity, we use $|\alpha| < 1/(1-\beta_1)$.

# 3 Algorithm Convergence Analysis

Prior to conducting convergence analysis for both non-convex [7] and convex [36] cases, we require some common assumptions that are widely used in related works [52, 9, 3].

**Assumption 3.1** ($L$-smoothness)**.** The function $\ell(\boldsymbol{\theta}, \boldsymbol{\zeta})$ is $L$-smooth if and only if there exists a constant $L$ such that

$$\|\nabla\ell(\boldsymbol{\theta}, \boldsymbol{\zeta}) - \nabla\ell(\boldsymbol{\theta} + \boldsymbol{\delta}, \boldsymbol{\zeta})\| \leq L\|\boldsymbol{\delta}\|$$

for all $\boldsymbol{\theta} \in \mathbb{R}^d, \boldsymbol{\delta} \in \mathbb{R}^d$ and $\boldsymbol{\zeta} \in \mathcal{Z}$.

**Assumption 3.2** (Unbiased, independent, and variance-bounded stochastic gradient)**.** We assume:

1. $\forall k \in \mathbb{N}, \quad \bar{\mathbf{g}}_k := \mathbb{E}\mathbf{g}_k = \nabla\mathbb{E}[\ell(\boldsymbol{\theta}_k, \boldsymbol{\zeta})]$
2. $\forall k \neq t \in \mathbb{N}, \mathbf{g}_k$ and $\mathbf{g}_t$ are independent
3. $\mathbb{E}\|\nabla\ell(\boldsymbol{\theta}, \boldsymbol{\zeta}) - \nabla\mathbb{E}\ell(\boldsymbol{\theta}, \boldsymbol{\zeta})\|^2 \leq \sigma^2, \forall \boldsymbol{\theta} \in \mathbb{R}^d$.

**Assumption 3.3** (Bounded gradient)**.** We suppose that at any time step $k$, gradients $\mathbf{g}_k$ are bounded i.e., $\|\mathbf{g}_k\|_\infty \leq G_\infty/16$.

**Assumption 3.4** (Bounded parameter)**.** We suppose that at any time step $k$, parameter $\boldsymbol{\theta}_k$ is bounded i.e., $\|\boldsymbol{\theta}_k\| \leq D$ and $\|\boldsymbol{\theta}_k\|_\infty \leq D_\infty$.

**Assumption 3.5** (Monotonicity)**.** We assume $\forall i \in [d]$,

$$\frac{\sqrt{k}\left(\sqrt{\mathbf{v}_{k,i}} + \varepsilon_1\right)\left(\left\|\mathbf{r}_k^{(i)}\right\| + \varepsilon_2\phi\left(\left\|\boldsymbol{\theta}_k^{(i)}\right\|\right)\right)}{\phi\left(\left\|\boldsymbol{\theta}_k^{(i)}\right\|\right)}$$

increases monotonically with respect to $k$.

**Assumption 3.6** (Convexity)**.** Assuming $\ell(\cdot, \boldsymbol{\zeta})$ is convex meaning $\forall \boldsymbol{\zeta} \in \mathbb{R}^s, \boldsymbol{\theta}, \boldsymbol{\theta}' \in \mathbb{R}^d, \gamma \in [0, 1]$, we have

$$\gamma\ell(\boldsymbol{\theta}, \boldsymbol{\zeta}) + (1 - \gamma)\ell(\boldsymbol{\theta}', \boldsymbol{\zeta}) \geq \ell(\gamma\boldsymbol{\theta} + (1 - \gamma)\boldsymbol{\theta}', \boldsymbol{\zeta}).$$

*Remark* 3.1. Assumptions 3.1 to 3.3 are common for the analysis of stochastic first-order methods in non-convex [11, 12, 35]. Assumptions 3.3 to 3.5 (or its analogue) and Assumption 3.6 usually appear in the analysis of convex optimization convergence [36, 52]. Except for a bounded factor, Assumption 3.5 and its analogue are common for convex case convergence analysis [52, 8]. If without it, the proof can not be finished. In fact, the optimizers in above literature and ours, also violating this kind of condition, can eventually converge during training.

**Theorem 1.** *If Assumptions 3.1 to 3.3, $\mu = \frac{\sqrt{1-\beta_3}G_\infty}{\varepsilon_1} \leq 1, \mathcal{Z}_k = b = \mathcal{O}\left(G_\infty\epsilon^{-2}\right), \lambda_k = \lambda(1 - \mu)^k, \eta_k^2 = \eta^2 \leq \frac{\varepsilon_1^3\hat{\varepsilon}_2^4\varepsilon_3^2(1-\beta_2)^2}{6dL^2G_\infty}$ and $T \geq \mathcal{O}\left(G_\infty^{1.5}\epsilon^{-2}\right)$ hold, Algorithm 1 satisfies:*

$$\frac{1}{T+1}\sum_{k=0}^{T}\mathbb{E}\left(\|\nabla f_k(\boldsymbol{\theta}_k)\|^2\right) \leq 4\epsilon^2,$$

*where*

$$f_k(\boldsymbol{\theta}) := \mathbb{E}_{\boldsymbol{\zeta}}[\ell(\boldsymbol{\theta}, \boldsymbol{\zeta})] + \frac{\lambda_k}{2}\|\boldsymbol{\theta}\|_{\sqrt{\mathbf{v}_k}}^2, \quad \|\boldsymbol{\theta}\|_{\sqrt{\mathbf{v}_k}}^2 := \langle\boldsymbol{\theta}, (\sqrt{\mathbf{v}_k} + \varepsilon_1)\boldsymbol{\theta}\rangle.$$

*If the above conditions, Assumption 3.4, and $T \geq \mathcal{O}\left(G_\infty^2 D\epsilon^{-2}\right)$ holds, Algorithm 1 satisfies:*

$$\frac{1}{T+1}\sum_{k=0}^{T}\mathbb{E}\left(\|\nabla f(\boldsymbol{\theta}_k)\|^2\right) \leq 6\epsilon^2.$$

*Remark* 3.2. The proof and detailed version considering more hyperparameters is given at Section C.2. There, we change $\beta_i$ to $1 - \beta_i$ for all $i \in [3]$ for brevity. Compared with Lamb [51], we use a different analysis method and lead to a similar result under weaker constraints in more hyperparameter cases which are omitted in Lamb for simplicity (Table 2).

Table 2: Convergence analysis comparison (Lamb vs. ALTO)

| | $T \geq$ | $b \geq$ | $\eta \leq$ | some $\beta$s = |
|---|---|---|---|---|
| Lamb | $\mathcal{O}\left(\epsilon^{-4}\right)$ | $\mathcal{O}\left(\epsilon^{-4}\right)$ | $\mathcal{O}\left(\epsilon^{-2}\right)$ | 0 or 1 |
| ALTO | $\mathcal{O}\left(\epsilon^{-2}\right)$ | $\mathcal{O}\left(\epsilon^{-2}\right)$ | $\mathcal{O}(1)$ | more general |

**Theorem 2.** *Suppose that Assumption 3.3, 3.4, 3.5, and 3.6 hold. Let $\eta_k = \frac{\eta}{\sqrt{k}}, \alpha_k \leq \frac{\alpha}{\sqrt{k}}, \lambda_k \leq \frac{\lambda}{\sqrt{k}}$, and $\frac{\beta_2^2}{\beta_3} < 1$. Algorithm 1 achieves the following guarantee*

$$R(T) \leq \mathcal{O}\left(T^{0.5}G_\infty d^{1.5}D_\infty^2\right) \sim \mathcal{O}\left(\sqrt{T}\right),$$

*where $R(T) := \sum_{k=1}^{T}\left(\ell_k(\boldsymbol{\theta}_k) - \ell_k(\boldsymbol{\theta}^*)\right)$.*

*Remark* 3.3. For the proof and more accurate estimation, see Section C.3. We obtain the same main term $\mathcal{O}\left(\sqrt{T}\right)$ as Adam when $T \to \infty$ up to an additional factor $\sqrt{d}$ without requirement $\beta_{1,k} = \beta_k \lambda^k$ and a different requirement $\frac{\beta_2^2}{\beta_3} < 1$, which is due to the factor $\frac{\left(\left\|\mathbf{r}_k^{(i)}\right\| + \varepsilon_2 \phi\left(\left\|\boldsymbol{\theta}_k^{(i)}\right\|\right)\right)}{\phi\left(\left\|\boldsymbol{\theta}_k^{(i)}\right\|\right)}$ in ALTO.

*Remark* 3.4. Although these theorems are only proved for Algorithm 1, analogous consequences also hold for Algorithm 2 up to a bounded factor of $\beta_i, i \in [3]$, since the bias correction is bounded between 1 and $1/(1 - \beta_i)$, similar statement in [29, 8].

## 4    Experiments

We mainly focus on large batch training problem and evaluate the Lamb-adapted optimizer ALTO. In the small batch case, there is also improvement but it is marginal Table 8. We compare ALTO with typical optimizers, including SGD, Adam, AdamW [28], Lamb, and AdaBelief [52] across various tasks (CV, NLP, reinforcement learning (RL)), diverse datasets (CIFAR-10, CIFAR-100 [24], ImageNet [15], CoNLL-2003 [38], IMDB [2], MRPC [46], and GPT-2 Output Dataset [1]), various architectures and scales (ResNet18, ResNet20, ResNet34, ResNet50 [15], VGG16 [41], DenseNet169 [18], LSTM [16], BERT [10], and GPT-2 [34]), distinct environments (Swimmer, Ant and Humanoid [44]), and different batch size (from 32 to 50K). The following analysis mainly focus on CV and NLP tasks in this section, and ALTO is particularly suitable for GPT-based generative models due to its large effective batch size.Additionally, we also conduct experiments on RL and LSTM tasks(Section D.1). Finally, ablation study and hyperparameters tuning experiments show the influence of ALTO's important terms and hyperparameters on optimization results, in order to demonstrate the necessity of their introduction and the relevant statements or explanations in former section. For fair comparison, we conduct experiments on the same conditions, which would be different from those in their own original papers. Therefore, this may cause a little difference in experiment results than what are reported in original papers (See Section D for more experimental details). Of course, we also conduct some typical experiments under the same conditions as their original papers. In all these experiments, the best results of different optimizers are in bold. Each result is the mean of three independently repeated experiments.

Table 3: Test top-1 Acc. (%) on ResNet20 with CIFAR10 and CIFAR100 and on ResNet34 with ImageNet (for hyperparameters and architecture details, see Section D.2)

| Dataset | CIFAR10 | | CIFAR100 | | ImageNet | |
|---|---|---|---|---|---|---|
| Batch Size | 128 | 16384 | 128 | 16384 | 256 | 4086 |
| SGD | **91.85** | 80.86 | 64.93 | 44.20 | **70.64** | 49.35 |
| Adam | 89.88 | 87.34 | 64.35 | 54.91 | 65.06 | 54.96 |
| AdamW | 90.54 | 82.29 | 64.62 | 52.95 | 69.64 | 68.40 |
| Lamb | 90.89 | 83.56 | 61.29 | 56.06 | 69.17 | 70.34 |
| AdaBelief | 91.12 | 88.03 | 64.44 | 52.94 | 70.12 | 70.18 |
| ALTO | 91.24 | **88.83** | **65.74** | **57.78** | 69.95 | **70.83** |

Table 4: Test top-1 Acc. (%) of different optimizers for ImageNet training with RESNET-50 using different batch sizes in 90 epochs. † is reported in [51]. (for hyperparameters and architecture details, see Section D.2)

| Batch Size | 1K | 2K | 4K | 8K | 16K | 32K |
|---|---|---|---|---|---|---|
| Adam | 73.08 | 73.08 | 73.32 | 73.11 | 73.09 | 72.50 |
| AdamW | 75.65 | 74.93 | 74.65 | 74.40 | 74.10 | 73.57 |
| Adabelief | 73.32 | 73.48 | 73.41 | 73.14 | 73.00 | 72.89 |
| Lamb | 77.06† | 77.11† | 76.92† | 76.89† | 76.66† | 76.42† |
| ALTO | **77.22** | **77.25** | **77.35** | **77.10** | **76.87** | **76.70** |

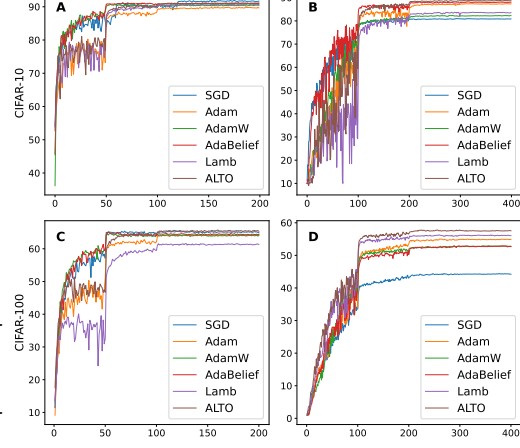

Figure 4: Test top-1 Acc. (%) on CIFAR-10 and CIFAR-100 with batch size 128 and 16384 on ResNet-20. The x-axis is epoch (for hyperparameters and archetecture details, see Section D.2).

**Machine configuration.** All our experiments were conducted on single node equipped with 4 NVIDIA 80GB A100 GPUs interconnected with PCI-E3.0. We remark that multi-node experiments were not performed due to our limited hardware resource.

Table 5: (a) Test top-1 accuracy (%) of ALTO and Lamb with batch size scaling. (b) Training time (s) with batch size 16384 to achieve the same test top-1 accuracy. Both (a) and (b) involve training VGG-16 on CIFAR100. (c) Comparative performance (train loss and test perplexity) of optimizers in training the GPT-2 (345M parameters) Model with batch size 4096 on Megatron-LM Framework by the OpenAI's open-sourced GPT-2 Output Dataset (1M).

| (a) | Batch Size | | | | | | | |
|---|---|---|---|---|---|---|---|---|
| | 200 | 500 | 1K | 2K | 5K | 10K | 25K | 50K |
| ALTO | 66.9 | **63.35** | 59.79 | **70.43** | **67.55** | **64.51** | **59.19** | **49.69** |
| Lamb | **67.3** | 63.3 | **59.82** | 70.24 | 67.05 | 62.92 | 57.79 | 39.91 |

| (b) | 20% | 30% | 40% | 50% | 60% |
|---|---|---|---|---|---|
| ALTO | **137.103** | **202.674** | **333.817** | **482.842** | **608.023** |
| Lamb | 195.992 | 276.694 | 409.277 | 582.210 | 864.669 |

| (c) | Loss | PPL |
|---|---|---|
| Adam | 4.43 | 87.74 |
| AdamW | 4.43 | 86.51 |
| Adabelief | 5.17 | 182.80 |
| Lion | 4.66 | 110.54 |
| Lamb | 4.39 | 83.13 |
| ALTO | **4.33** | **78.37** |

**Hyperparameters.** Though ALTO introduces five extra hyperparameters compared with Adam, we usually and only adjust parameter $\beta_1$ and $\eta$ according to batch size. It is clear that the larger the batch size is, the larger the $-\alpha$ and $\beta_1$ should be. Hence, we set $\alpha = 0.5, \beta_1 = 0.01$ in small batch training (batch size <1K) and $\alpha = -5, \beta_1 = 0.99$ in large batch case (batch size ≥1K), unless otherwise specified. If not mentioned, we set $\beta_2 = 0.9, \beta_3 = 0.99, \lambda = 10^{-4}, \varepsilon_1 = 10^{-6}, \varepsilon_2 = 10^{-6}, \varepsilon_3 = 10^{-10}$. These parameters allow ALTO ample room for performance improvement. We only adjust $\beta_1$ and $\eta$ for ALTO, while for other optimizers, we tune all hyperparameters.

## 4.1 Image Classification

We conduct experiments on a variety of convolutional neural networks [26, 25] (CNNs) and the datasets mentioned above. Due to space limitations, we only give a part of representative experiment results here (see Section D.1 for more experiments and details). As ALTO is tailored for large-batch training (in case 16384), it not only outperforms all listed competitors on these three datasets, but also achieves a better result(70.83) than SGD (70.64) on a relatively small batch (256) on ImageNet. There is a widely accepted view that if the same learning rate is used, small batch SGD typically converges more slowly than the Adam family (Adam, AdamW, AdaBelief), but often achieves superior convergence result [52, 28] (Figure 4). This may be because the loss of large batch better approximates the landscape. In contrast, ALTO beats them only on CIFAR-100 for small-batch (128) case (Table 3). The larger the batch size, the larger the advantage of ALTO over other optimizers (Table 4, Table 5 (a), Figure 12 and Figure 13 in Section D.1). We visualize the training process in Figure 4. As ALTO uses history information $\mathbf{a}_k$ to guide the training in later stage (after epoch 50 and 100 for small batch size 128 or epoch 100 and 200 for large batch size 16384) where gradient information is noisy, it outperforms other optimizers to a greater extent when the batch size is larger, implying more accurate gradients-decaying direction $(\bar{\mathbf{g}}_k - \bar{\mathbf{g}}_{k-1})$ information $\mathbf{a}_k$ containing. In large batch cases (16384), ALTO achieves the best accuracy of other optimizers in our experiments using only half the number of epochs. Honestly, due to the extra acceleration $\mathbf{a}$, ALTO's epoch computation time is longer than that of Lamb. Considering these two factors, we find that ALTO leads to less training time to reach a given accuracy(Table 5(b)). For more details, see Section D.1. Finally, ALTO as an optimizer developed based on Lamb, it outperforms Lamb in different batch size cases (Table 4) at nearly any training stage (Figure 4). For additional experimental details, refer to Section D.2.

## 4.2 NLP

Transformer-based attention neural networks [45] are heavily used in natural language processing. To demonstrate ALTO's ability to train popular large language model (LLM), we have employed two of them. One model we mainly focus on is GPT-2 with 345M parameters on Megatron-LM Framework [40] for pre-training tasks in GPT-2 Output Dataset (Table 5(c)), and the other, introduced in Table 9 (Section D.1) for experiment completeness, is BERT$_{base}$ with 110M parameters for fine-tuning tasks in three datasets, where we also observe that as the batch size increases, ALTO's advantages become more pronounced and stable. We choose these two relatively small LLMs, due to our limited available computational resources. In the pre-training task, ALTO achieves an obvious generalization advantage (78.3 test perplexity) over other optimizers (Table 5). This is due to the massive equivalent batch size, batch size (4096)×sentence length (20-30), caused by the GPT

task, where every token needs to be predicted. Therefore, ALTO is suitable for pre-training current generative LLMs. Compared with a recent proposed optimizer Lion [6], ALTO achieves its final perplexity using only one-third of its iteration (also epoch) count, though ALTO requires 332ms and lion requires 253ms per iteration. If ALTO is applied to LLM training, another concern is the additional GPU memory overhead. Due to the lack of computational resource, directly measuring extra GPU memory overhead on real LLMs like GPT-4 is unrealistic, but we can estimate it via the undetermined coefficients method, about 2% more than Lamb, as shown below. Compared with Lamb, the memory consumption of ALTO increases $\frac{\text{ALTO memory}}{\text{Lamb memory}} - 1 = \frac{d + c_1 db + c_2 d}{c_1 db + c_2 d} - 1 = \frac{1}{c_1 b + c_2}$, where $d$ is the number of parameters, $b$ is the batch size, and $c_1, c_2$ are some coefficients to be determined. The reason is the extra term $\mathbf{a}$ leads to memory consumption $d$, the forward and backward process for every sample and parameter lead to a consumption $c_1 db$, and intermediates like $m, v$, etc. lead to a consumption $c_2 d$. According to the above formula $\frac{1}{c_1 b + c_2}$ and Table 6 (b), we find this growth rate is irrelevant to $d$. Taking BERT$_{\text{base}}$, a model based on transformer, as example, we have Table 6 (a) with $d$ fixed and Table 6 (b) with $b$ fixed. According to the data of batch size 32 and 1024 in Table 6 (a), we have $c_1 = 0.048$ and $c_2 = 0.7877$, and find it fits well if batch size is 512. For experiment details, see Section D.2.

Table 6: The comparison of GPU memory consumption (MiB) of ALTO and Lamb (a) using BERT$_{\text{base}}$ with different batch sizes. (b) using batch size 1024 on BERT$_{\text{base}}$ with varying degrees of parameter reduction.

| (a) **Batch Size** | 32 | 512 | 1024 |
|---|---|---|---|
| Lamb | 5491 | 58636 | 109006 |
| ALTO | 7869 (+43.31%) | 60822(+3.73%) | 111228 (+2.04%) |

| (b) #Parameters | 44526345 | 65789961 | 87053577 | 108317193 |
|---|---|---|---|---|
| Lamb | 34776 | 59269 | 84240 | 108998 |
| ALTO | 35568 (+2.27%) | 60805 (+2.59%) | 85946 (+2.02%) | 111220 (+2.03%) |

Table 7: Results of ablation study (%) on CIFAR10 (batch size 16384) and MRPC (batch size 2048) on the test dataset.

| | CIFAR10 | | MRPC | | | | | |
|---|---|---|---|---|---|---|---|---|
| Feature | Top-1 Acc. | Δ | Acc. | F1 | Prec | Recall | Avg | Δ |
| ALTO | **88.83** | - | **74.95** | **83.05** | **75.48** | 92.32 | **81.45** | - |
| - $a_m$ | 87.06 | -1.77 | 70.08 | 80.87 | 70.34 | 95.11 | 79.10 | -2.35 |
| - $a_v$ | 87.08 | -1.75 | 67.13 | 79.78 | 67.49 | 97.55 | 77.98 | -3.47 |
| - $bc$ | 87.51 | -1.32 | 72.92 | 81.56 | 74.53 | 90.06 | 79.76 | -1.69 |
| - $lrr$ | 86.22 | -2.61 | 72.86 | 81.69 | 74.09 | 91.02 | 79.91 | -1.54 |
| - $(a_m + a_v)$ | 88.11 | -0.72 | 69.44 | 80.64 | 69.67 | 95.72 | 78.86 | -2.59 |
| - $(a_m + a_v + bc)$ | 84.91 | -3.92 | 71.53 | 81.00 | 72.80 | 91.28 | 79.15 | -2.3 |
| - $(a_m + a_v + bc + lrr)$ | 77.55 | -11.28 | 66.49 | 79.87 | 66.49 | **100.0** | 78.21 | -3.24 |

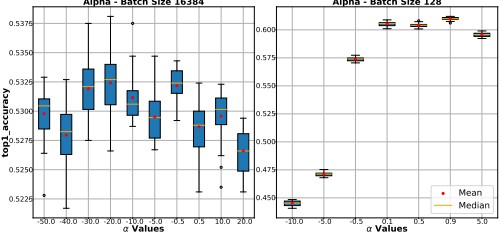

Figure 5: The distribution of top-1 accuracy with different $(\beta_1, \alpha)$ after ResNet-18 trained 180 epochs on CIFAR-100.

## 4.3 Ablation Study.

To demonstrate the necessity of each component introduced in our algorithm, we remove the acceleration terms $\mathbf{a}$ in $\mathbf{m}$ ($a_m$) and $\mathbf{v}$ ($a_v$), the bias correction term $1/(1 - \beta_i^k)$, $i \in 2, 3$ ($bc$), and the layerwise learning rate regularization factor as introduced in Lamb ($lrr$). We then conduct an ablation study (Table 7). The results show that all these components are indispensable, and that ALTO and Lamb complement each other well.

## 4.4 The Choice of $\beta_1$ and $\alpha$

According to Equation (3), $\beta_1$ measures the persistence of exploration, while $|\alpha|$ determines the scale of local minima that can be escaped during the exploration as discussed in design part. Experimental results reveal the effects of $\beta_1$ and $\alpha$ on ALTO's performance (Figure 5). Generally, a larger batch size requires a larger $\beta_1$ and a smaller $\alpha$. For large batches, we set $\alpha$ to be negative for larger exploration range, flatter minima (Figure 3), and this leads to a better test performance. However, for small batches we set it positive. Although a negative $\alpha$ suggests flatter local minima, the resulting improvement in generalization is insufficient to offset the usually lower training loss achieved with positive $\alpha$. If a flat minimum is desired (Figure 3), setting $\alpha$ to a negative value is also acceptable.

# 5   Related Works

Lion [6] using double momentum method computes the $\text{EMA}_2$ of the gradient $(g_k)$, while ALTO first computes an EMA of acceleration $(g_k - g_{k-1})$, adds the EMA to $g_k$, and then computes EMA of the addition. Moreover, ALTO uses a negative $\alpha$, whereas Lion uses a positive effective $\alpha$. Additionally, ALTO considers the second moment, while Lion does not.

Adan [48] also uses EMA of $g_k - g_{k-1}$. However, there are two major differences. First, Adan introduces the EMA of $g_k - g_{k-1}$ directly to the momentum in update equation, but we introduce it as an acceleration to gradient $g_k$. This means we use the EMA of the EMA of $g_k - g_{k-1}$ to estimate the momentum in the update equation which is more natural in form. Second, in large-batch training, to accelerate the optimizer for exploration, its hyperparameter $\alpha$ is usually set to a negative value, such as -5 in our experiments. In contrast, the equivalent $\alpha$ of Adan is positive.

Indeed, ALTO requires more memory and computation than these two optimizers per iteration (Table 12 in Section D.1), but ALTO can achieve higher accuracy (Tables 10 and 11 in Section D.1) and require only one third of the number of iterations compared to LION to reach a reasonable perplexity (Figure 9 in Section D.1). This implies that the training time required to reach an acceptable pre-training perplexity (PPL=200) using ALTO is 56.2% less than using LION (Section D.1). However, when compared with matrix-based optimizer like Muon [21] and Shampoo [14] with computational complexity $\mathcal{O}(N^{\frac{3}{2}})$ ($N$, the number of parameter), the extra computation related to $\mathbf{a}$ is much less, with a complexity of $\mathcal{O}(N)$.

A class of recently popular optimizers, which reduce the variance of the loss function $\sigma_k := \sqrt{\mathbb{E}\left|\ell_k(\boldsymbol{\theta}, \zeta) - \mathbb{E}\ell(\boldsymbol{\theta}, \zeta)\right|^2}$ [4], are related to our adaptor design idea that reduces the norm of gradient $\|\nabla \ell_k(\boldsymbol{\theta})\|^2$. The relationship between $\sigma_k(\boldsymbol{\theta})$ and $\|\nabla \ell_k(\boldsymbol{\theta})\|$ is given by $\|\nabla \ell_k(\boldsymbol{\theta})\|\sqrt{\mathbb{E}\delta_k^2(\boldsymbol{\theta})} = \sigma_k(\boldsymbol{\theta})$, if we define $\delta_k(\boldsymbol{\theta}) := \frac{\mathbb{E}\ell(\boldsymbol{\theta}) - \ell_k(\boldsymbol{\theta})}{\|\nabla \ell_k(\boldsymbol{\theta})\|}$ called horizontal amplitude at $\boldsymbol{\theta}$ and assume $\|\nabla \ell_k(\boldsymbol{\theta})\| \neq 0$ at $\boldsymbol{\theta}$. The optimization objectives of the two methods are the same up to a standard deviation of $\delta_k$. From the definition of the horizontal amplitude $\delta_k$, we find $\mathbb{E}\ell(\boldsymbol{\theta}) = \ell_k(\boldsymbol{\theta}) + \delta_k(\boldsymbol{\theta})\|\nabla \ell_k(\boldsymbol{\theta})\| \approx \ell_k(\boldsymbol{\theta} + \delta_k(\boldsymbol{\theta})\frac{\nabla \ell_k(\boldsymbol{\theta})}{\|\nabla \ell_k(\boldsymbol{\theta})\|})$. This means if we regard the graph of $\mathbb{E}\ell(\boldsymbol{\theta})$ as a translation of the graph of $\ell_k(\boldsymbol{\theta})$, the smallest translation length should be $\delta_k$. The $\delta_k$ measures how much the graph of $\ell_k(\boldsymbol{\theta})$ oscillates around the graph of $\mathbb{E}\ell(\boldsymbol{\theta})$.

Apart from first-order optimizers, our method "E" can also adapt zero-order optimizers [27], second-order optimizers [49, 17] and matrix-based optimizers (Muon, Shampoo), by replacing gradient estimator $\mathbf{g}_k$ with $\mathbf{g}_k + \alpha \mathbf{a}_k$ (Algorithm 5).

# 6   Limitations and Future Work

**The introduction of two hyperparameters** ($\beta_1$ **and** $\alpha$). The optimal values of the two hyperparameters vary for different tasks, origin optimizers, and platforms. We use one default setting, which may not be the best but is consistently well-behaved under various conditions. In the following process of maintaining the adaptor "E" and optimizer ALTO package, we will further improve the parameter settings.

**The improvement is limited in small batch case**. Small training batch size usually leads to approximately optimal test performance, which limits the potential for improvement. Meanwhile, the gradient estimator in small batch case is noisy, so we should not use a large $\beta_1$, which limits the effect of adaptor.

# 7   Conclusion and Results

We propose a gradient-based optimizer adaptor, which can make the optimizer continue exploring the parameter space along valleys rather than circling around the minimum that traps the optimizer. We use it as a suitable tool for large-batch training tasks. The large-scale exploration capability of the adapted optimizer can mitigate the constraints imposed by unadjustable learning rates. We conduct experiments on typical large-batch training tasks, and the adapted Lamb (ALTO) outperforms current top optimizers, especially in NLP tasks, since their effective batch sizes are extremely large.

# 8 Acknowledgements

This work is supported by the following funding: National Science Foundation of China (92270206, 62032023, 62372435, T2125013) and Huawei Technologies Co., Ltd. The model training was performed on the robotic AI-Scientist platform of Chinese Academy of Science. We extend our sincere gratitude to Xirui Yang for insightful discussions during the conceptualization phase of this research. Special thanks to Dr. Jianchao Tan at Meituan Co., Ltd. for deploying ALTO on their internal large language models and utilizing it for pre-training, whose expertise in enterprise deployment was crucial for practical and industrial verification. Finally, special thanks to Jiacheng Li for his discussions and core contributions in experimental implementation of computer vision, natural language processing, and reinforcement learning tasks.

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

# Appendix

# A The Challenge in Large-Batch Training

For clearly explaining the challenge of large batch training, we conducted preliminary experiments on a simple regression task using a basic neural network architecture.

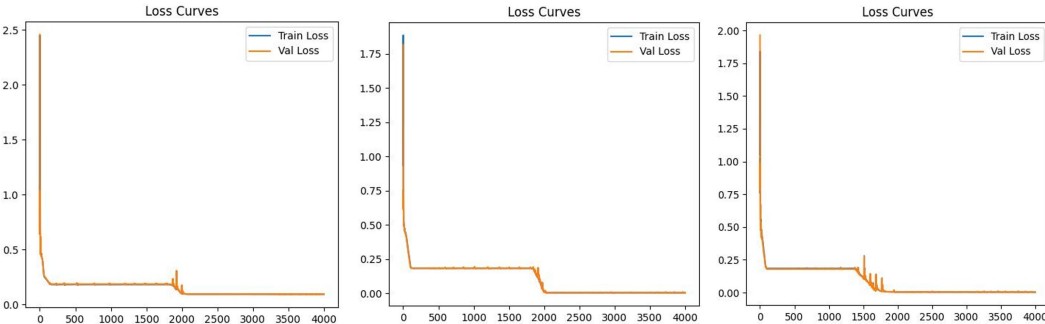

Figure 6: training with batchsize 32768 of Adam

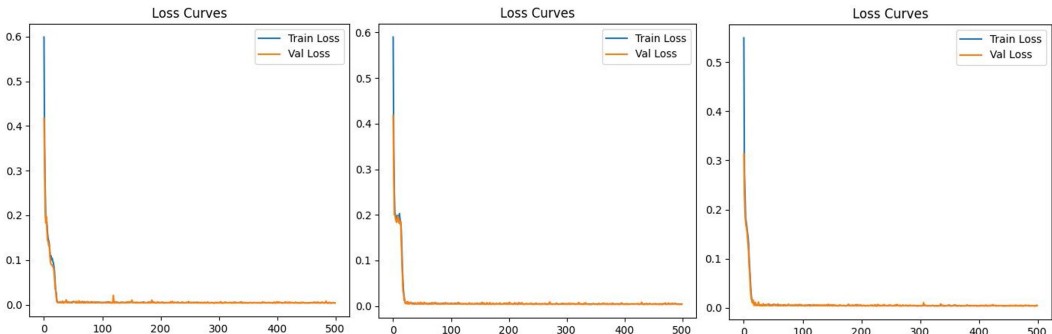

Figure 7: training with batchsize 256 of Adam

In this task, we constructed a three-layer fully connected neural network, with the final layer consisting of a single neuron to fit the regression task: y=sin(x), where the model input is x and the output is y. For this simple univariate function fitting task, our training set was constructed with uniformly sampled points from y=sin(x), where x ranges from -10 to 10, with a total of 32,786 / 0.8 samples. Noise with a mean of zero was added to the corresponding y values. In the dataset, 80% was used as the training set and 20% as the validation set.

We conducted a total of six training runs, dividing them into two sets based on batch sizes: three runs with a batch size of 256, and three runs with a batch size of 32,768. The resulting loss-epoch graphs are shown in Figures 6 and 7. It can be observed that in all instances of training with the larger batch size, the loss-epoch graphs exhibited a 'staircase' pattern, characterized by a sudden drop in the loss value at a certain point. This phenomenon was not observed in the training runs using the smaller batch size. It is noteworthy that in this study, we did not employ a learning rate scheduler, but rather trained consistently with a fixed learning rate. Therefore, the occurrence of this phenomenon warrants particular attention.

In a common training task, we would stop too early to the 'sudden step' observed in the training with larger batch sizes. Training that number of epochs consumes too much computational resource. Notably, this phenomenon occurred even in such a simple task with a basic network architecture. Therefore, in experiments involving more complex tasks and models, it's possible that this 'sudden step' will occur. Please notice that the training finishes using 20 or even less epochs in small batch case. In contrast, the training finishes after around 2000 epochs. This means 32786-batch training requires more than 100 times of epochs that are required by 256-batch training, meanwhile 100 approximate $\frac{32786}{256}$. In fact, it is impossible to train so many epochs (no acceleration in large-batch

training). The only method is to enlarge learning rate with same proportion. However, the batch size can be infinitely enlarged, whereas the learning rate can be not.

# B  Adapted Optimizers

---

Algorithm 3: ESGD

**Input:** initialize $\boldsymbol{\theta}_1$, learning rate $\eta_k$, momentum factor $(\beta_1, \beta_2) \in [0, 1)^3$, acceleration factor $|\alpha| < 1/(1 - \beta_1)$, $\mathbf{a}_0 = 0$, and $\mathbf{m}_0 = 0$.
**Output:** $\{\boldsymbol{\theta}_k\}_{k=1}^{T}$.
1: **while** $k < T$ **do**
2:    $\mathbf{g}_k = \frac{1}{|\mathcal{Z}_k|} \sum_{\boldsymbol{\zeta}_i \in \mathcal{Z}_k} \nabla \ell(\boldsymbol{\theta}_k, \boldsymbol{\zeta}_i)$;
3:    $\mathbf{a}_k = \beta_1 \mathbf{a}_{k-1} + (1 - \beta_1)(\mathbf{g}_k - \mathbf{g}_{k-1})$
4:    $\mathbf{m}_k = \beta_2 \mathbf{m}_{k-1} + (\mathbf{g}_k + \alpha \mathbf{a}_k)$;
5:    $\boldsymbol{\theta}_{k+1} = \boldsymbol{\theta}_k - \eta \mathbf{m}_k$
6: **end while**

Algorithm 4: EAdam

**Input:** initialize $\boldsymbol{\theta}_1$, learning rate $\eta_k$, momentum factor $(\beta_1, \beta_2, \beta_3) \in [0, 1)^3$, stable parameter $\varepsilon_1 > 0$, acceleration factor $|\alpha| < 1/(1 - \beta_1)$, $\mathbf{a}_0 = 0$, $\mathbf{m}_0 = 0$, and $\mathbf{v}_0 = 0$.
**Output:** $\{\boldsymbol{\theta}_k\}_{k=1}^{T}$.
1: **while** $k < T$ **do**
2:    $\mathbf{g}_k = \frac{1}{|\mathcal{Z}_k|} \sum_{\boldsymbol{\zeta}_i \in \mathcal{Z}_k} \nabla \ell(\boldsymbol{\theta}_k, \boldsymbol{\zeta}_i)$;
3:    $\mathbf{a}_k = \beta_1 \mathbf{a}_{k-1} + (1 - \beta_1)(\mathbf{g}_k - \mathbf{g}_{k-1})$;
4:    $\mathbf{m}_k = \beta_2 \mathbf{m}_{k-1} + (1 - \beta_2)(\mathbf{g}_k + \alpha \mathbf{a}_k)$;
5:    $\mathbf{v}_k = \beta_3 \mathbf{v}_{k-1} + (1 - \beta_3)[\mathbf{g}_k + \alpha \mathbf{a}_k]^2$;
6:    $\hat{\mathbf{m}}_k = \mathbf{m}_k / (1 - \beta_2^k)$;
7:    $\hat{\mathbf{v}}_k = \mathbf{v}_k / (1 - \beta_3^k)$;
8:    $\boldsymbol{r}_k = \hat{\mathbf{m}}_k / (\sqrt{\hat{\mathbf{v}}_k} + \varepsilon_1)$;
9:    $\boldsymbol{\theta}_{k+1} = \boldsymbol{\theta}_k - \eta \boldsymbol{r}_k$
10: **end while**

---

Algorithm 5: Generic form of E-adapted optimizer

---

**Initialize** $\boldsymbol{\theta}_1$, gradient estimation operation $\phi(\cdot)$, standard origin optimzier updating operation $\psi(\cdot)$, number of iterations $T$, and learning rate $\eta_k > 0$ at iteration $k$,
**for** $k = 1, 2, \ldots, T$ **do**
  **1. Gradient estimation:**

$$\mathbf{g}_k = \phi(\{\ell(\boldsymbol{\theta}_k, \boldsymbol{\zeta}_j)\}_{j \in \mathcal{Z}_k}^{k}), \tag{6}$$

where $\mathcal{Z}_k$ denotes a set of mini-batch stochastic samples used at iteration $k$,
  **2. Gradient replacement:**

$$\textcolor{red}{\mathbf{g}_k \leftarrow \mathbf{g}_k + \alpha \mathbf{a}_k, \text{where } \mathbf{a}_k = \beta_1 \mathbf{a}_{k-1} + (1 - \beta_1)(\mathbf{g}_k - \mathbf{g}_{k-1})}, \tag{7}$$

  **3. Standard parameter updating of original optimizer:**

$$\boldsymbol{\theta}_t = \psi(\mathbf{g}_k, \mathbf{g}_{k-1}, \ldots, \mathbf{g}_1, \boldsymbol{\theta}_1, \eta_t). \tag{8}$$

**end for**

---

# C  Convergence Analysis

## C.1  Preliminary

Before starting the proof, we first provide all notations here for looking up. Let

1. $\left\langle \mathbf{x}, \left(\sqrt{\mathbf{v}_k} + \varepsilon_1\right) \mathbf{y} \right\rangle_{\sqrt{\mathbf{v}_k}} := \left\langle \mathbf{x}, \left(\sqrt{\mathbf{v}_k} + \varepsilon_1\right) \mathbf{y} \right\rangle$,
2. $\mathbf{u}_k := \beta_2 \mathbf{u}_{k-1} + (1 - \beta_2) \mathbf{g}_k$,
3. $\mathbf{n}_k := \beta_2 \mathbf{n}_{k-1} + (1 - \beta_2) \mathbf{a}_k$,
4. $\mathbf{m}_k = \mathbf{u}_k + \alpha \mathbf{n}_k$
5. $\mathbf{p}_k := \mathbf{m}_k / (\sqrt{\mathbf{v}_k} + \varepsilon_1)$
6. $\varepsilon_2 := \hat{\varepsilon}_2 + \lambda_k \eta$
7. $\tilde{\boldsymbol{\theta}}_k := (\sqrt{\mathbf{v}_k} + \varepsilon_1) \boldsymbol{\theta}_k$

## C.2 Convergence Analysis of ALTO for Non-Convex Optimization

In this subsection, we give the convergence analysis of ALTO in non-convex case for Algorithm 6.

---

**Algorithm 6: ALTO Vanilla**

---

**Input:** initialize $\boldsymbol{\theta}_1$, learning rate $\eta_k$, momentum factor $(\beta_1, \beta_2, \beta_3) \in [0,1)^3$, stable parameter $\varepsilon_1, \varepsilon_2 > 0$, acceleration factor $|\alpha| < 1/\beta_1$, weight decay $\lambda_k > 0$, $\mathbf{a}_0 = 0$, $\mathbf{m}_0 = 0$, and $\mathbf{v}_0 = 0$.

**Output:** $\{\boldsymbol{\theta}_k\}_{k=1}^T$.

1: **while** $k < T$ **do**
2:     $\mathbf{g}_k = \frac{1}{|\mathcal{Z}_k|} \sum_{\boldsymbol{\zeta}_i \in \mathcal{Z}_k} \nabla \ell(\boldsymbol{\theta}_k, \boldsymbol{\zeta}_i)$;
3:     $\mathbf{a}_k = (1 - \beta_1)\mathbf{a}_{k-1} + \beta_1(\mathbf{g}_k - \mathbf{g}_{k-1})$
4:     $\mathbf{m}_k = (1 - \beta_2)\mathbf{m}_{k-1} + \beta_2(\mathbf{g}_k + \alpha \mathbf{a}_k)$;
5:     $\mathbf{v}_k = (1 - \beta_3)\mathbf{v}_{k-1} + \beta_3[\mathbf{g}_k + \alpha \mathbf{a}_k]^2$;
6:     $\boldsymbol{r}_k = \mathbf{m}_k / (\sqrt{\mathbf{v}_k} + \varepsilon_1) + \lambda \boldsymbol{\theta}_k$
7:     $\boldsymbol{\theta}_{k+1}^{(i)} = \boldsymbol{\theta}_k^{(i)} - \frac{\eta_k \mathbf{r}_k^{(i)} \phi\left(\|\boldsymbol{\theta}_k^{(i)}\|\right)}{\left(\|\mathbf{r}_k^{(i)}\| + \varepsilon_2 \phi\left(\|\boldsymbol{\theta}_k^{(i)}\|\right)\right)}$
8: **end while**

---

**Lemma 1.** *If Assumption 3.2 holds, we have*

$$\mathbb{E}\|\bar{\mathbf{g}}_k - \mathbf{g}_k\|^2 \leq \frac{\sigma^2}{b}.$$

*Proof.*

$$
\begin{aligned}
\mathbb{E}\|\bar{\mathbf{g}}_k - \mathbf{g}_k\|^2 &= \frac{1}{|\mathcal{Z}_k|^2}\| \sum_{\boldsymbol{\zeta}_i \in \mathcal{Z}_k} \mathbf{g}_k - |\mathcal{Z}_k|\,\bar{\mathbf{g}}_k\|^2 \\
&= \frac{1}{|\mathcal{Z}_k|^2} \sum_{\boldsymbol{\zeta}_i \in \mathcal{Z}_k} \|\ell(\boldsymbol{\theta}_k, \boldsymbol{\zeta}_i) - \mathbb{E}\ell(\boldsymbol{\theta}, \boldsymbol{\zeta})\|^2 \\
&\leq \frac{\tilde{\sigma}^2}{b}
\end{aligned}
$$

$\square$

**Lemma 2** (Bound for $\|(\boldsymbol{\theta}_{k+1} - \boldsymbol{\theta}_k)\|^2$ and $\|(\boldsymbol{\theta}_{k+1} - \boldsymbol{\theta}_k)\|_{\sqrt{\mathbf{v}_k}}^2$)**.**

$$\frac{\eta^2 \varepsilon_3^2}{G_\infty^2 d}\mathbb{E}\left\|\mathbf{m}_k + \lambda_k \tilde{\boldsymbol{\theta}}_k\right\|^2 \leq \mathbb{E}\|\boldsymbol{\theta}_k - \boldsymbol{\theta}_{k+1}\|^2 \leq \mathbb{E}\left\|\mathbf{m}_k + \lambda_k \tilde{\boldsymbol{\theta}}_k\right\|^2 \frac{\eta^2}{\varepsilon_1^2(\hat{\varepsilon}_2 + \lambda_k \eta)^2} \leq \mathbb{E}\left\|\mathbf{m}_k + \lambda_k \tilde{\boldsymbol{\theta}}_k\right\|^2 \frac{\eta^2}{\varepsilon_1^2 \hat{\varepsilon}_2^2}.$$

$$\frac{\eta^2 \varepsilon_3^2}{G_\infty d}\mathbb{E}\left\|\mathbf{m}_k + \lambda_k \tilde{\boldsymbol{\theta}}_k\right\|^2 \leq \mathbb{E}\|\boldsymbol{\theta}_k - \boldsymbol{\theta}_{k+1}\|_{\sqrt{\mathbf{v}_k}}^2 \leq \mathbb{E}\left\|\mathbf{m}_k + \lambda_k \tilde{\boldsymbol{\theta}}_k\right\|^2 \frac{\eta^2}{\varepsilon_1(\hat{\varepsilon}_2 + \lambda_k \eta)^2} \leq \mathbb{E}\left\|\mathbf{m}_k + \lambda_k \tilde{\boldsymbol{\theta}}_k\right\|^2 \frac{\eta^2}{\varepsilon_1 \hat{\varepsilon}_2^2}.$$

*Proof.* According to the update equation

$$\boldsymbol{\theta}_{k+1}^{(i)} = \boldsymbol{\theta}_k^{(i)} - \eta \mathbf{r}_k^{(i)} \frac{\phi\left(\left\|\boldsymbol{\theta}_k^{(i)}\right\|\right)}{\left\|\mathbf{r}_k^{(i)}\right\| + \varepsilon_2 \phi\left(\left\|\boldsymbol{\theta}_k^{(i)}\right\|\right)},$$

we have

$$\lambda_k \boldsymbol{\theta}_k + \mathbf{p}_k = \frac{\lambda_k \tilde{\boldsymbol{\theta}}_k + \mathbf{m}_k}{\sqrt{\mathbf{v}_k} + \varepsilon_1} = \frac{\left\|\mathbf{r}_k^{(i)}\right\| + \varepsilon_2 \phi\left(\left\|\boldsymbol{\theta}_k^{(i)}\right\|\right)}{\eta \phi\left(\left\|\boldsymbol{\theta}_k^{(i)}\right\|\right)}(\boldsymbol{\theta}_k - \boldsymbol{\theta}_{k+1}).$$

Then, compute the second moment on both side

$$\mathbb{E}\left\|(\boldsymbol{\theta}_k - \boldsymbol{\theta}_{k+1})\right\|^2 = \mathbb{E}\left\|\frac{\lambda_k \tilde{\boldsymbol{\theta}}_k + \mathbf{m}_k}{\sqrt{\mathbf{v}_k} + \varepsilon_1}\frac{\eta\phi\left(\left\|\boldsymbol{\theta}_k^{(i)}\right\|\right)}{\left\|\mathbf{r}_k^{(i)}\right\| + \varepsilon_2\phi\left(\left\|\boldsymbol{\theta}_k^{(i)}\right\|\right)}\right\|^2$$

$$\leq \mathbb{E}\left\|\mathbf{m}_k + \lambda_k\tilde{\boldsymbol{\theta}}_k\right\|^2 \frac{\eta^2}{\varepsilon_1^2\left(\hat{\varepsilon}_2 + \lambda_k\eta\right)^2} \leq \mathbb{E}\left\|\mathbf{m}_k + \lambda_k\tilde{\boldsymbol{\theta}}_k\right\|^2 \frac{\eta^2}{\varepsilon_1^2\hat{\varepsilon}_2^2}.$$

we find the following estimate

$$\frac{\eta\varepsilon_3}{G_\infty\sqrt{d}} \leq \frac{1}{\sqrt{\mathbf{v}_k} + \varepsilon_1}\frac{\eta\phi\left(\left\|\boldsymbol{\theta}_k^{(i)}\right\|\right)}{\left\|\mathbf{r}_k^{(i)}\right\| + \varepsilon_2\phi\left(\left\|\boldsymbol{\theta}_k^{(i)}\right\|\right)} \leq \frac{\eta}{\varepsilon_1\left(\hat{\varepsilon}_2 + \lambda_k\eta\right)} \leq \frac{\eta}{\varepsilon_1\hat{\varepsilon}_2}.$$

based on [Lemma 10](), switching $\beta_2$ and $\beta_3$ pair with $1 - \beta_2$ and $1 - \beta_3$ pair, then with unrelated constant omitted, we have

$$\frac{\left\|\mathbf{r}_k^{(i)}\right\|}{\phi\left(\left\|\boldsymbol{\theta}_k^{(i)}\right\|\right)} \leq \frac{\beta_2}{\varepsilon_3}\sqrt{\frac{d\left(1 - \beta_3\right)}{\beta_3\left(\left(1 - \beta_3\right) - \left(1 - \beta_2\right)^2\right)}} \sim \frac{\sqrt{d}}{\varepsilon_3}$$

Thus,

$$\frac{\eta^2\varepsilon_3^2}{G_\infty^2 d}\mathbb{E}\left\|\mathbf{m}_k + \lambda_k\tilde{\boldsymbol{\theta}}_k\right\|^2 \leq \mathbb{E}\left\|\boldsymbol{\theta}_k - \boldsymbol{\theta}_{k+1}\right\|^2 \leq \mathbb{E}\left\|\mathbf{m}_k + \lambda_k\tilde{\boldsymbol{\theta}}_k\right\|^2 \frac{\eta^2}{\varepsilon_1^2\left(\hat{\varepsilon}_2 + \lambda_k\eta\right)^2} \leq \mathbb{E}\left\|\mathbf{m}_k + \lambda_k\tilde{\boldsymbol{\theta}}_k\right\|^2 \frac{\eta^2}{\varepsilon_1^2\hat{\varepsilon}_2^2}.$$

$$\frac{\eta^2\varepsilon_3^2}{G_\infty d}\mathbb{E}\left\|\mathbf{m}_k + \lambda_k\tilde{\boldsymbol{\theta}}_k\right\|^2 \leq \mathbb{E}\left\|\boldsymbol{\theta}_k - \boldsymbol{\theta}_{k+1}\right\|^2_{\sqrt{\mathbf{v}_k}} \leq \mathbb{E}\left\|\mathbf{m}_k + \lambda_k\tilde{\boldsymbol{\theta}}_k\right\|^2 \frac{\eta^2}{\varepsilon_1\left(\hat{\varepsilon}_2 + \lambda_k\eta\right)^2} \leq \mathbb{E}\left\|\mathbf{m}_k + \lambda_k\tilde{\boldsymbol{\theta}}_k\right\|^2 \frac{\eta^2}{\varepsilon_1\hat{\varepsilon}_2^2}.$$

$\square$

**Lemma 3.** *If [Assumptions 3.2]() and [3.3]() hold. We have:* $\left\|\mathbf{u}_k\right\|_\infty \leq G_\infty, \left\|\mathbf{m}_k\right\|_\infty \leq G_\infty, \left\|\mathbf{v}_k\right\|_\infty \leq G_\infty^2.$

*Proof.* By the definition of $\mathbf{a}_k, \mathbf{u}_k, \mathbf{n}_k, \mathbf{v}_k$, we can have that:

$$\mathbf{n}_k = \beta_2\sum_{t=1}^k\left(1 - \beta_2\right)^{k-t}\mathbf{a}_t$$

$$\mathbf{v}_k = \beta_3\sum_{t=1}^k\left(1 - \beta_3\right)^{k-t}\left(\mathbf{g}_t + \alpha_t\mathbf{a}_t\right)^2$$

$$\mathbf{u}_k = \beta_2\sum_{t=1}^k\left(1 - \beta_2\right)^{k-t}\mathbf{g}_t$$

$$\mathbf{a}_k = \beta_1\sum_{t=1}^k\left(1 - \beta_1\right)^{k-t}\left(\mathbf{g}_t - \mathbf{g}_{t-1}\right)$$

$$= -\beta_1\sum_{t=1}^k\beta_1\left(1 - \beta_1\right)^{k-t-1}\mathbf{g}_t + \beta_1\mathbf{g}_k.$$

Considering $|\alpha\beta_1| < 1$, we get:

$$\|\mathbf{u}_k\|_\infty \le \frac{G_\infty}{2},$$

$$\|\mathbf{a}_k\|_\infty \le \frac{\beta_1 G_\infty}{2}$$

$$\|\mathbf{n}_k\|_\infty =\le \frac{\beta_1 G_\infty}{2}$$

$$\|\mathbf{m}_k\|_\infty = \|\mathbf{u}_k + \alpha\mathbf{n}_k\|_\infty \le G_\infty$$

$$\|\mathbf{v}_k\|_\infty \le G_\infty^2.$$

$\square$

**Lemma 4.** *If Assumptions 3.2 and 3.3 hold, we have:*

$$\left| \left( \frac{\sqrt{\mathbf{v}_{k-1}} + \varepsilon}{\sqrt{\mathbf{v}_k} + \varepsilon} \right)_i - 1 \right| \le \frac{\sqrt{\beta_3} G_\infty}{\varepsilon},$$

*which implies*

$$\lambda_{k+1} \|\boldsymbol{\theta}_{k+1}\|_{\sqrt{\mathbf{v}_{k+1}}}^2 \le \frac{\lambda_{k+1}}{1-\mu} \|\boldsymbol{\theta}_{k+1}\|_{\sqrt{\mathbf{v}_k}}^2 = \lambda_k \|\boldsymbol{\theta}_{k+1}\|_{\sqrt{\mathbf{v}_k}}^2.$$

*Proof.* Give any index $i \in [d]$ and the definitions of $\mathbf{v}_k$, we have:

$$\left| \left( \frac{\sqrt{\mathbf{v}_{k-1}} + \varepsilon}{\sqrt{\mathbf{v}_k} + \varepsilon} \right)_i - 1 \right| = \left| \left( \frac{\sqrt{\mathbf{v}_{k-1}} - \sqrt{\mathbf{v}_k}}{\sqrt{\mathbf{v}_k} + \varepsilon} \right)_i \right|.$$

Note that, by the definition of $\mathbf{v}_k$, we have:

$$\left| \left( \frac{\sqrt{\mathbf{v}_{k-1}} - \sqrt{\mathbf{v}_k}}{\sqrt{\mathbf{v}_k} + \varepsilon} \right)_i \right| \le \left| \left( \frac{\sqrt{|\mathbf{v}_{k-1} - \mathbf{v}_k|}}{\sqrt{\mathbf{v}_k} + \varepsilon} \right)_i \right|$$

$$= \sqrt{\beta_3} \left( \frac{\sqrt{\left|\mathbf{v}_{k-1} - (\mathbf{g}_k + \alpha\mathbf{a}_k)^2\right|}}{\sqrt{\mathbf{v}_k} + \varepsilon} \right)_i \le \frac{\sqrt{\beta_3} G_\infty}{\varepsilon},$$

$\square$

**Lemma 5.** *Consider a moving average sequence:*

$$\mathbf{u}_k = (1 - \beta_2)\mathbf{u}_{k-1} + \beta_2\mathbf{g}_k, \tag{9}$$

*Then we have:*

$$\mathbb{E}\left( \|\mathbf{u}_k - \bar{\mathbf{g}}_k\|^2 \right) \le (1 - \beta_2)\mathbb{E}\left( \|\mathbf{u}_{k-1} - \bar{\mathbf{g}}_{k-1}\|^2 \right) + \frac{(1-\beta_2)^2 L^2}{\beta_2}\mathbb{E}\left( \|\boldsymbol{\theta}_{k-1} - \boldsymbol{\theta}_k\|^2 \right) + \beta_2^2\sigma^2.$$

*Proof.* According to Equation (9), we have

$$\mathbf{u}_k - \bar{\mathbf{g}}_k = (1 - \beta_2)\left(\mathbf{u}_{k-1} - \bar{\mathbf{g}}_{k-1}\right) + (1 - \beta_2)\left(\bar{\mathbf{g}}_{k-1} - \bar{\mathbf{g}}_k\right) + \beta_2\left(\mathbf{g}_k - \bar{\mathbf{g}}_k\right).$$

Then, take the second moment and then expectation on both sides:

$$\mathbb{E}\left(\|\mathbf{u}_k - \bar{\mathbf{g}}_k\|^2\right)$$

$$=(1-\beta_2)^2\mathbb{E}\left(\|\mathbf{u}_{k-1} - \bar{\mathbf{g}}_{k-1}\|^2\right) + (1-\beta_2)^2\mathbb{E}\left(\|\bar{\mathbf{g}}_{k-1} - \bar{\mathbf{g}}_k\|^2\right) + \frac{\beta_2^2\sigma^2}{b} +$$

$$2(1-\beta_2)^2\mathbb{E}\left(\langle\mathbf{u}_{k-1} - \bar{\mathbf{g}}_{k-1}, \bar{\mathbf{g}}_{k-1} - \bar{\mathbf{g}}_k\rangle\right)$$

$$\leq \left((1-\beta_2)^2 + (1-\beta_2)^2 a\right)\mathbb{E}\left(\|\mathbf{u}_{k-1} - \bar{\mathbf{g}}_{k-1}\|^2\right) +$$

$$\left(1+\frac{1}{a}\right)(1-\beta_2)^2\mathbb{E}\left(\|\bar{\mathbf{g}}_{k-1} - \bar{\mathbf{g}}_k\|^2\right) + \frac{\beta_2^2\sigma^2}{b}$$

$$\overset{(a)}{\leq}(1-\beta_2)\mathbb{E}\left(\|\mathbf{u}_{k-1} - \bar{\mathbf{g}}_{k-1}\|^2\right) + \frac{(1-\beta_2)^2}{\beta_2}\mathbb{E}\left(\|\bar{\mathbf{g}}_{k-1} - \bar{\mathbf{g}}_k\|^2\right) + \frac{\beta_2^2\sigma^2}{b}$$

$$\leq(1-\beta_2)\mathbb{E}\left(\|\mathbf{u}_{k-1} - \bar{\mathbf{g}}_{k-1}\|^2\right) + \frac{(1-\beta_2)^2 L^2}{\beta_2}\mathbb{E}\left(\|\boldsymbol{\theta}_{k-1} - \boldsymbol{\theta}_k\|^2\right) + \frac{\beta_2^2\sigma^2}{b},$$

where for (a), we set $a = \frac{\beta_2}{1-\beta_2}$. $\qquad\square$

**Lemma 6.** *Consider a moving average sequence:*

$$\mathbf{a}_k = (1-\beta_1)\mathbf{a}_{k-1} + \beta_1\left(\mathbf{g}_k - \mathbf{g}_{k-1}\right) \tag{10}$$

*Then we have:*

$$\mathbb{E}\left(\|\mathbf{a}_k\|^2\right) \leq (1-\beta_1)\mathbb{E}\left(\|\mathbf{a}_{k-1}\|^2\right) + \beta_1\mathbb{E}\left(\|\bar{\mathbf{g}}_k - \bar{\mathbf{g}}_{k-1}\|^2\right) + 2\beta_1^2\sigma^2.$$

*Proof.* Take the second moment and then expectation on both sides of Equation (10):

$$\mathbb{E}\left(\|\mathbf{a}_k\|^2\right) = (1-\beta_1)^2\mathbb{E}\left(\|\mathbf{a}_{k-1}\|^2\right) + \beta_1^2\mathbb{E}\left(\|\mathbf{g}_k - \mathbf{g}_{k-1}\|^2\right) + 2\beta_1(1-\beta_1)\mathbb{E}\left(\langle\mathbf{a}_{k-1}, \mathbf{g}_k - \mathbf{g}_{k-1}\rangle\right)$$

$$= (1-\beta_1)^2\mathbb{E}\left(\|\mathbf{a}_{k-1}\|^2\right) + \beta_1^2\mathbb{E}\left(\|\mathbf{g}_k - \mathbf{g}_{k-1}\|^2\right) + 2\beta_1(1-\beta_1)\mathbb{E}\left(\langle\mathbf{a}_{k-1}, \bar{\mathbf{g}}_k - \mathbf{g}_{k-1}\rangle\right) \tag{11}$$

$$\leq (1-\beta_1)^2\mathbb{E}\left(\|\mathbf{a}_{k-1}\|^2\right) + \beta_1^2\mathbb{E}\left(\|\bar{\mathbf{g}}_k - \bar{\mathbf{g}}_{k-1}\|^2\right) + 2\beta_1(1-\beta_1)\mathbb{E}\left(\langle\mathbf{a}_{k-1}, \bar{\mathbf{g}}_k - \mathbf{g}_{k-1}\rangle\right) + \frac{2\beta_1^2\sigma^2}{b} \tag{12}$$

$$\leq (1-\beta_1)^2\mathbb{E}\left(\|\mathbf{a}_{k-1}\|^2\right) + \beta_1^2\mathbb{E}\left(\|\bar{\mathbf{g}}_k - \bar{\mathbf{g}}_{k-1}\|^2\right) + 2\beta_1(1-\beta_1)\mathbb{E}\left(\langle\mathbf{a}_{k-1}, \bar{\mathbf{g}}_k - \bar{\mathbf{g}}_{k-1}\rangle\right) + \frac{2\beta_1^2\sigma^2}{b} \tag{13}$$

$$\leq (1-\beta_1)\mathbb{E}\left(\|\mathbf{a}_{k-1}\|^2\right) + \beta_1\mathbb{E}\left(\|\bar{\mathbf{g}}_k - \bar{\mathbf{g}}_{k-1}\|^2\right) + \frac{2\beta_1^2\sigma^2}{b}, \tag{14}$$

where for Equation (11), we utilize the independence between $\mathbf{g}_k$ and $\mathbf{a}_{k-1}$, while for inequality (12):

$$\mathbb{E}\left(\|\mathbf{g}_k - \mathbf{g}_{k-1}\|^2\right) \leq \mathbb{E}\left(\|\mathbf{g}_k - \bar{\mathbf{g}}_k\|^2\right) + \mathbb{E}\left(\|\bar{\mathbf{g}}_{k-1} - \mathbf{g}_{k-1}\|^2\right) + \mathbb{E}\left(\|\bar{\mathbf{g}}_k - \bar{\mathbf{g}}_{k-1}\|^2\right),$$

for inequality (13), we know:

$$\mathbb{E}\left(\langle\mathbf{a}_{k-1}, \bar{\mathbf{g}}_{k-1} - \mathbf{g}_{k-1}\rangle\right)$$
$$= \mathbb{E}\left(\langle(1-\beta_1)\mathbf{a}_{k-2} + \beta_1\left(\mathbf{g}_{k-1} - \mathbf{g}_{k-2}\right), \bar{\mathbf{g}}_{k-1} - \mathbf{g}_{k-1}\rangle\right)$$
$$= \mathbb{E}\left(\langle(1-\beta_1)\mathbf{a}_{k-2} - \beta_1\mathbf{g}_{k-2}, \bar{\mathbf{g}}_{k-1} - \mathbf{g}_{k-1}\rangle\right) + \beta_1\mathbb{E}\left(\langle\mathbf{g}_{k-1} - \bar{\mathbf{g}}_{k-1} + \bar{\mathbf{g}}_{k-1}, \bar{\mathbf{g}}_{k-1} - \mathbf{g}_{k-1}\rangle\right)$$
$$= -\beta_1\mathbb{E}\left(\|\bar{\mathbf{g}}_{k-1} - \mathbf{g}_{k-1}\|^2\right),$$

and thus

$$\mathbb{E}\left(\langle\mathbf{a}_{k-1}, \bar{\mathbf{g}}_k - \mathbf{g}_{k-1}\rangle\right) = \mathbb{E}\left(\langle\mathbf{a}_{k-1}, \bar{\mathbf{g}}_k - \bar{\mathbf{g}}_{k-1}\rangle\right) - \beta_1\mathbb{E}\left(\|\bar{\mathbf{g}}_{k-1} - \mathbf{g}_{k-1}\|^2\right).$$

Finally, for inequality (14), we use:

$$2\mathbb{E}\left(\langle\mathbf{a}_{k-1}, \bar{\mathbf{g}}_k - \bar{\mathbf{g}}_{k-1}\rangle\right) \leq \mathbb{E}\left(\|\mathbf{a}_{k-1}\|^2\right) + \mathbb{E}\left(\|\bar{\mathbf{g}}_k - \bar{\mathbf{g}}_{k-1}\|^2\right).$$

$\qquad\square$

**Lemma 7.** *Consider a moving average sequence:*

$$\mathbf{n}_k = (1 - \beta_2)\mathbf{n}_{k-1} + \beta_2\mathbf{a}_k, \tag{15}$$

*we have:*

$$\mathbb{E}\left(\|\mathbf{n}_k\|^2\right) \le (1 - \beta_2)\mathbb{E}\left(\|\mathbf{n}_{k-1}\|^2\right) + \beta_2\mathbb{E}\left(\|\mathbf{a}_k\|^2\right).$$

*Proof.* Take the second moment and then expectation on both sides of Equation (15):

$$\mathbb{E}\left(\|\mathbf{n}_k\|^2\right) = \mathbb{E}\left(\|(1 - \beta_2)\mathbf{n}_{k-1} + \beta_2\mathbf{a}_k\|^2\right),$$

$$\le (1 - \beta_2)\mathbb{E}\left(\|\mathbf{n}_{k-1}\|^2\right) + \beta_2\mathbb{E}\left(\|\mathbf{g}_k - \mathbf{g}_{k-1}\|^2\right).$$

$\square$

**Lemma 8.** *Assume Assumption 3.1 holds, with $\eta \le \frac{\varepsilon_1\hat{\varepsilon}_2}{3L}$, and then we have:*

$$f_{k+1}\left(\boldsymbol{\theta}_{k+1}\right) \le f_k\left(\boldsymbol{\theta}_k\right) - \frac{\hat{\varepsilon}_2\eta\varepsilon_3^2}{3dG_\infty}\mathbb{E}\left\|\mathbf{m}_k + \lambda_k\tilde{\boldsymbol{\theta}}_k\right\|^2 + \frac{\eta}{\varepsilon_1\hat{\varepsilon}_2}\mathbb{E}\|\bar{\mathbf{g}}_k - \mathbf{m}_k\|^2 + \frac{\sigma^2\eta}{b\varepsilon_1\hat{\varepsilon}_2}.$$

*Proof.* Recall that the update of ALTO is the following

$$\boldsymbol{\theta}_{k+1}^{(i)} = \boldsymbol{\theta}_k^{(i)} - \eta_k\mathbf{r}_k^{(i)}\frac{\phi\left(\left\|\boldsymbol{\theta}_k^{(i)}\right\|\right)}{\left\|\mathbf{r}_k^{(i)}\right\| + \varepsilon_2\phi\left(\left\|\boldsymbol{\theta}_k^{(i)}\right\|\right)},$$

Thus,

$$\lambda_k\boldsymbol{\theta}_k + \mathbf{p}_k = \frac{\lambda_k\tilde{\boldsymbol{\theta}}_k + \mathbf{m}_k}{\sqrt{\mathbf{v}_k} + \varepsilon_1} = \frac{\left\|\mathbf{r}_k^{(i)}\right\| + \varepsilon_2\phi\left(\left\|\boldsymbol{\theta}_k^{(i)}\right\|\right)}{\eta\phi\left(\left\|\boldsymbol{\theta}_k^{(i)}\right\|\right)}\left(\boldsymbol{\theta}_k - \boldsymbol{\theta}_{k+1}\right). \tag{16}$$

Using Taylor expansion, we have:

$$f_{k+1}\left(\boldsymbol{\theta}_{k+1}\right) \le \mathbb{E}\left(\ell_{k+1}\left(\boldsymbol{\theta}_k\right) + \left\langle\nabla\ell_{k+1}\left(\boldsymbol{\theta}_k\right), \boldsymbol{\theta}_{k+1} - \boldsymbol{\theta}_k\right\rangle + \frac{L}{2}\|\boldsymbol{\theta}_{k+1} - \boldsymbol{\theta}_k\|^2 + \frac{\lambda_{k+1}}{2}\|\boldsymbol{\theta}_{k+1}\|^2_{\sqrt{\mathbf{v}_{k+1}}}\right)$$

$$\overset{Lemma\ 4}{\le} \mathbb{E}\left(\ell_k\left(\boldsymbol{\theta}_k\right) + \left\langle\nabla\ell_k\left(\boldsymbol{\theta}_k\right), \boldsymbol{\theta}_{k+1} - \boldsymbol{\theta}_k\right\rangle + \frac{L}{2}\|\boldsymbol{\theta}_{k+1} - \boldsymbol{\theta}_k\|^2 + \frac{\lambda_k}{2}\|\boldsymbol{\theta}_{k+1}\|^2_{\sqrt{\mathbf{v}_k}}\right)$$

$$\le f_k\left(\boldsymbol{\theta}_k\right) + \mathbb{E}\left(\left\langle\boldsymbol{\theta}_{k+1} - \boldsymbol{\theta}_k, \lambda_k\boldsymbol{\theta}_k + \frac{\mathbf{g}_k}{\sqrt{\mathbf{v}_k} + \varepsilon_1}\right\rangle_{\sqrt{\mathbf{v}_k}} + \frac{L/\varepsilon_1 + \lambda_k}{2}\|\boldsymbol{\theta}_{k+1} - \boldsymbol{\theta}_k\|^2_{\sqrt{\mathbf{v}_k}}\right) \tag{17}$$

$$= f_k\left(\boldsymbol{\theta}_k\right) + \frac{L/\varepsilon_1 + \lambda_k}{2}\mathbb{E}\|\boldsymbol{\theta}_{k+1} - \boldsymbol{\theta}_k\|^2_{\sqrt{\mathbf{v}_k}} + \mathbb{E}\left\langle\boldsymbol{\theta}_{k+1} - \boldsymbol{\theta}_k, \lambda_k\boldsymbol{\theta}_k + \mathbf{p}_k + \frac{\mathbf{g}_k - \mathbf{m}_k}{\sqrt{\mathbf{v}_k} + \varepsilon_1}\right\rangle_{\sqrt{\mathbf{v}_k}}$$

$$\overset{Equation\ (16)}{=} f_k\left(\boldsymbol{\theta}_k\right) + \left(\frac{L/\varepsilon_1 + \lambda_k}{2} - \frac{\hat{\varepsilon}_2 + \eta\lambda_k}{\eta}\right)\mathbb{E}\|\boldsymbol{\theta}_{k+1} - \boldsymbol{\theta}_k\|^2_{\sqrt{\mathbf{v}_k}} + \mathbb{E}\left\langle\boldsymbol{\theta}_{k+1} - \boldsymbol{\theta}_k, \frac{\mathbf{g}_k - \mathbf{m}_k}{\sqrt{\mathbf{v}_k} + \varepsilon_1}\right\rangle_{\sqrt{\mathbf{v}_k}}$$

$$\le f_k\left(\boldsymbol{\theta}_k\right) + \left(\frac{L/\varepsilon_1}{2} - \frac{\hat{\varepsilon}_2}{\eta}\right)\mathbb{E}\|\boldsymbol{\theta}_{k+1} - \boldsymbol{\theta}_k\|^2_{\sqrt{\mathbf{v}_k}} + \frac{\hat{\varepsilon}_2}{2\eta}\mathbb{E}\|\boldsymbol{\theta}_{k+1} - \boldsymbol{\theta}_k\|^2_{\sqrt{\mathbf{v}_k}} + \frac{\eta}{2\hat{\varepsilon}_2\varepsilon_1}\mathbb{E}\|\mathbf{g}_k - \mathbf{m}_k\|^2$$

$$\tag{18}$$

$$\le f_k\left(\boldsymbol{\theta}_k\right) - \frac{\hat{\varepsilon}_2}{3\eta}\mathbb{E}\|\boldsymbol{\theta}_{k+1} - \boldsymbol{\theta}_k\|^2_{\sqrt{\mathbf{v}_k}} + \frac{\eta}{2\varepsilon_1\hat{\varepsilon}_2}\mathbb{E}\|\mathbf{g}_k - \mathbf{m}_k\|^2 \tag{19}$$

$$\overset{Lemma\ 2}{\le} f_k\left(\boldsymbol{\theta}_k\right) - \frac{\hat{\varepsilon}_2\eta\varepsilon_3^2}{3dG_\infty}\mathbb{E}\left\|\mathbf{m}_k + \lambda_k\tilde{\boldsymbol{\theta}}_k\right\|^2 + \frac{\eta}{2\varepsilon_1\hat{\varepsilon}_2}\mathbb{E}\|\mathbf{g}_k - \mathbf{m}_k\|^2,$$

$$\le f_k\left(\boldsymbol{\theta}_k\right) - \frac{\hat{\varepsilon}_2\eta\varepsilon_3^2}{3dG_\infty}\mathbb{E}\left\|\mathbf{m}_k + \lambda_k\tilde{\boldsymbol{\theta}}_k\right\|^2 + \frac{\eta}{\varepsilon_1\hat{\varepsilon}_2}\mathbb{E}\|\bar{\mathbf{g}}_k - \mathbf{m}_k\|^2 + \frac{\eta}{\varepsilon_1\hat{\varepsilon}_2}\mathbb{E}\|\bar{\mathbf{g}}_k - \mathbf{g}_k\|^2,$$

$$\overset{Lemma\ 1}{\le} f_k\left(\boldsymbol{\theta}_k\right) - \frac{\hat{\varepsilon}_2\eta\varepsilon_3^2}{3dG_\infty}\mathbb{E}\left\|\mathbf{m}_k + \lambda_k\tilde{\boldsymbol{\theta}}_k\right\|^2 + \frac{\eta}{\varepsilon_1\hat{\varepsilon}_2}\mathbb{E}\|\bar{\mathbf{g}}_k - \mathbf{m}_k\|^2 + \frac{\sigma^2\eta}{b\varepsilon_1\hat{\varepsilon}_2},$$

and inequality (17) is from:

$$\|\boldsymbol{\theta}_{k+1}\|_{\sqrt{\mathbf{v}_k}}^2 = \left(\|\boldsymbol{\theta}_k\|_{\sqrt{\mathbf{v}_k}}^2 + 2\langle\boldsymbol{\theta}_{k+1} - \boldsymbol{\theta}_k, \boldsymbol{\theta}_k\rangle_{\sqrt{\mathbf{v}_k}} + \|\boldsymbol{\theta}_{k+1} - \boldsymbol{\theta}_k\|_{\sqrt{\mathbf{v}_k}}^2\right),$$

and to obtain inequality (18), we utilize:

$$\left\langle\boldsymbol{\theta}_{k+1} - \boldsymbol{\theta}_k, \frac{\mathbf{g}_k - \mathbf{m}_k}{\sqrt{\mathbf{v}_k} + \varepsilon_1}\right\rangle_{\sqrt{\mathbf{v}_k}} \le \frac{1}{2\eta}\|\boldsymbol{\theta}_{k+1} - \boldsymbol{\theta}_k\|_{\sqrt{\mathbf{v}_k}}^2 + \frac{\eta}{2\varepsilon_1}\|\mathbf{g}_k - \mathbf{m}_k\|^2,$$

To get inequality (19), we use condition $\eta \le \frac{\varepsilon_1 \hat{\varepsilon}_2}{3L}$.

$\square$

**Theorem 1.** *Suppose Assumption 3.1, 3.2, and 3.3 hold, $\mu := \sqrt{\beta_3} G_\infty / \varepsilon_1 < 1$,*

$$b \ge \frac{18 G_\infty \sigma^2}{\hat{\varepsilon}_2^2 \varepsilon_1 \epsilon^2 \varepsilon_3^2}, \quad \eta^2 \le \frac{\varepsilon_1^3 \hat{\varepsilon}_2^4 \varepsilon_3^2 \beta_2^2}{6dL^2 G_\infty}, \quad \alpha^2 \le \frac{dG_\infty}{\varepsilon_1 \hat{\varepsilon}_2^2 \varepsilon_3^2 \beta_2^2}, \quad T \ge \max\left\{\frac{18 G_\infty \Delta_0}{\eta \hat{\varepsilon}_2 \epsilon^2 \varepsilon_3^2}, \frac{18 G_\infty \sigma^2}{\beta_2 \hat{\varepsilon}_2^2 \varepsilon_1 \epsilon^2 \varepsilon_3^2}\right\}$$

*where $\Delta_0 := f_0(\boldsymbol{\theta}_0) - f_0^*$ and Algorithm 6 satisfies:*

$$\frac{1}{T+1}\sum_{k=0}^{T}\mathbb{E}\left(\left\|\mathbf{m}_k + \lambda_k\tilde{\boldsymbol{\theta}}_k\right\|^2\right) \le \epsilon^2,$$

*and*

$$\frac{1}{T+1}\sum_{k=0}^{T}\mathbb{E}\left(\|\mathbf{u}_k - \bar{\mathbf{g}}_k\|^2\right) \le \frac{\epsilon^2}{4}, \quad \frac{1}{T+1}\sum_{k=0}^{T}\mathbb{E}\left(\alpha^2 \|\mathbf{n}_k\|^2\right) \le \frac{\epsilon^2}{4}.$$

*Hence, we have*

$$\frac{1}{T+1}\sum_{k=0}^{T}\mathbb{E}\left(\|\nabla f_k(\boldsymbol{\theta}_k)\|^2\right) \le 4\epsilon^2.$$

*Based on these conditions, if Assumption 3.4, and*

$$T \ge \frac{6\lambda^2 D G_\infty^2}{\epsilon^2(2-\mu)\mu},$$

*we have*

$$\frac{1}{T+1}\sum_{k=0}^{T}\mathbb{E}\left(\|\nabla f(\boldsymbol{\theta}_k)\|^2\right) \le 6\epsilon^2.$$

*Proof.* We have:

$$\|\mathbf{m}_k - \bar{\mathbf{g}}_k\|^2 \le 2\|\mathbf{u}_k - \bar{\mathbf{g}}_k\|^2 + 2\alpha^2\|\mathbf{n}_k\|^2$$

By Lemmas 5 to 8, we have:

$$f_{k+1}(\boldsymbol{\theta}_{k+1}) \le f_k(\boldsymbol{\theta}_k) - \frac{\hat{\varepsilon}_2 \eta \varepsilon_3^2}{3dG_\infty}\left\|\mathbf{m}_k + \lambda_k\tilde{\boldsymbol{\theta}}_k\right\|^2 + \frac{2\eta}{\varepsilon_1 \hat{\varepsilon}_2}\|\bar{\mathbf{g}}_k - \mathbf{u}_k\|^2 + \frac{2\eta}{\varepsilon_1 \hat{\varepsilon}_2}\alpha^2\|\mathbf{n}_k\|^2 + \frac{\sigma^2\eta}{b\varepsilon_1 \hat{\varepsilon}_2} \tag{20}$$

$$\mathbb{E}\left(\|\mathbf{u}_{k+1} - \bar{\mathbf{g}}_{k+1}\|^2\right) \le (1-\beta_2)\mathbb{E}\left(\|\mathbf{u}_k - \bar{\mathbf{g}}_k\|^2\right) + \frac{(1-\beta_2)^2 L^2}{\beta_2}\mathbb{E}\left(\|\boldsymbol{\theta}_{k+1} - \boldsymbol{\theta}_k\|^2\right) + \frac{\beta_2^2\sigma^2}{b} \tag{21}$$

$$\mathbb{E}\left(\|\mathbf{n}_{k+1}\|^2\right) \le (1-\beta_2)\mathbb{E}\left(\|\mathbf{n}_k\|^2\right) + \beta_2\mathbb{E}\left(\|\mathbf{a}_k\|^2\right) \tag{22}$$

$$\mathbb{E}\left(\|\mathbf{a}_{k+1}\|^2\right) \le (1-\beta_1)\mathbb{E}\left(\|\mathbf{a}_k\|^2\right) + \beta_1\mathbb{E}\left(\|\bar{\mathbf{g}}_{k+1} - \bar{\mathbf{g}}_k\|^2\right) + \frac{2\beta_1^2\sigma^2}{b} \tag{23}$$

Then by adding Equation (20) with $\frac{\eta}{\beta_2 \varepsilon_1 \hat{\varepsilon}_2} \times$ Equation (21), $\frac{\eta\alpha^2}{\beta_2 \varepsilon_1 \hat{\varepsilon}_2} \times$ Equation (22), and $\frac{\eta\alpha^2}{\beta_1 \varepsilon_1 \hat{\varepsilon}_2} \times$ Equation (23), we can get:

$$\mathbb{E}\left(\Phi_{k+1}\right) \le \mathbb{E}\left(\Phi_k\right) - \frac{\hat{\varepsilon}_2 \eta \varepsilon_3^2}{3dG_\infty}\mathbb{E}\left\|\mathbf{m}_k + \lambda_k \tilde{\boldsymbol{\theta}}_k\right\|^2 + \frac{\sigma^2 \eta}{b\varepsilon_1\hat{\varepsilon}_2} + \frac{\eta}{\beta_2\varepsilon_1\hat{\varepsilon}_2}\left(\frac{(1-\beta_2)^2 L^2}{\beta_2}\mathbb{E}\left\|\boldsymbol{\theta}_{k+1} - \boldsymbol{\theta}_k\right\|^2 + \frac{\beta_2^2\sigma^2}{b}\right)$$

$$+ \frac{\eta\alpha^2}{\beta_1\varepsilon_1\hat{\varepsilon}_2}\left(\beta_1 L^2 \mathbb{E}\left\|\boldsymbol{\theta}_{k+1} - \boldsymbol{\theta}_k\right\|^2 + \frac{2\beta_1^2\sigma^2}{b}\right)$$

$$\le \mathbb{E}\left(\Phi_k\right) - \frac{\hat{\varepsilon}_2 \eta \varepsilon_3^2}{3dG_\infty}\mathbb{E}\left\|\mathbf{m}_k + \lambda_k \tilde{\boldsymbol{\theta}}_k\right\|^2 + \frac{\eta L^2}{\varepsilon_1\hat{\varepsilon}_2}\left(\frac{(1-\beta_2)^2}{\beta_2^2} + \alpha^2\right)\mathbb{E}\left\|\boldsymbol{\theta}_{k+1} - \boldsymbol{\theta}_k\right\|^2 + \left(\frac{\beta_2 + 2\alpha^2\beta_1 + 1}{b}\right)\frac{\eta\sigma^2}{\varepsilon_1\hat{\varepsilon}_2}$$

$$\le \mathbb{E}\left(\Phi_k\right) - \frac{\hat{\varepsilon}_2 \eta \varepsilon_3^2}{3dG_\infty}\mathbb{E}\left\|\mathbf{m}_k + \lambda_k \tilde{\boldsymbol{\theta}}_k\right\|^2 + \frac{\eta L^2}{\beta_2^2\varepsilon_1\hat{\varepsilon}_2}\mathbb{E}\left\|\boldsymbol{\theta}_{k+1} - \boldsymbol{\theta}_k\right\|^2 + \frac{\beta_e\eta\sigma^2}{\varepsilon_1\hat{\varepsilon}_2}$$

$$\le \mathbb{E}\left(\Phi_k\right) + \left(\frac{\eta^3 L^2}{\beta_2^2\varepsilon_1^3\hat{\varepsilon}_2^3} - \frac{\hat{\varepsilon}_2 \eta \varepsilon_3^2}{3dG_\infty}\right)\mathbb{E}\left\|\mathbf{m}_k + \lambda_k \tilde{\boldsymbol{\theta}}_k\right\|^2 + \frac{\beta_e\eta\sigma^2}{\varepsilon_1\hat{\varepsilon}_2}$$

$$\le \mathbb{E}\left(\Phi_k\right) - \frac{\eta\hat{\varepsilon}_2\varepsilon_3^2}{6dG_\infty}\mathbb{E}\left\|\mathbf{m}_k + \lambda_k \tilde{\boldsymbol{\theta}}_k\right\|^2 + \frac{\beta_e\eta\sigma^2}{\varepsilon_1\hat{\varepsilon}_2},$$

where we let:

$$\Phi_k := f_k\left(\boldsymbol{\theta}_k\right) - f_k^* + \frac{\eta}{\beta_2\varepsilon_1\hat{\varepsilon}_2}\left\|\mathbf{u}_k - \bar{\mathbf{g}}_k\right\|^2 + \frac{\eta\alpha^2}{\beta_2\varepsilon_1\hat{\varepsilon}_2}\left\|\mathbf{n}_k\right\|^2 + \frac{\eta\alpha^2}{\beta_1\varepsilon_1\hat{\varepsilon}_2}\left\|\mathbf{a}_k\right\|^2,$$

$$\beta_e = \frac{\beta_2 + 2\alpha^2\beta_1 + 1}{b} \quad \eta^2 \le \frac{\varepsilon_1^3\hat{\varepsilon}_2^4\varepsilon_3^2\beta_2^2}{6dL^2G_\infty},$$

By telescoping sum, we have:

$$\sum_{k=0}^{T}\mathbb{E}\left(\Phi_{k+1}\right) \le \sum_{k=0}^{T}\mathbb{E}\left(\Phi_k\right) - \frac{\eta\hat{\varepsilon}_2\varepsilon_3^2}{6dG_\infty}\sum_{k=0}^{T}\mathbb{E}\left\|\mathbf{m}_k + \lambda_k \tilde{\boldsymbol{\theta}}_k\right\|^2 + (T+1)\frac{\beta_e\eta\sigma^2}{\varepsilon_1\hat{\varepsilon}_2}.$$

Hence, we can get:

$$\frac{1}{T+1}\sum_{k=0}^{T}\mathbb{E}\left(\left\|\mathbf{m}_k + \lambda_k \tilde{\boldsymbol{\theta}}_k\right\|^2\right) \le \frac{6dG_\infty\Phi_0}{\eta T\hat{\varepsilon}_2\varepsilon_3^2} + \frac{6dG_\infty\beta_e\sigma^2}{\varepsilon_1\hat{\varepsilon}_2^2\varepsilon_3^2} = \frac{6dG_\infty\Delta_0}{\eta T\hat{\varepsilon}_2\varepsilon_3^2} + \frac{6dG_\infty\sigma^2}{\beta_2\varepsilon_1\hat{\varepsilon}_2^2T\varepsilon_3^2} + \frac{6dG_\infty\beta_e\sigma^2}{\varepsilon_1\hat{\varepsilon}_2^2\varepsilon_3^2} \le \epsilon^2,$$

where

$$\beta_e \le \frac{\hat{\varepsilon}_2^2\varepsilon_1\epsilon^2\varepsilon_3^2}{18G_\infty\sigma^2}, \quad T \ge \max\left\{\frac{18G_\infty\Delta_0}{\eta\hat{\varepsilon}_2\epsilon^2\varepsilon_3^2}, \frac{18G_\infty\sigma^2}{\beta_2\hat{\varepsilon}_2^2\varepsilon_1\epsilon^2\varepsilon_3^2}\right\}.$$

From Equation (21), we can conclude that:

$$\frac{1}{T+1}\sum_{k=0}^{T}\mathbb{E}\left(\left\|\mathbf{u}_k - \bar{\mathbf{g}}_k\right\|^2\right) \le \frac{\sigma^2}{\beta_2 T} + \frac{L^2\eta^2\epsilon^2}{\varepsilon_1^2\hat{\varepsilon}_2^2\beta_2^2} + \frac{\beta_2\sigma^2}{b} \ll \frac{\epsilon^2}{4}.$$

From Equation (22), we can conclude that:

$$\frac{1}{T+1}\sum_{k=0}^{T}\mathbb{E}\left(\left\|\mathbf{n}_k\right\|^2\right) \le \frac{1}{T+1}\sum_{k=0}^{T}\mathbb{E}\left(\left\|\mathbf{a}_k\right\|^2\right) \tag{24}$$

From Equation (23), we can conclude that:

$$\frac{1}{T+1}\sum_{k=0}^{T}\mathbb{E}\left(\left\|\mathbf{a}_k\right\|^2\right) \le \frac{L^2\eta^2\epsilon^2}{\varepsilon_1^2\hat{\varepsilon}_2^2} + \frac{2\beta_1\sigma^2}{b} \tag{25}$$

Combining Equations (24) and (25), we have

$$\frac{1}{T+1}\sum_{k=0}^{T}\mathbb{E}\left(\alpha^2\left\|\mathbf{n}_k\right\|^2\right) \le \frac{1}{T+1}\sum_{k=0}^{T}\mathbb{E}\left(\alpha^2\left\|\mathbf{a}_k\right\|^2\right) \le \frac{\alpha^2 L^2\eta^2\epsilon^2}{\varepsilon_1^2\hat{\varepsilon}_2^2} + \frac{2\alpha^2\beta_1\sigma^2}{b} \ll \epsilon^2/4$$

where

$$\alpha^2 \le \frac{dG_\infty}{\varepsilon_1 \hat{\varepsilon}_2^2 \varepsilon_3^2 \beta_2^2}$$

Finally, for $\nabla f_k$, we have:

$$\frac{1}{T+1} \sum_{k=0}^{T} \mathbb{E}\left(\|\nabla f_k(\boldsymbol{\theta}_k)\|^2\right)$$

$$= \frac{1}{T+1} \sum_{k=0}^{T} \mathbb{E}\left(\left\|\nabla\left(\frac{\lambda_k}{2}\|\boldsymbol{\theta}_k\|_{\sqrt{\mathbf{v}_k}}^2 + \mathbb{E}_{\boldsymbol{\zeta}}\left[f\left(\boldsymbol{\theta}_k, \boldsymbol{\zeta}\right)\right]\right)\right\|^2\right)$$

$$= \frac{1}{T+1} \sum_{k=0}^{T} \mathbb{E}\left(\left\|\lambda_k \tilde{\boldsymbol{\theta}}_k + \bar{\mathbf{g}}_k + \mathbf{m}_k - \mathbf{u}_k - \alpha \mathbf{n}_k\right\|^2\right)$$

$$\le \frac{1}{T+1}\left(\sum_{k=0}^{T} \mathbb{E}\left(2\left\|\mathbf{m}_k + \lambda_k \tilde{\boldsymbol{\theta}}_k\right\|^2 + 4\|\mathbf{u}_k - \bar{\mathbf{g}}_k\|^2 + 4\alpha^2 \|\mathbf{n}_k\|^2\right)\right)$$

$$\le 4\epsilon^2.$$

If $\boldsymbol{\theta}$ is bounded, we have

$$\frac{1}{T+1} \sum_{k=0}^{T} \mathbb{E}\lambda_k^2 \left\|\tilde{\boldsymbol{\theta}}_k\right\|^2$$

$$\overset{Lemma\ 3}{\le} \frac{\lambda^2 DG_\infty^2}{T+1} \sum_{k=0}^{T} (1-\mu)^{2k}$$

$$\le \frac{\lambda^2 DG_\infty^2}{T\mu(2-\mu)}$$

$$\le \frac{\epsilon^2}{6}.$$

For $\nabla f$, we have:

$$\frac{1}{T+1} \sum_{k=0}^{T} \mathbb{E}\left(\|\nabla f(\boldsymbol{\theta}_k)\|^2\right)$$

$$= \frac{1}{T+1} \sum_{k=0}^{T} \mathbb{E}\left(\left\|\bar{\mathbf{g}}_k + (\mathbf{m}_k - \mathbf{u}_k - \alpha \mathbf{n}_k) + \lambda_k \tilde{\boldsymbol{\theta}}_k - \lambda_k \tilde{\boldsymbol{\theta}}_k\right\|^2\right)$$

$$\le \frac{1}{T+1}\left(\sum_{k=0}^{T} \mathbb{E}\left(2\left\|\mathbf{m}_k + \lambda_k \tilde{\boldsymbol{\theta}}_k\right\|^2 + 6\|\mathbf{u}_k - \bar{\mathbf{g}}_k\|^2 + 6\alpha^2 \|\mathbf{n}_k\|^2 + 6\lambda_k^2 \left\|\tilde{\boldsymbol{\theta}}_k\right\|^2\right)\right)$$

$$\le 6\epsilon^2.$$

$\square$

## C.3 Layerwise Convergence Analysis of ALTO for Convex Optimization

In this subsection, we give the convergence analysis of ALTO in non-convex case for Algorithm 7.

**Lemma 9.** *If Assumption 3.4 holds, we have*

$$\left\|\frac{\mathbf{r}_{k,i}\phi\left(\left\|\boldsymbol{\theta}_k^{(i)}\right\|\right)}{\left\|\mathbf{r}_k^{(i)}\right\| + \varepsilon_2 \phi\left(\left\|\boldsymbol{\theta}_k^{(i)}\right\|\right)}\right\| \le \sqrt{h}D_\infty.$$

---

**Algorithm 7: ALTO Vanilla**

---

**Input:** initialize $\boldsymbol{\theta}_1$, learning rate $\eta_k$, momentum factor $(\beta_1, \beta_2, \beta_3) \in [0,1)^3$, stable parameter $\varepsilon_1, \varepsilon_2 > 0$, acceleration factor $|\alpha| < 1/(1-\beta_1)$, weight decay $\lambda_k > 0$, $\mathbf{a}_0 = 0$, $\mathbf{m}_0 = 0$, and $\mathbf{v}_0 = 0$.

**Output:** $\{\boldsymbol{\theta}_k\}_{k=1}^{T}$.

    **while** $k < T$ **do**

        Compute $\mathbf{g}_k = \frac{1}{|\mathcal{Z}_k|} \sum_{\boldsymbol{\zeta}_i \in \mathcal{Z}_k} \nabla \ell (\boldsymbol{\theta}_k, \boldsymbol{\zeta}_i)$;

        $\mathbf{a}_k = \beta_1 \mathbf{a}_{k-1} + (1-\beta_1)(\mathbf{g}_k - \mathbf{g}_{k-1})$

        $\mathbf{m}_k = \beta_2 \mathbf{m}_{k-1} + (1-\beta_2)(\mathbf{g}_k + \alpha \mathbf{a}_k)$;

        $\mathbf{v}_k = \beta_3 \mathbf{v}_{k-1} + (1-\beta_3)[\mathbf{g}_k + \alpha \mathbf{a}_k]^2$;

        $\boldsymbol{r}_k = \mathbf{m}_k / (\sqrt{\mathbf{v}_k} + \varepsilon_1) + \lambda \boldsymbol{\theta}_k$

        $\boldsymbol{\theta}_{k+1}^{(i)} = \boldsymbol{\theta}_k^{(i)} - \dfrac{\eta_k \mathbf{r}_k^{(i)} \phi \left( \left\| \boldsymbol{\theta}_k^{(i)} \right\| \right)}{\left( \left\| \mathbf{r}_k^{(i)} \right\| + \varepsilon_2 \phi \left( \left\| \boldsymbol{\theta}_k^{(i)} \right\| \right) \right)}$

    **end while**

---

*Proof.* Obviously,

$$\phi \left( \left\| \boldsymbol{\theta}_k^{(i)} \right\| \right) \le D_\infty$$

and

$$\frac{\mathbf{r}_{k,i}}{\left\| \mathbf{r}_k^{(i)} \right\| + \varepsilon_2 \phi \left( \left\| \boldsymbol{\theta}_k^{(i)} \right\| \right)} \le \frac{\mathbf{r}_{k,i}}{\left\| \mathbf{r}_k^{(i)} \right\|}$$

Therefore,

$$\left\| \frac{\mathbf{r}_{k,i}}{\left\| \mathbf{r}_k^{(i)} \right\|} \right\|^2 = \sum_{j=1}^{h} \sum_{i=1}^{d_j} \frac{\mathbf{r}_{k,i}^2}{\left\| \mathbf{r}_k^{(i)} \right\|^2} = \sum_{j=1}^{h} 1 = h$$

Finally, we have

$$\left\| \frac{\mathbf{r}_{k,i} \phi \left( \left\| \boldsymbol{\theta}_k^{(i)} \right\| \right)}{\left\| \mathbf{r}_k^{(i)} \right\| + \varepsilon_2 \phi \left( \left\| \boldsymbol{\theta}_k^{(i)} \right\| \right)} \right\| \le \sqrt{h} D_\infty$$

$\square$

**Lemma 10.** *If* $\beta_2^2 / \beta_3 < 1$, *we have*

$$\frac{\left\| \mathbf{r}_k^{(i)} \right\| + \varepsilon_2 \phi \left( \left\| \boldsymbol{\theta}_k^{(i)} \right\| \right)}{\phi \left( \left\| \boldsymbol{\theta}_k^{(i)} \right\| \right)} \le \frac{1-\beta_2}{\varepsilon_3} \sqrt{\frac{d\beta_3}{(1-\beta_3)(\beta_3 - \beta_2^2)}}.$$

*Proof.*

$$\mathbf{m}_k = (1-\beta_2) \sum_{t=1}^{k} \beta_2^{k-t} (\mathbf{g}_t + \alpha_t \mathbf{a}_t)$$

$$\mathbf{v}_k = (1-\beta_3) \sum_{t=1}^{k} \beta_3^{k-t} (\mathbf{g}_t + \alpha_t \mathbf{a}_t)^2$$

Therefore, with unrelated constant omitted, we have

$$\frac{\left\| \mathbf{r}_k^{(i)} \right\| + \varepsilon_2 \phi \left( \left\| \boldsymbol{\theta}_k^{(i)} \right\| \right)}{\phi \left( \left\| \boldsymbol{\theta}_k^{(i)} \right\| \right)} \le \frac{1}{\varepsilon_3} \left\| \frac{\mathbf{m}_k}{\sqrt{\mathbf{v}_k}} \right\|.$$

In fact,

$$\left\|\frac{\mathbf{m}_k}{\sqrt{\mathbf{v}_k}}\right\| = \left(\sum_{i=1}^{d} \frac{\left((1-\beta_2)\sum_{t=1}^{k}\beta_2^{k-t}(\mathbf{g}_{t,i}+\alpha_t\mathbf{a}_{t,i})\right)^2}{(1-\beta_3)\sum_{t=1}^{k}\beta_3^{k-t}(\mathbf{g}_{t,i}+\alpha_i\mathbf{a}_{t,i})^2}\right)^{\frac{1}{2}}$$

$$= \frac{1-\beta_2}{\sqrt{1-\beta_3}}\left(\sum_{i=1}^{d}\frac{\left(\sum_{t=1}^{k}\beta_2^{k-t}(\mathbf{g}_{t,i}+\alpha_t\mathbf{a}_{t,i})\right)^2}{\sum_{t=1}^{k}\beta_3^{k-t}(\mathbf{g}_{t,i}+\alpha_i\mathbf{a}_{t,i})^2}\right)^{\frac{1}{2}}$$

$$\leq \frac{1-\beta_2}{\sqrt{1-\beta_3}}\left(\sum_{i=1}^{d}\frac{\left(\sum_{t=1}^{k}\beta_3^{k-t}(\mathbf{g}_{t,i}+\alpha_t\mathbf{a}_{t,i})^2\right)\left(\sum_{t=1}^{k}(\beta_2^2\beta_3^{-1})^{k-t}\right)}{\sum_{t=1}^{k}\beta_3^{k-t}(\mathbf{g}_{t,i}+\alpha_i\mathbf{a}_{t,i})^2}\right)^{\frac{1}{2}}$$

$$\leq (1-\beta_2)\sqrt{\frac{d\beta_3}{(1-\beta_3)(\beta_3-\beta_2^2)}}.$$

$\square$

**Lemma 11** (Bound for $\sum_{k=1}^{T}\langle\boldsymbol{\theta}^* - \boldsymbol{\theta}_k, \mathbf{m}_k\rangle$).

$$\sum_{k=1}^{T}\langle\boldsymbol{\theta}^* - \boldsymbol{\theta}_k, \mathbf{m}_k\rangle \leq \frac{(1-\beta_2)\,dD_\infty^2\sqrt{T}G_\infty}{\eta\varepsilon_3}\sqrt{\frac{d\beta_3}{(1-\beta_3)(\beta_3-\beta_2^2)}} + \lambda\sqrt{T}D^2G_\infty$$

$$+ \eta\sqrt{Thd}D_\infty G_\infty + \lambda\eta(1+\log T)D\sqrt{h}D_\infty G_\infty$$

*Proof.* We focus on the $i^{\text{th}}$ dimension of the parameter vector $\boldsymbol{\theta}_k \in R^d$. From the update rules presented in Algorithm 2,

$$\boldsymbol{\theta}_{k+1,i} = \boldsymbol{\theta}_{k,i} - \eta_k\mathbf{r}_{k,i}\frac{\phi\left(\left\|\boldsymbol{\theta}_k^{(i)}\right\|\right)}{\left\|\mathbf{r}_k^{(i)}\right\| + \varepsilon_2\phi\left(\left\|\boldsymbol{\theta}_k^{(i)}\right\|\right)}$$

$$= \boldsymbol{\theta}_{k,i} - \eta_k\left(\frac{\mathbf{m}_{k,i}}{\sqrt{\mathbf{v}_{k,i}}+\varepsilon_1} + \lambda_k\boldsymbol{\theta}_{k,i}\right)\frac{\phi\left(\left\|\boldsymbol{\theta}_k^{(i)}\right\|\right)}{\left\|\mathbf{r}_k^{(i)}\right\| + \varepsilon_2\phi\left(\left\|\boldsymbol{\theta}_k^{(i)}\right\|\right)}$$

Subtract the scalar $\boldsymbol{\theta}_{,i}^*$ and square both sides of the above update rule, we have,

$$\left(\boldsymbol{\theta}_{k+1,i} - \boldsymbol{\theta}_{,i}^*\right)^2 = \left(\boldsymbol{\theta}_{k,i} - \boldsymbol{\theta}_{,i}^*\right)^2 + \eta_k^2\left(\frac{\mathbf{r}_{k,i}\phi\left(\left\|\boldsymbol{\theta}_k^{(i)}\right\|\right)}{\left\|\mathbf{r}_k^{(i)}\right\| + \varepsilon_2\phi\left(\left\|\boldsymbol{\theta}_k^{(i)}\right\|\right)}\right)^2$$

$$- 2\left(\boldsymbol{\theta}_{k,i} - \boldsymbol{\theta}_{,i}^*\right)\eta_k\left(\frac{\mathbf{m}_{k,i}}{\sqrt{\mathbf{v}_{k,i}}+\varepsilon_1} + \lambda_k\boldsymbol{\theta}_{k,i}\right)\frac{\phi\left(\left\|\boldsymbol{\theta}_k^{(i)}\right\|\right)}{\left\|\mathbf{r}_k^{(i)}\right\| + \varepsilon_2\phi\left(\left\|\boldsymbol{\theta}_k^{(i)}\right\|\right)}$$

Rearrange the above equation, we have

$$
\begin{aligned}
\mathbf{m}_{k,i}\left(\boldsymbol{\theta}_{k,i}-\boldsymbol{\theta}_{,i}^{*}\right) =& \frac{\sqrt{\mathbf{v}_{k,i}}+\varepsilon_{1}}{2\eta_{k}}\frac{\left\|\mathbf{r}_{k}^{(i)}\right\|+\varepsilon_{2}\phi\left(\left\|\boldsymbol{\theta}_{k}^{(i)}\right\|\right)}{\phi\left(\left\|\boldsymbol{\theta}_{k}^{(i)}\right\|\right)}\left(\left(\boldsymbol{\theta}_{k,i}-\boldsymbol{\theta}_{,k}^{*}\right)^{2}-\left(\boldsymbol{\theta}_{k+1,i}-\boldsymbol{\theta}_{,i}^{*}\right)^{2}\right)\\
&+\lambda_{k}\left(\sqrt{\mathbf{v}_{k,i}}+\varepsilon_{1}\right)\left(\boldsymbol{\theta}_{,i}^{*}-\boldsymbol{\theta}_{k,i}\right)\boldsymbol{\theta}_{k,i}\\
&+\eta_{k}\frac{\sqrt{\mathbf{v}_{k,i}}+\varepsilon_{1}}{2}\mathbf{r}_{k,i}\frac{\mathbf{r}_{k,i}\phi\left(\left\|\boldsymbol{\theta}_{k}^{(i)}\right\|\right)}{\left\|\mathbf{r}_{k}^{(i)}\right\|+\varepsilon_{2}\phi\left(\left\|\boldsymbol{\theta}_{k}^{(i)}\right\|\right)}\\
=&\frac{\sqrt{\mathbf{v}_{k,i}}+\varepsilon_{1}}{2\eta_{k}}\frac{\left\|\mathbf{r}_{k}^{(i)}\right\|+\varepsilon_{2}\phi\left(\left\|\boldsymbol{\theta}_{k}^{(i)}\right\|\right)}{\phi\left(\left\|\boldsymbol{\theta}_{k}^{(i)}\right\|\right)}\left(\left(\boldsymbol{\theta}_{k,i}-\boldsymbol{\theta}_{,k}^{*}\right)^{2}-\left(\boldsymbol{\theta}_{k+1,i}-\boldsymbol{\theta}_{,i}^{*}\right)^{2}\right)\\
&+\lambda_{k}\left(\sqrt{\mathbf{v}_{k,i}}+\varepsilon_{1}\right)\left(\boldsymbol{\theta}_{,i}^{*}-\boldsymbol{\theta}_{k,i}\right)\boldsymbol{\theta}_{k,i}\\
&+\frac{\eta_{k}}{2}\left(\mathbf{m}_{k,i}+\lambda_{k}\tilde{\boldsymbol{\theta}}_{k,i}\right)\frac{\mathbf{r}_{k,i}\phi\left(\left\|\boldsymbol{\theta}_{k}^{(i)}\right\|\right)}{\left\|\mathbf{r}_{k}^{(i)}\right\|+\varepsilon_{2}\phi\left(\left\|\boldsymbol{\theta}_{k}^{(i)}\right\|\right)}
\end{aligned}
$$

Obtain the regret upper bound via summing the above equation across $k \in [T]$ and all the dimensions for $i \in [d]$, and then using Lemma 9, we have the following inequality:

$$
\begin{aligned}
\sum_{k=1}^{T}\langle\boldsymbol{\theta}^{*}-\boldsymbol{\theta}_{k},\mathbf{m}_{k}\rangle \leq& \sum_{i=1}^{d}\frac{G_{\infty}}{2\eta}\frac{\left\|r_{1}^{(i)}\right\|+\varepsilon_{2}\phi\left(\left\|\boldsymbol{\theta}_{1}^{(i)}\right\|\right)}{\phi\left(\left\|\boldsymbol{\theta}_{1}^{(i)}\right\|\right)}\left(\boldsymbol{\theta}_{1,i}-\boldsymbol{\theta}_{,i}^{*}\right)^{2}\\
&+\sum_{i=1}^{d}\sum_{k=2}^{T}\frac{\left(\boldsymbol{\theta}_{k,i}-\boldsymbol{\theta}_{,i}^{*}\right)^{2}}{2}\left(\frac{\left(\sqrt{\mathbf{v}_{k,i}}+\varepsilon_{1}\right)\left(\left\|\mathbf{r}_{k}^{(i)}\right\|+\varepsilon_{2}\phi\left(\left\|\boldsymbol{\theta}_{k}^{(i)}\right\|\right)\right)}{\eta_{k}\phi\left(\left\|\boldsymbol{\theta}_{k}^{(i)}\right\|\right)}-\frac{\left(\sqrt{\widehat{v}_{k-1,i}}+\varepsilon_{1}\right)\left(\left\|\mathbf{r}_{k-1}^{(i)}\right\|+\varepsilon_{2}\phi\left(\left\|\boldsymbol{\theta}_{k-1}^{(i)}\right\|\right)\right)}{\eta_{k-1}\phi\left(\left\|\boldsymbol{\theta}_{k-1}^{(i)}\right\|\right)}\right)\\
&+\sum_{k=1}^{T}\lambda_{k}\langle\boldsymbol{\theta}^{*}-\boldsymbol{\theta}_{k},\boldsymbol{\theta}_{k}\left(\sqrt{\mathbf{v}_{k}}+\varepsilon_{1}\right)\rangle+\sum_{k=1}^{T}\left(\frac{\eta_{k}G_{\infty}\sqrt{hd}D_{\infty}}{2}+\frac{\lambda_{k}\eta_{k}D\sqrt{h}G_{\infty}D_{\infty}}{2}\right)
\end{aligned}
$$

Using telescoping sum, Cauchy-Schwarz inequality, with some negative end terms omitted, considering Lemma 10, we have:

$$
\begin{aligned}
\sum_{k=1}^{T}\langle\boldsymbol{\theta}^{*}-\boldsymbol{\theta}_{k},\mathbf{m}_{k}\rangle \leq& \frac{(1-\beta_{2})D^{2}G_{\infty}}{\eta\varepsilon_{3}}\sqrt{\frac{d\beta_{3}}{(1-\beta_{3})(\beta_{3}-\beta_{2}^{2})}}+\frac{(1-\beta_{2})dD_{\infty}^{2}\sqrt{T}G_{\infty}}{\eta\varepsilon_{3}}\sqrt{\frac{d\beta_{3}}{(1-\beta_{3})(\beta_{3}-\beta_{2}^{2})}}\\
&+\lambda\sqrt{T}D^{2}G_{\infty}+\eta\sqrt{T}\sqrt{hd}D_{\infty}G_{\infty}+\lambda\eta\left(1+\log T\right)D\sqrt{h}D_{\infty}G_{\infty}\\
\sim& \frac{(1-\beta_{2})dD_{\infty}^{2}\sqrt{T}G_{\infty}}{\eta\varepsilon_{3}}\sqrt{\frac{d\beta_{3}}{(1-\beta_{3})(\beta_{3}-\beta_{2}^{2})}}\\
&+\lambda\sqrt{T}D^{2}G_{\infty}+\eta\sqrt{Thd}D_{\infty}G_{\infty}+\lambda\eta\left(1+\log T\right)D\sqrt{h}D_{\infty}G_{\infty}
\end{aligned}
$$

$\square$

**Lemma 12** (Bound for $\sum_{k=1}^{T}\langle\boldsymbol{\theta}_{k-1}-\boldsymbol{\theta}_{k},\mathbf{m}_{k-1}\rangle$)**.**

$$
\sum_{k=1}^{T}\langle\boldsymbol{\theta}_{k-1}-\boldsymbol{\theta}_{k},\mathbf{m}_{k-1}\rangle \leq \eta\sqrt{Thd}D_{\infty}G_{\infty}
$$

*Proof.* Using Lemma 9, we have

$$\sum_{k=1}^{T} \langle \boldsymbol{\theta}_{k-1} - \boldsymbol{\theta}_k, \mathbf{m}_{k-1} \rangle \leq \sum_{k=1}^{T} \left\langle \frac{\eta_{k-1}\mathbf{r}_{k-1}\phi\left(\left\|\boldsymbol{\theta}_{k-1}^{(\cdot)}\right\|\right)}{\left\|\mathbf{r}_{k-1}^{(\cdot)}\right\| + \varepsilon_2\phi\left(\left\|\boldsymbol{\theta}_{k-1}^{(\cdot)}\right\|\right)}, \mathbf{m}_{k-1} \right\rangle$$

$$\leq \sum_{k=1}^{T} \eta_{k-1} \left\| \frac{\mathbf{r}_{k-1}\phi\left(\left\|\boldsymbol{\theta}_{k-1}^{(\cdot)}\right\|\right)}{\left\|\mathbf{r}_{k-1}^{(\cdot)}\right\| + \varepsilon_2\phi\left(\left\|\boldsymbol{\theta}_{k-1}^{(\cdot)}\right\|\right)} \right\| \|\mathbf{m}_{k-1}\|$$

$$\leq \eta\sqrt{Thd}D_\infty G_\infty$$

$\square$

**Theorem 2.** *Suppose Assumptions 3.3 to 3.6 hold. If $\frac{\beta_2^2}{\beta_3} < 1$, $\eta_k = \frac{\eta}{\sqrt{k}}$, $\alpha_k \leq \frac{\alpha}{\sqrt{k}}$, $\lambda_k \leq \frac{\lambda}{\sqrt{k}}$, Algorithm 7 achieves the following guarantee, for all $T \geq 1$.*

$$R(T) \leq \frac{dD_\infty^2\sqrt{T}G_\infty}{\eta\varepsilon_3}\sqrt{\frac{d\beta_3}{(1-\beta_3)(\beta_3-\beta_2^2)}} + \alpha\sqrt{T}DG_\infty\sqrt{d} + \frac{\lambda\sqrt{T}D^2G_\infty}{(1-\beta_2)}$$

$$+ \frac{\eta\sqrt{Tdh}G_\infty D_\infty}{(1-\beta_2)} + \frac{\lambda\eta(1+\log T)D\sqrt{h}D_\infty G_\infty}{(1-\beta_2)}$$

*Proof.* Using Assumption 3.6, we have,

$$\ell_k(\boldsymbol{\theta}_k) - \ell_k(\theta^*) \leq \mathbf{g}_k^T(\boldsymbol{\theta}_k - \theta^*) = \sum_{i=1}^{d} \mathbf{g}_{k,i}(\boldsymbol{\theta}_{k,i} - \boldsymbol{\theta}_{,i}^*)$$

We focus on the $i^{\text{th}}$ dimension of the parameter vector $\boldsymbol{\theta}_k \in \mathbb{R}^d$. Based on the update rules in Algorithm 7, we have

$$\boldsymbol{\theta}_{k+1,i} = \boldsymbol{\theta}_{k,i} - \eta_k\mathbf{r}_{k,i}\frac{\phi\left(\left\|\boldsymbol{\theta}_k^{(i)}\right\|\right)}{\left\|\mathbf{r}_k^{(i)}\right\| + \varepsilon_2\phi\left(\left\|\boldsymbol{\theta}_k^{(i)}\right\|\right)}$$

$$= \boldsymbol{\theta}_{k,i} - \eta_k\left(\frac{\beta_2}{\sqrt{\mathbf{v}_{k,i}} + \varepsilon_1}\mathbf{m}_{k-1,i} + \frac{(1-\beta_2)}{\sqrt{\mathbf{v}_{k,i}} + \varepsilon_1}(\mathbf{g}_{k,i} + \alpha_k\mathbf{a}_{k,i}) + \lambda_k\boldsymbol{\theta}_{k,i}\right)\frac{\phi\left(\left\|\boldsymbol{\theta}_k^{(i)}\right\|\right)}{\left\|\mathbf{r}_k^{(i)}\right\| + \varepsilon_2\phi\left(\left\|\boldsymbol{\theta}_k^{(i)}\right\|\right)}.$$

Subtract the scalar $\boldsymbol{\theta}_{,i}^*$ and square both sides of the above update rule, we have,

$$\left(\boldsymbol{\theta}_{k+1,i} - \boldsymbol{\theta}_{,i}^*\right)^2 = \left(\boldsymbol{\theta}_{k,i} - \boldsymbol{\theta}_{,i}^*\right)^2 + \eta_k^2\left(\frac{\mathbf{r}_{k,i}\phi\left(\left\|\boldsymbol{\theta}_k^{(i)}\right\|\right)}{\left\|\mathbf{r}_k^{(i)}\right\| + \varepsilon_2\phi\left(\left\|\boldsymbol{\theta}_k^{(i)}\right\|\right)}\right)^2$$

$$- 2\left(\boldsymbol{\theta}_{k,i} - \boldsymbol{\theta}_{,i}^*\right)\eta_k\left(\frac{\beta_2}{\sqrt{\mathbf{v}_{k,i}} + \varepsilon_1}\mathbf{m}_{k-1,i} + \frac{(1-\beta_2)}{\sqrt{\mathbf{v}_{k,i}} + \varepsilon_1}(\mathbf{g}_{k,i} + \alpha_k\mathbf{a}_{k,i}) + \lambda_k\boldsymbol{\theta}_{k,i}\right)\frac{\phi\left(\left\|\boldsymbol{\theta}_k^{(i)}\right\|\right)}{\left\|\mathbf{r}_k^{(i)}\right\| + \varepsilon_2\phi\left(\left\|\boldsymbol{\theta}_k^{(i)}\right\|\right)}$$

Rearrange the above equation and use Young's inequality, $ab \leq a^2/2 + b^2/2$. Then

$$\mathbf{g}_{k,i}\left(\boldsymbol{\theta}_{k,i}-\boldsymbol{\theta}_{,i}^{*}\right)=\frac{\sqrt{\mathbf{v}_{k,i}}+\varepsilon_{1}}{2\eta_{k}\left(1-\beta_{2}\right)}\frac{\left\|\mathbf{r}_{k}^{(i)}\right\|+\varepsilon_{2}\phi\left(\left\|\boldsymbol{\theta}_{k}^{(i)}\right\|\right)}{\phi\left(\left\|\boldsymbol{\theta}_{k}^{(i)}\right\|\right)}\left(\left(\boldsymbol{\theta}_{k,i}-\boldsymbol{\theta}_{,k}^{*}\right)^{2}-\left(\boldsymbol{\theta}_{k+1,i}-\boldsymbol{\theta}_{,i}^{*}\right)^{2}\right)$$

$$+\frac{\beta_{2}}{\left(1-\beta_{2}\right)}\left(\boldsymbol{\theta}_{,i}^{*}-\boldsymbol{\theta}_{k,i}\right)\mathbf{m}_{k-1,i}+\left(\boldsymbol{\theta}_{,i}^{*}-\boldsymbol{\theta}_{k,i}\right)\alpha_{k}\mathbf{a}_{k,i}+\lambda_{k}\frac{\sqrt{\mathbf{v}_{k,i}}+\varepsilon_{1}}{\left(1-\beta_{2}\right)}\left(\boldsymbol{\theta}_{,i}^{*}-\boldsymbol{\theta}_{k,i}\right)\boldsymbol{\theta}_{k,i}$$

$$+\eta_{k}\frac{\sqrt{\mathbf{v}_{k,i}}+\varepsilon_{1}}{2\left(1-\beta_{2}\right)}\mathbf{r}_{k,i}\frac{\mathbf{r}_{k,i}\phi\left(\left\|\boldsymbol{\theta}_{k}^{(i)}\right\|\right)}{\left\|\mathbf{r}_{k}^{(i)}\right\|+\varepsilon_{2}\phi\left(\left\|\boldsymbol{\theta}_{k}^{(i)}\right\|\right)}$$

$$=\frac{\sqrt{\mathbf{v}_{k,i}}+\varepsilon_{1}}{2\eta_{k}\left(1-\beta_{2}\right)}\frac{\left\|\mathbf{r}_{k}^{(i)}\right\|+\varepsilon_{2}\phi\left(\left\|\boldsymbol{\theta}_{k}^{(i)}\right\|\right)}{\phi\left(\left\|\boldsymbol{\theta}_{k}^{(i)}\right\|\right)}\left(\left(\boldsymbol{\theta}_{k,i}-\boldsymbol{\theta}_{,k}^{*}\right)^{2}-\left(\boldsymbol{\theta}_{k+1,i}-\boldsymbol{\theta}_{,i}^{*}\right)^{2}\right)$$

$$+\frac{\beta_{2}}{\left(1-\beta_{2}\right)}\left(\boldsymbol{\theta}_{,i}^{*}-\boldsymbol{\theta}_{k,i}\right)\mathbf{m}_{k-1,i}+\left(\boldsymbol{\theta}_{,i}^{*}-\boldsymbol{\theta}_{k,i}\right)\alpha_{k}\mathbf{a}_{k,i}+\lambda_{k}\frac{\sqrt{\mathbf{v}_{k,i}}+\varepsilon_{1}}{\left(1-\beta_{2}\right)}\left(\boldsymbol{\theta}_{,i}^{*}-\boldsymbol{\theta}_{k,i}\right)\boldsymbol{\theta}_{k,i}$$

$$+\frac{\eta_{k}}{2\left(1-\beta_{2}\right)}\left(\mathbf{m}_{k,i}+\lambda_{k}\tilde{\theta}_{k,i}\right)\frac{\mathbf{r}_{k,i}\phi\left(\left\|\boldsymbol{\theta}_{k}^{(i)}\right\|\right)}{\left\|\mathbf{r}_{k}^{(i)}\right\|+\varepsilon_{2}\phi\left(\left\|\boldsymbol{\theta}_{k}^{(i)}\right\|\right)}$$

Obtaining the regret upper bound via summing the above equation across $k\in 1\ldots T$ and all the dimensions for $i\in 1,\ldots,d$ and considering Lemma 9, we have the following inequality:

$$R(T)\leq\sum_{i=1}^{d}\frac{G_{\infty}}{2\eta\left(1-\beta_{2}\right)}\frac{\left\|\mathbf{r}_{1}^{(i)}\right\|+\varepsilon_{2}\phi\left(\left\|\boldsymbol{\theta}_{1}^{(i)}\right\|\right)}{\phi\left(\left\|\boldsymbol{\theta}_{1}^{(i)}\right\|\right)}\left(\boldsymbol{\theta}_{1,i}-\boldsymbol{\theta}_{,i}^{*}\right)^{2}$$

$$+\sum_{i=1}^{d}\sum_{k=2}^{T}\frac{1}{2\left(1-\beta_{2}\right)}\left(\boldsymbol{\theta}_{t,i}-\boldsymbol{\theta}_{,i}^{*}\right)^{2}$$

$$\left(\frac{\left(\sqrt{\mathbf{v}_{k,i}}+\varepsilon_{1}\right)\left(\left\|\mathbf{r}_{k}^{(i)}\right\|+\varepsilon_{2}\phi\left(\left\|\boldsymbol{\theta}_{k}^{(i)}\right\|\right)\right)}{\eta_{k}\phi\left(\left\|\boldsymbol{\theta}_{k}^{(i)}\right\|\right)}-\frac{\left(\sqrt{\mathbf{v}_{k-1,i}}+\varepsilon_{1}\right)\left(\left\|\mathbf{r}_{k-1}^{(i)}\right\|+\varepsilon_{2}\phi\left(\left\|\boldsymbol{\theta}_{k-1}^{(i)}\right\|\right)\right)}{\eta_{k-1}\phi\left(\left\|\boldsymbol{\theta}_{k-1}^{(i)}\right\|\right)}\right)$$

$$+\sum_{k=1}^{T}\left(\frac{\beta_{2}}{\left(1-\beta_{2}\right)}\left\langle\boldsymbol{\theta}^{*}-\boldsymbol{\theta}_{k},\mathbf{m}_{k-1}\right\rangle+\alpha_{k}\left\langle\boldsymbol{\theta}^{*}-\boldsymbol{\theta}_{k},a_{k}\right\rangle+\frac{\lambda_{k}}{\left(1-\beta_{2}\right)}\left\langle\boldsymbol{\theta}^{*}-\boldsymbol{\theta}_{k},\boldsymbol{\theta}_{k}\left(\sqrt{\mathbf{v}_{k}}+\varepsilon_{1}\right)\right\rangle\right)$$

$$+\sum_{k=1}^{T}\left(\frac{\eta_{k}G_{\infty}\sqrt{h}dD_{\infty}}{2\left(1-\beta_{2}\right)}+\frac{\lambda_{k}\eta_{k}D\sqrt{h}G_{\infty}D_{\infty}}{2\left(1-\beta_{2}\right)}\right)$$

Using telescoping sum, Cauchy-Schwarz inequality, with some negative end terms omitted and considering Lemmas 10 to 12 we have:

$$R(T) \leq \frac{D^2 G_\infty}{\eta \varepsilon_3} \sqrt{\frac{d\beta_3}{(1-\beta_3)(\beta_3 - \beta_2^2)}} + \frac{dD_\infty^2 \sqrt{T} G_\infty}{\eta \varepsilon_3} \sqrt{\frac{d\beta_3}{(1-\beta_3)(\beta_3 - \beta_2^2)}}$$

$$+ \frac{\beta_2}{(1-\beta_2)} \left( \sum_{k=1}^{T} \langle \boldsymbol{\theta}^* - \boldsymbol{\theta}_{k-1}, \mathbf{m}_{k-1} \rangle + \sum_{k=1}^{T} \langle \boldsymbol{\theta}_{k-1} - \boldsymbol{\theta}_k, \mathbf{m}_{k-1} \rangle \right) + \alpha \sqrt{T} D G_\infty \sqrt{d} + \frac{\lambda \sqrt{T} D^2 G_\infty}{(1-\beta_2)}$$

$$+ \frac{\eta \sqrt{Tdh} G_\infty D_\infty}{(1-\beta_2)} + \frac{\lambda \eta (1 + \log T) D \sqrt{h} D_\infty G_\infty}{(1-\beta_2)}$$

$$\sim \frac{dD_\infty^2 \sqrt{T} G_\infty}{\eta \varepsilon_3} \sqrt{\frac{d\beta_3}{(1-\beta_3)(\beta_3 - \beta_2^2)}} + \alpha \sqrt{T} D G_\infty \sqrt{d} + \frac{\lambda \sqrt{T} D^2 G_\infty}{(1-\beta_2)}$$

$$+ \frac{\eta \sqrt{Tdh} G_\infty D_\infty}{(1-\beta_2)} + \frac{\lambda \eta (1 + \log T) D \sqrt{h} D_\infty G_\infty}{(1-\beta_2)}$$

$\square$

# D   Experimental Details

## D.1   Supplementary Experiments

Table 8: Test top-1 Acc. (%) on ResNet20 with CIFAR10 and CIFAR100 and on ResNet34 with ImageNet (for hyperparameters and architecture details, see Section D.2)

| Dataset | CIFAR10 | CIFAR100 | ImageNet |
|---------|---------|----------|----------|
| batch size | 128 | 128 | 256 |
| SGD | **91.85** | 64.93 | **70.64** |
| ESGD | 91.83 | **65.01** | 70.59 |
| Adam | 89.88 | 64.35 | 65.06 |
| EAdam | **90.12** | **65.24** | **65.31** |
| Lamb | 90.89 | 61.29 | 69.17 |
| ALTO | **91.24** | **65.74** | **69.95** |

**Different Optimization Test Functions**

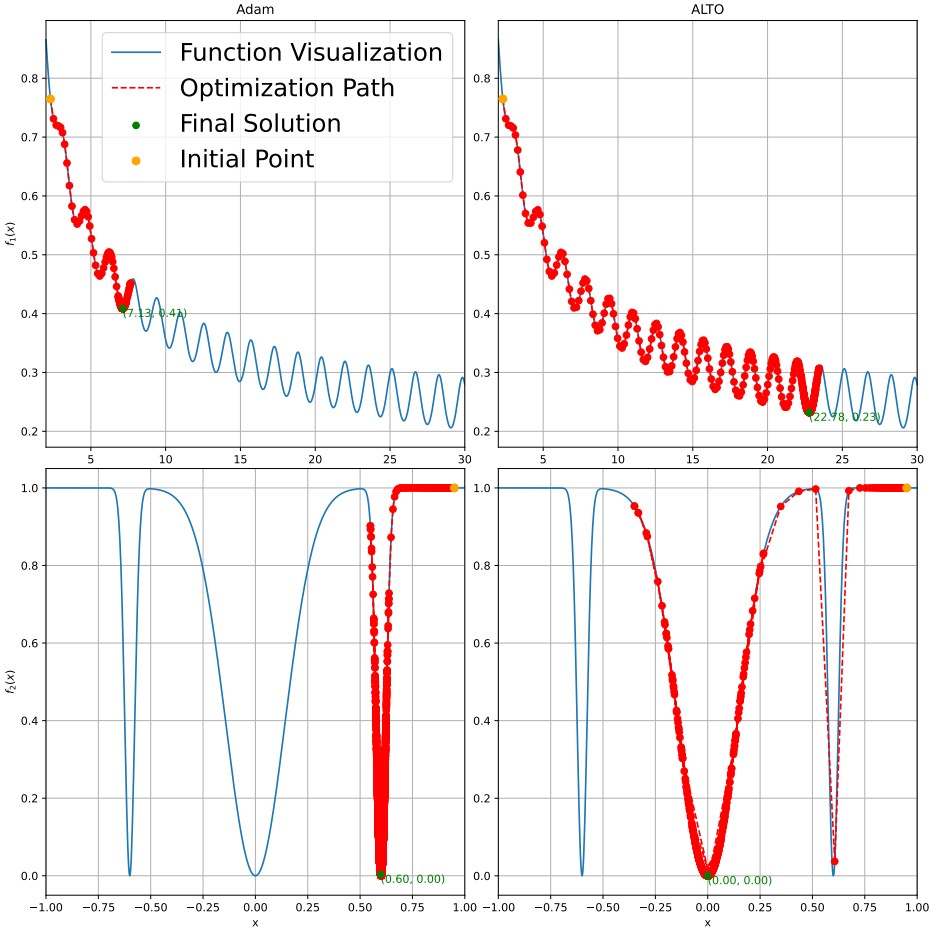

Figure 8: Optimization process of Adam ($\beta_1 = 0.95$) and ALTO ($\beta_1 = 0.99, \beta_2 = 0.95, \alpha = -80$) on functions $f_1(x) = 1/(0.05x sin^2(2x) + ln(x+1)), x_0 = 2.3$ and Adam ($\beta_1 = 0.9999$) and ALTO ($\beta_1 = 0.99, \beta_2 = 0.9, \alpha = -90$) on $f_2(x) = 1 - e^{-900(x-0.6)^2} - e^{-900(x+0.6)^2} - e^{-25x^2}, x_0 = 0.95$.

**Performances of Different Optimizers on BERT$_{base}$**

Because most of the parameters of the model have been pre-trained to a relative accurate extent, the advantage of ALTO would be limited but still stable. Even so, it still demonstrates broad adaptability, at least on the datasets CONLL 2003, IMDB, and MRPC . The smaller the dataset is (IMDB (25K), CoNLL2003 (Ĩ4K), and MRPC (3668)) or the larger the batch size is (32, 1024, 4096), the bigger the advantage of ALTO over other optimizers is. This is maily because ALTO is designed for large-batch training.

Table 9: Test result (%) of fine-tuning BERT$_{base}$ model on the CONLL2003, IMDB and MRPC tasks.

| BtSz | Optimizer | CoNLL2003 (~14K) | | | IMDB (25K) | | | | MRPC (3668) | | | | Avg |
|---|---|---|---|---|---|---|---|---|---|---|---|---|---|
| | | F1 | Prec. | Recall | Acc. | F1 | Prec. | Recall | Acc. | F1 | Prec. | Recall | F1 and Acc |
| 32 | Adam | 86.97 | 85.70 | 87.15 | 91.77 | 91.52 | **94.43** | 88.78 | 84.46 | 88.68 | 85.99 | 91.54 | 88.68 |
| | AdamW | 89.32 | 88.41 | 90.25 | 92.42 | 92.43 | 92.24 | 92.62 | 84.63 | 88.70 | **86.75** | 90.75 | 89.50 |
| | AdaBelief | 89.98 | 89.01 | 90.98 | 92.93 | 92.85 | 93.01 | 91.86 | 83.36 | 87.87 | 85.24 | 90.67 | 89.39 |
| | Lamb | 89.86 | 88.82 | 90.93 | 93.13 | 93.09 | 93.56 | **92.63** | 84.40 | 88.59 | 86.22 | 91.10 | 89.81 |
| | ALTO | **90.29** | **89.53** | **91.06** | **93.24** | **93.19** | 93.82 | 92.57 | **84.86** | **89.01** | 86.07 | **92.15** | **90.11** |
| 1024 | Adam | 89.50 | 88.09 | 90.96 | 92.97 | 92.91 | 93.79 | 92.04 | 79.76 | 85.05 | 83.58 | 86.57 | 88.03 |
| | AdamW | 89.46 | 88.17 | 90.78 | 92.98 | 92.91 | 93.92 | 91.92 | 80.86 | 86.32 | 82.24 | 90.84 | 88.50 |
| | AdaBelief | 90.38 | 89.66 | 91.12 | 92.82 | 92.69 | **94.46** | 90.98 | 79.53 | 84.61 | **84.58** | 84.65 | 88.00 |
| | Lamb | 90.05 | 89.37 | 90.75 | 92.07 | 92.30 | 89.74 | **95.01** | 80.98 | 86.04 | 84.03 | 88.14 | 88.28 |
| | ALTO | **90.59** | **89.95** | **91.24** | **93.10** | **93.17** | 92.19 | 94.17 | **81.39** | **86.77** | 82.26 | **91.80** | **89.00** |

**Performance Comparison with Related Works.**

Table 10: Test top-1 Acc. (%) on ResNet20 with CIFAR10 and CIFAR100 and on ResNet34 with ImageNet (for hyperparameters and architecture details, see Section D.2)

| Dataset | CIFAR10 | | CIFAR100 | | ImageNet | |
|---|---|---|---|---|---|---|
| batch size | 128 | 16384 | 128 | 16384 | 256 | 4086 |
| Adan | 90.62 | 76.42 | 62.27 | 47.96 | 69.36 | 67.66 |
| ALTO | **91.24** | **88.83** | **65.74** | **57.78** | **69.95** | **70.83** |

Table 11: Comparative performance of LION and ALTO in training the GPT-2 (345M parameters) Model with 4096 batch size on Megatron-LM Framework [40] by the OpenAI's open-sourced GPT-2 Output Dataset (1M).

| Optimizer | Train Loss | Test PPL |
|---|---|---|
| LION | 4.66 | 110.54 |
| ALTO | **4.33** | **78.37** |

Table 12: Comparison of average elapsed time per iteration for ALTO and LION. LION computes 31.2% faster than ALTO at each iteration. When the global batch size divided by the micro batch size equals data parallelism, it indicates that each GPU performs only one forward pass and one backward pass per iteration.

| Optimizer | Time (ms) |
|---|---|
| LION | **253** |
| ALTO | 332 |

**VGG and DenseNet for CIFARs**

In addition to ResNet, we expanded our experiments to include other network architectures such as VGG-16 and DenseNet, particularly focusing on training with large batch sizes on CIFAR10 and CIFAR100 datasets. The network structures for VGG-16 and DenseNet are outlined in Tables 13 and 14:

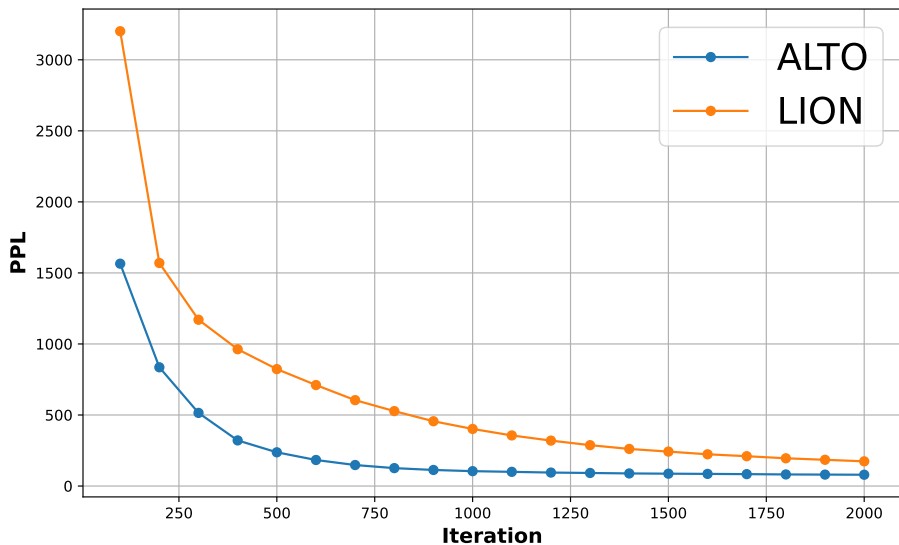

Figure 9: PPL Variation over First 2000 Iterations on 345M GPT with GPT-2-Output-Dataset. In this experiment, the number of iterations required for train_PPL to converge to 200 was 66% fewer for ALTO compared to LION.

VGG-16 comprises approximately 33.73 million parameters

DenseNet-169 comprises approximately 7.27 million parameters.

Table 13: Architecture of VGG-16

| Layer Type | Configuration |
|---|---|
| Convolutional | 2x[96], 2x[128], 3x[256], 3x[512], 3x[512] |
| Pooling | 5 MaxPool (2x2, stride 2) |
| Fully Connected | 3 FC (4096, 4096, num_classes) |
| Batch Normalization | After each Convolution |
| Activation | ReLU |
| Dropout | 0.3 after first two FC layers |

Table 14: Architecture of DenseNet-169

| Layer Type | Configuration |
|---|---|
| Basic Convolution | Initial Conv Layer |
| Dense Blocks | 6, 12, 32, 32 Bottlenecks per block |
| Transition Layers | Between Dense Blocks |
| Fully Connected | FC (num_planes, 256, 128, num_classes) |
| Growth Rate | 32 |
| Reduction | 0.5 after each Dense Block |
| Batch Normalization | After each Convolution in Bottleneck |
| Activation | ReLU |
| Dropout | 0.4 after FC layers |

In our experiments, we employed various neural network architectures distinct from those described in the main text, utilizing prominent optimizers such as SGD, Adam, AdamW, AdaBelief, and Lamb for comparison against our optimization algorithm, ALTO2. These experiments were conducted

on CIFAR10 and CIFAR100 datasets. The results obtained using VGG-16 and DenseNet-169 are presented in Table 15.

Table 15: Acc. (%) of VGG and DenseNet on CIFAR10 and CIFAR100 for batch size 16384

| Dataset | CIFAR10 | | CIFAR100 | |
| --- | --- | --- | --- | --- |
| Architecture | VGG-16 | DenseNet-169 | VGG-16 | DenseNet-169 |
| SGD | 86.56 | 65.44 | 55.46 | 41.75 |
| Adam | 85.01 | 81.89 | 40.44 | 48.41 |
| AdamW | 83.33 | 81.55 | 37.62 | 48.58 |
| Lamb | 90.63 | 85.77 | 60.44 | 53.75 |
| AdaBelief | 87.87 | 76.88 | 59.44 | 43.41 |
| ALTO | **90.92** | **86.19** | **63.15** | **54.24** |

During the training process, the acc-epoch and loss-acc relationships are depicted in Figures 10 and 11, respectively. Our optimization algorithm, in these extended experiments, was employed with a training parameter of batch size = 16,384. This approach was taken to explore its optimization effectiveness across various network architectures and to ensure that our optimization algorithm possesses robust model generalizability.

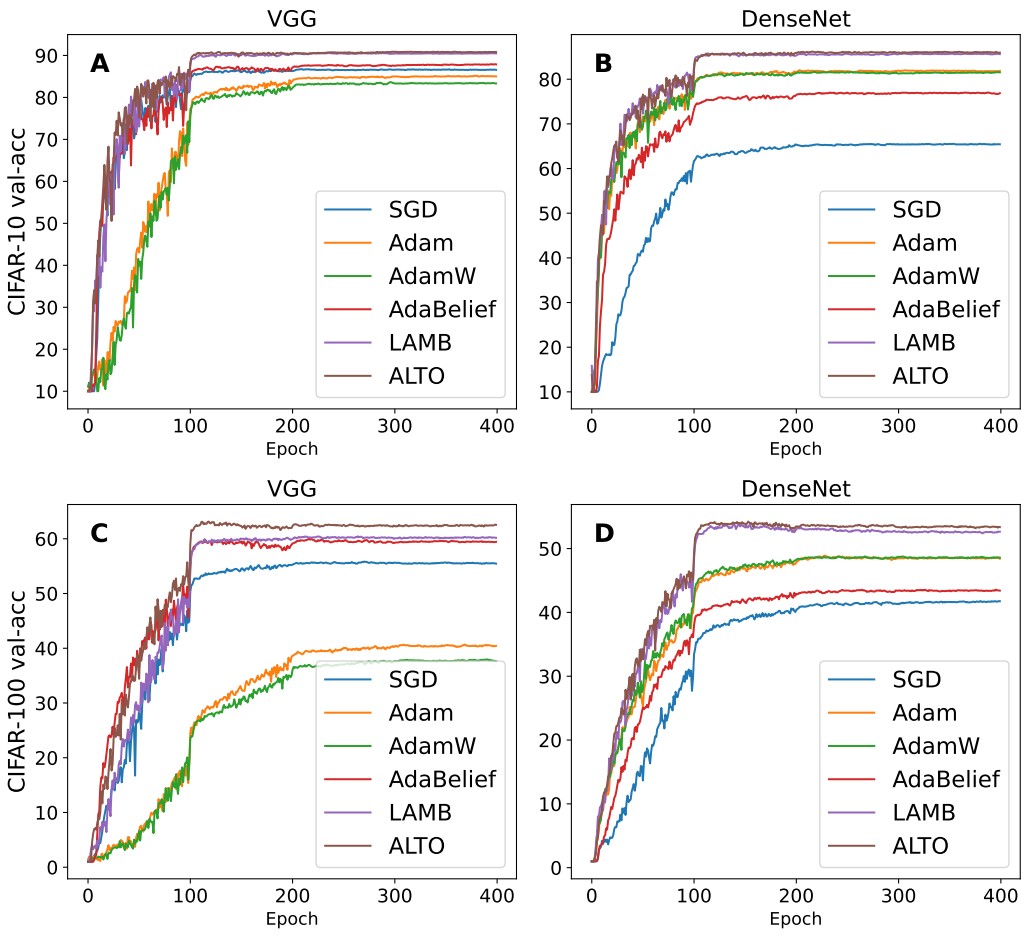

Figure 10: test accuracy with batch size 16384

**Comparative Illustration of Training with Batches from Small to Large**

We conducted a comparison of training the VGG-16 convolutional neural network using the ALTO and Lamb optimizers on the CIFAR-100 dataset. Our comparison involved batch sizes ranging from

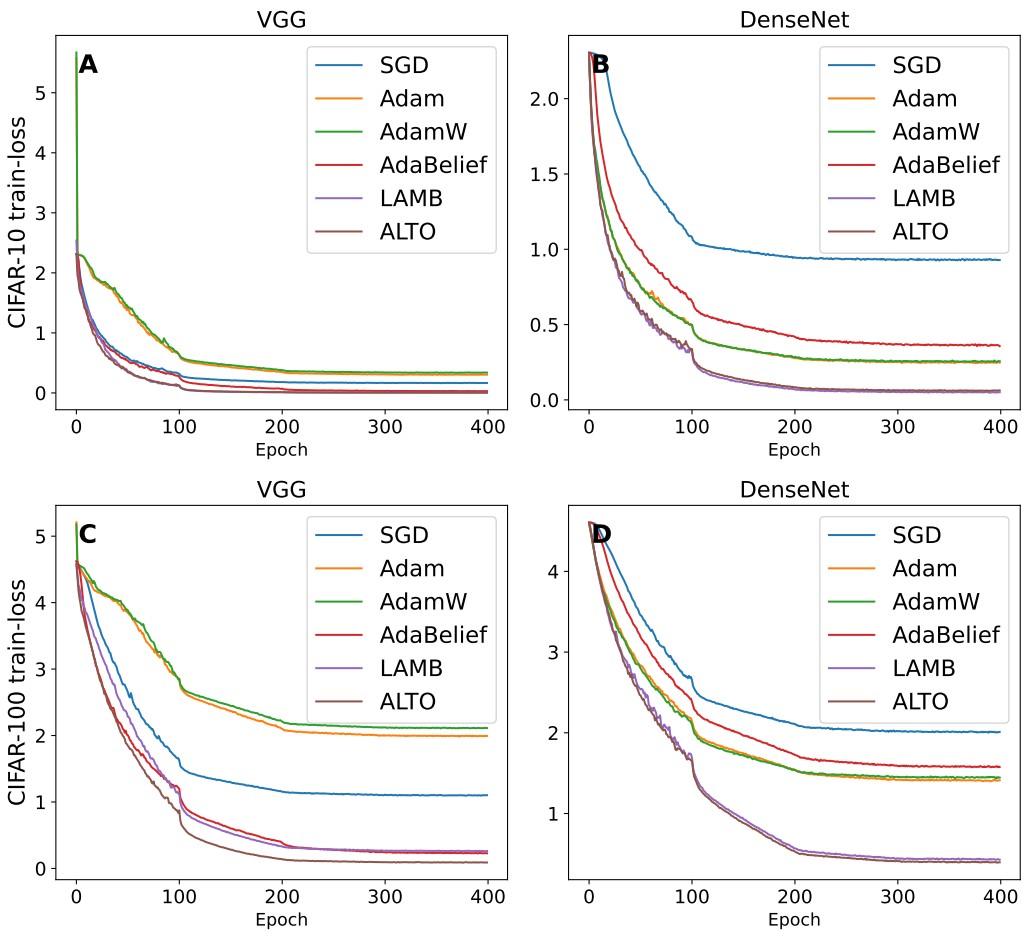

Figure 11: training loss with batch size 16384

200 to 50,000 (the total number of training samples in the CIFAR-100 dataset). The results of the experiment are presented in the following Table 16 and Figures 12 and 13.

In this experiment, to more objectively demonstrate the effectiveness of the ALTO optimizer, both $\alpha$ and $\beta$ were held constant across all batch sizes during training, and all other hyperparameters were set to their default values. This experiment provides a clear illustration of the performance of the ALTO optimizer.

### The Relationship between Epoch and Training Speed, and Comparative Experiments

To explore the relationship between batch size and training speed, we conducted experiments with the ALTO training setup for VGG-16 on CIFAR-100. The batch size ranged from 200 to the total number of training samples (50K). We calculated the time required for each epoch during stable training. For details, see Table 17.

'Computation Time' represents the total time spent on computational tasks during the training of a single epoch. It encompasses the duration of all the calculations performed by the algorithm. In contrast, 'Total Time' not only accounts for the 'Computation Time' but also includes the time overhead caused by the data transfer between the CPU and GPU, as well as the time required for memory allocation (malloc) operations. The latter is subject to the influence of the machine's capabilities and can vary significantly with different hardware configurations. With ongoing advancements in the field of high-performance computing, the time costs associated with these hardware-dependent processes are expected to diminish. Therefore, the 'Computation Time' is of greater interest from an algorithmic

Table 16: Comparison of ALTO and Lamb Optimizers Across Different Batches with Corresponding Learning Rates, Beta Values, Accuracy, and Loss

| Batch Size | Learning Rate | $\beta_1$ | ALTO | | Lamb | |
|---|---|---|---|---|---|---|
| | | | Accuracy | Loss | Accuracy | Loss |
| 200 | 0.001 | 0.999 | 66.9 | 0.0034 | 67.3 | 0.0025 |
| 500 | 0.001 | 0.999 | 63.35 | 0.0029 | 63.3 | 0.0034 |
| 1K | 0.001 | 0.999 | 59.79 | 0.0046 | 59.82 | 0.0057 |
| 2K | 0.01 | 0.999 | 70.43 | 0.0021 | 70.24 | 0.002 |
| 5K | 0.01 | 0.999 | 67.55 | 0.0008 | 67.05 | 0.0009 |
| 10K | 0.01 | 0.999 | 64.51 | 0.0023 | 62.92 | 0.0046 |
| 15K | 0.01 | 0.999 | 64.38 | 0.0065 | 62.64 | 0.0132 |
| 25K | 0.01 | 0.999 | 59.19 | 0.2003 | 57.79 | 0.4173 |
| 30K | 0.01 | 0.999 | 58.69 | 0.2859 | 58.16 | 0.3718 |
| 40K | 0.01 | 0.999 | 58.72 | 0.3535 | 57.11 | 0.5281 |
| 50K | 0.01 | 0.999 | 49.69 | 1.4763 | 39.91 | 2.0295 |

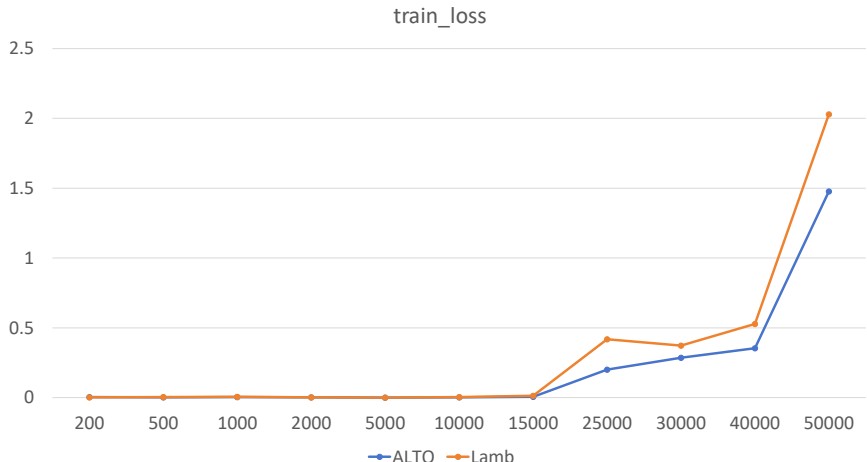

Figure 12: loss of training with batch size 200 to 50K of ALTO and Lamb

Table 17: The relationship between batch size and training speed, showing total time(s) and pure computation time for 1 epoch.

| Batch Size | Total Time | Computation Time |
|---|---|---|
| 200 | 14.264 | 12.327 |
| 500 | 6.533 | 5.330 |
| 1000 | 4.147 | 3.034 |
| 2000 | 3.290 | 1.917 |
| 5000 | 3.488 | 1.233 |
| 10000 | 4.563 | 1.026 |
| 25000 | 7.542 | 0.891 |
| 50000 | 13.100 | 0.847 |

perspective, as it more accurately reflects the efficiency of the algorithm itself, independent of the underlying hardware performance

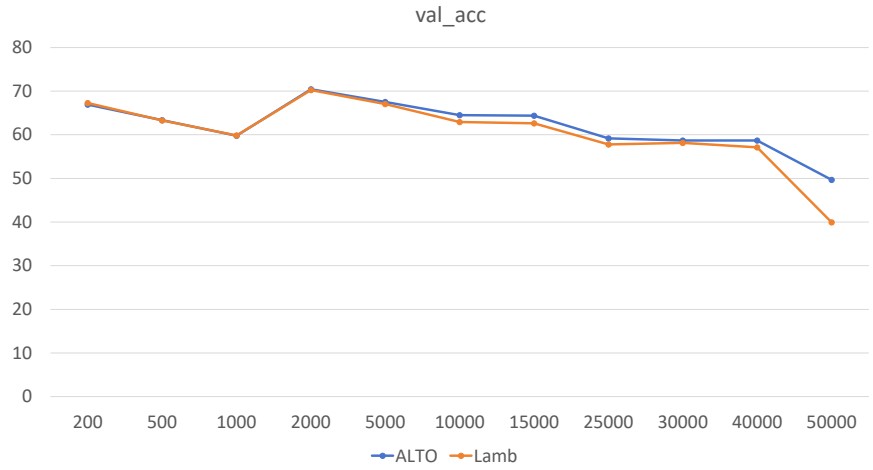

Figure 13: Acc(%) of training with batch size 200 to 50K of ALTO and Lamb

We conducted comparative experiments on VGG-16 using the CIFAR-100 dataset to compare the training time required for ALTO and Lamb to reach the same level of accuracy. In these experiments of batch size 16384, both algorithms maintained a consistent learning rate throughout the training process. For ALTO, the hyperparameters were set with a fixed $\beta_1$ of 0.999 and $\alpha$ of -5. For details, see Table 18.

Table 18: Comparison of the time it takes for ALTO and Lamb to achieve the same accuracy

| ACC(%) | ALTO Total Time (s) | ALTO Computation Time (s) | Lamb Total Time (s) | Lamb Computation Time (s) |
|---|---|---|---|---|
| 20 | 137.103 | 5.348 | 195.992 | 7.341 |
| 30 | 202.674 | 7.905 | 276.694 | 10.364 |
| 40 | 333.817 | 13.020 | 409.277 | 15.330 |
| 50 | 482.842 | 18.833 | 582.210 | 21.807 |
| 60 | 608.023 | 23.715 | 864.669 | 32.387 |
| 70 | - | - | - | - |

**Reinforcement Learning**

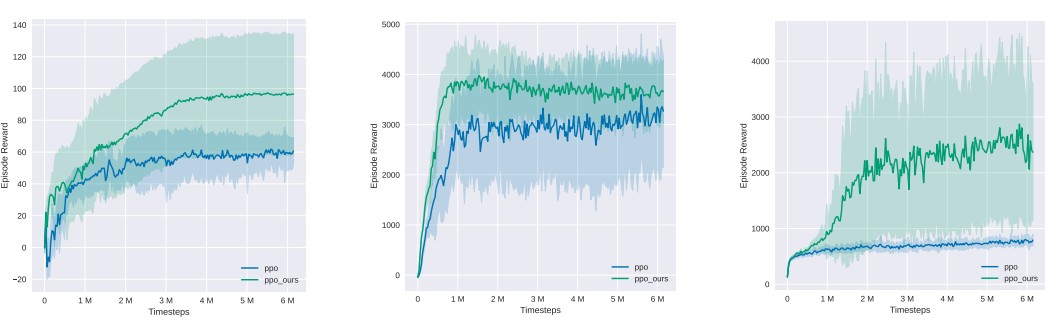

Figure 14: Swimmer          Figure 15: Ant          Figure 16: Humanoid

Specifically, we tested three sets of reinforcement learning games in three environments simulated by the MuJoCo engine: Swimmer-v3, Ant-v3, and Humanoid-v3. We conducted our experiments using the Proximal Policy Optimization (PPO) algorithm [39], which is widely used in the field of

reinforcement learning. The experiments were based on the widely-used open-source codebase Tianshou [47]. In our comparative experiments, we replaced the default optimizer of the framework with the ALTO optimizer, while keeping other hyperparameters as the default settings of the framework, such as lr=3e-4, epoch=200, and each set of experiments was conducted using ten random seeds.

In the Swimmer-v3 environment, there is a simple organism with two to three segments, moving in a two-dimensional space. Here, the challenge is to learn to coordinate its body segments to propel forward in water. In the Ant-v3 environment, the agent is a four-legged creature similar to an ant. The objective is to control this ant to move as quickly as possible in a three-dimensional space. The Humanoid-v3 environment is one of the most complex settings within the MuJoCo suite. It involves a bipedal humanoid robot that needs to learn walking, running, and maintaining balance. The challenge here lies in the high dimensionality of the action space and the requirement for intricate balance and coordination.

The training tasks in our three reinforcement learning experiments vary in complexity. The results show that our optimization algorithm is highly versatile.The results of three comparative experiments are shown in Figures 14 to 16.

**Full-Batch Training for LSTM**

We also investigated the impact of large-scale training of ALTO on traditional network architectures. Specifically, for the Named Entity Recognition (NER) task on the CONLL2003 dataset, we conducted full batch training on a two-layer BiLSTM network (with a batch size equal to the size of the training dataset, which is 14,987 samples). A BiLSTM network is an extension of the traditional LSTM model, which processes data in both forward and backward directions, providing a richer context for each sequence element. This bi-directional approach allows the model to capture dependencies from both past and future states, enhancing its ability to recognize and classify entities in text sequences. In this task, we continued to identify the optimal learning rate setting for each optimization algorithm. Additionally, a uniform learning rate adjustment strategy was employed, where the learning rate was reduced to one-tenth of its original value after every 100 training epochs, with a total of 400 training epochs.Our BiLSTM network structure is shown in Table 19.

Table 19: Architecture of BiLSTM Network for NER

| Layer Type | Configuration |
| --- | --- |
| Embedding Layer | Dimension: 100 |
| BiLSTM Layer | Hidden Dimension: 128, |
|  | Num Layers: 2, |
|  | Bidirectional: True, |
|  | Dropout: 0.5 |
| Dropout | Dropout: 0.5 |
| Fully Connected Layer | Output Dimension: 9 |

Table 20: Comparison of Different Optimizers for Full Batch Training on BiLSTM

| Optimizer | Train Loss | Test Acc | Test F1 |
| --- | --- | --- | --- |
| Adam | 0.0663 | 15.00 | 9.66 |
| AdamW | 0.0374 | 27.39 | 15.89 |
| AdaBelief | 0.0476 | 25.68 | 13.98 |
| Lamb | 0.0164 | 61.19 | 52.20 |
| ALTO | 0.0139 | **62.36** | **57.27** |

The performance of all optimizers in this task is shown in Table 20. All optimizers were tested across learning rates [0.005, 0.01, 0.05], with each optimization algorithm selecting the best performing learning rate. The F1 scores corresponding to each learning rate for all optimizers are shown in Table 21.

In this full-batch experiment targeting the CONLL2003 dataset, the hyperparameter $\beta_1$ for ALTO is set to 0.7, with all other parameters remaining at their default values.

Table 21: F1 Scores for Different Optimizers at Various Learning Rates

| Optimizer | LR = 0.005 | LR = 0.01 | LR = 0.05 |
|---|---|---|---|
| Adam | 0 | **9.66** | 0.12 |
| AdamW | 8.24 | **15.89** | 4.69 |
| AdaBelief | 0 | 0.24 | **13.98** |
| Lamb | 10.35 | 20.27 | **52.20** |
| ALTO | - | 24.43 | **57.27** |

## D.2 Achitectures and Hyperparameters

**CV Projects**

In our CIFAR training tasks, we employed the ResNet20 architecture as part of our experimental setup. This section provides a detailed overview of the network structure and the number of parameters involved in the model.

The ResNet20 model used in our experiments is a variant of the ResNet architecture, specifically designed for the CIFAR10 and CIFAR100 dataset. The model structure is as follows:

Table 22: Architecture of ResNet20

| Layer Type | Output Size | Details |
|---|---|---|
| Convolution + BN | 32x32 | 3x3 conv, 16, stride 1, padding 1 |
| BasicBlock x3 | 32x32 | [3x3 conv, 16] x 2 each |
| BasicBlock x3 | 16x16 | [3x3 conv, 32, stride 2] + [3x3 conv, 32] x 2 each |
| BasicBlock x3 | 8x8 | [3x3 conv, 64, stride 2] + [3x3 conv, 64] x 2 each |
| Global Avg Pooling | 1x1 | Avg pool |
| Fully Connected | number of classes | number of classes-way softmax |

The ResNet20 model for CIFAR10 and CIFAR100, as implemented in our experiments, comprises approximately 0.27 million parameters. This count includes parameters from all convolutional layers, batch normalization layers, and the final fully connected layer, which is shown in Table 22.

In the experiments conducted on the CIFAR10 and CIFAR100 datasets, the default hyperparameters of the optimizer were used. When the batch size was set to 16,384, $\beta_1$ was fixed at 0.99, whereas for a batch size of 256, $\beta_1$ was set to 0.1. For all compared optimizers at a batch size of 256, the learning rate was adjusted within the set {0.01, 0.05, 0.1} to select the best model, with a total of 200 training epochs. Conversely, at a batch size of 16,384, the learning rate was fine-tuned between 0.01 and 0.1 to determine the optimal model, over a course of 400 training epochs. The optimal learning rates for each optimizer on CIFAR-10 and CIFAR-100 are shown in Tables 23 and 24, respectively.In all experiments of CIFAR10 and CIFAR100, the learning rate was reduced by a factor of ten after every quarter epoch of training: $\eta = \eta * 0.1$ when $T\%(\frac{epoch}{4}) = 0$.

Table 23: Learning Rates for CIFAR10 training

| Batch Size | 128 | 16384 |
|---|---|---|
| SGD | 0.05 | 0.1 |
| others | 0.01 | 0.05 |

In the ablation study section, which focuses on the CIFAR100 dataset, we experimented with adjusting optimizer parameters for different batch sizes using the ResNet18 network architecture. The details of this network architecture are presented in Table 25.

In our experiments targeting ImageNet, we employed the ResNet34 as the fitting model. The specific network structure is detailed in Tables 26 and 27.

Table 24: Learning Rates for CIFAR100 training

| Batch Size | 128 | 16384 |
|---|---|---|
| SGD | 0.05 | 0.1 |
| others | 0.01 | 0.01 |

Table 25: Architecture of ResNet18 for CIFAR100 in Ablation Study

| Layer Type | Output Size | Details |
|---|---|---|
| Convolution + BN | 32x32 | 3x3 conv, 64, stride 1, padding 1 |
| BasicBlock x2 | 32x32 | [3x3 conv, 64] x 2 each |
| BasicBlock x2 | 16x16 | [3x3 conv, 128, stride 2] + [3x3 conv, 128] x 2 each |
| BasicBlock x2 | 8x8 | [3x3 conv, 256, stride 2] + [3x3 conv, 256] x 2 each |
| BasicBlock x2 | 4x4 | [3x3 conv, 512, stride 2] + [3x3 conv, 512] x 2 each |
| Global Avg Pooling | 1x1 | Avg pool |
| Fully Connected | 100 | 100-way softmax |

Table 26: Architecture of ResNet34 for ImageNet

| Layer Type | Output Size | Details |
|---|---|---|
| Convolution + BN | 112x112 | 7x7 conv, 64, stride 2, padding 3 |
| Max Pooling | 56x56 | 3x3 max pool, stride 2, padding 1 |
| BasicBlock x3 | 56x56 | [3x3 conv, 64] x 2 each |
| BasicBlock x4 | 28x28 | [3x3 conv, 128, stride 2] + [3x3 conv, 128] x 2 each |
| BasicBlock x6 | 14x14 | [3x3 conv, 256, stride 2] + [3x3 conv, 256] x 2 each |
| BasicBlock x3 | 7x7 | [3x3 conv, 512, stride 2] + [3x3 conv, 512] x 2 each |
| Global Avg Pooling | 1x1 | Avg pool |
| Fully Connected | number of classes | Linear layer with 1000 outputs (for ImageNet) |

Table 27: Architecture of ResNet-50 for ImageNet

| Layer Type | Output Size | Details |
|---|---|---|
| Convolution + BN | 112x112 | 7x7 conv, 64, stride 2, padding 3 |
| Max Pooling | 56x56 | 3x3 max pool, stride 2, padding 1 |
| Bottleneck x3 | 56x56 | [1x1 conv, 64] + [3x3 conv, 64] + [1x1 conv, 256] x 3 |
| Bottleneck x4 | 28x28 | [1x1 conv, 128] + [3x3 conv, 128] + [1x1 conv, 512] x 4 |
| Bottleneck x6 | 14x14 | [1x1 conv, 256] + [3x3 conv, 256] + [1x1 conv, 1024] x 6 |
| Bottleneck x3 | 7x7 | [1x1 conv, 512] + [3x3 conv, 512] + [1x1 conv, 2048] x 3 |
| Global Avg Pooling | 1x1 | Avg pool |
| Fully Connected | number of classes | Linear layer with 1000 outputs (for ImageNet) |

In these experiments, we trained models using two different batch sizes, with corresponding adjustments in learning rate and optimizer settings. In all our training sessions for ImageNet, the range of learning rates was set to {1e-3, 5e-3, 1e-2, 1e-1}.

For a batch size of 256, we selected the best model based on performance over different learning rates. The learning rate scheduler is T=90 and $\eta = \eta * 0.1$ when epoch%30=0. For the ALTO optimizer specifically, we set $\beta_1$ to 0.01, keeping other hyperparameters at their default values.

For a larger batch size of 4096, we also selected the optimal model over different learning rates. The number of training epochs and the learning rate reduction schedule remained the same as for the batch size 256 training works. In the case of the ALTO optimizer for this larger batch size, $\beta_1$ was set to 0.99, while other parameters were maintained at their default settings. For each optimizer, the learning rate and accuracy corresponding to the best-performing model we ultimately selected are shown in Table 28. This approach ensured consistency in training duration and adjustment of the

learning rate across different scales of batch sizes while tailoring the optimizer settings to specific batch size requirements

Table 28: Learning Rates (LR) and Top-1 and Top-5 Accuracy (%) of ResNet34 on ImageNet-1k

| Batch Size | 256 | | 4096 | | LR (256) | LR (4096) |
|---|---|---|---|---|---|---|
| | Top-1 Acc. | Top-5 Acc. | Top-1 Acc. | Top-5 Acc. | | |
| SGD | **70.64** | **89.68** | 49.35 | 74.41 | 1e-2 | 1e-2 |
| Adam | 65.06 | 86.47 | 54.96 | 79.07 | 1e-3 | 1e-2 |
| AdamW | 69.64 | 88.90 | 68.40 | 88.07 | 1e-3 | 1e-2 |
| AdaBelief | 70.12 | 89.24 | 70.18 | 89.26 | 1e-3 | 1e-2 |
| Lamb | 69.17 | 88.81 | 70.34 | 89.55 | 5e-3 | 1e-2 |
| ALTO | 69.95 | 88.94 | **70.83** | **89.64** | 1e-3 | 1e-2 |

In the experiment with ResNet-50 on ImageNet-100, where batch sizes ranged from 1K to 32K, the learning rate scheduling method incorporated a combination of warmup and polynomial decay to optimize the training process. The random seed was set to 42, ensuring reproducibility across different runs. The specific learning rates and warmup epochs for each batch size are detailed in Table 29. This approach aids in stabilizing the training in its early phases by gradually increasing the learning rate from a lower initial value during the warmup period, followed by a polynomial decay to finely tune the model as it converges to optimal solutions.

Table 29: Learning Rate and Warmup Epochs for Different Batch Sizes

| Batch Size | 1K | 2K | 4K | 8K | 16K | 32K |
|---|---|---|---|---|---|---|
| Learning Rate | $\frac{4}{2^{2.5} \times 100}$ | $\frac{4}{2^{2.0} \times 100}$ | $\frac{4}{2^{1.5} \times 100}$ | $\frac{4}{2^{1.0} \times 100}$ | $\frac{4}{2^{0.5} \times 100}$ | $\frac{4}{2^{0.0} \times 100}$ |
| Warmup Epochs | 0.625 | 1.25 | 2.5 | 5 | 10 | 20 |

**NLP Projects**

In the experiments focusing on natural language processing, we utilized the BERT-base model for fine-tuning on downstream tasks. This approach allowed us to evaluate the performance of various optimizers under different batch size settings. The BERT-base model, known for its efficacy in a range of NLP tasks, comprises a specific network architecture. The detailed structure of BERT-base, including its layers and configurations, is presented in Table 30. At the same time, the network structure for the pre-training of the GPT architecture is shown in Table 31.

Table 30: BERT-base Network Structure

| Layer Type | Description |
|---|---|
| Embedding Layers | Token, Segment, Positional Embeddings Size: 768 |
| Transformer Blocks | 12 Layers Hidden size: 768 Feed-forward/filter size: 3072 Attention heads: 12 |
| Output Layer | Linear Layer with Softmax Function |

In our experiments on the CONLL2003 Named Entity Recognition (NER) task, each set of experiments was fine-tuned over 5 epochs. For training with batch size of 1024, the learning rate of AdaBelief, ALTO and Lamb were set to 1e-3; for Adam and AdamW, the learning rate was set to 5e-5 because for Adam, AdamW, and AdaBelief, increasing the learning rate similarly results in poorer convergence performance in this task. We meticulously recorded the F1-score on the validation set after each epoch. The model yielding the highest F1-score on the validation set was subsequently used for calculating metrics on the test set. Table 32 shows the performance of each optimizer with different learning rate in experiments with a batch size of 1024.

Table 31: GPT Model Architecture

| Component | Specification |
|---|---|
| Layers | 24 Transformer blocks |
| Hidden Size | 1024 |
| Attention Heads | 16 |
| Sequence Length | 1024 |
| Max Position Embeddings | 1024 |

Table 32: F1-score of Experiments Using Various Optimizers with different learning rate for CONLL2003

| Optimizer | 1e-3 | 5e-4 | 5e-5 |
|---|---|---|---|
| Adam | 86.62 | 89.45 | 89.50 |
| AdamW | 83.51 | 89.29 | 89.46 |
| AdaBelief | 90.38 | 90.14 | 59.20 |
| Lamb | 90.05 | 88.74 | - |
| ALTO | 90.59 | 88.77 | - |

Accuracy was not utilized as a metric in this task due to its limited effectiveness in scenarios with imbalanced data. Accuracy may yield misleading results by overemphasizing the majority class while neglecting the model's performance on the minority class. Hence, more informative metrics like F1-score, precision, and recall were employed to provide a balanced evaluation of the model's performance across different classes.

For the IMDB task with batch size of 1024, the learning rate was selected in {5e-4, 5e-5, 1e-5}. Each set of experiments was fine-tuned over 3 epochs. The accuracy with different learning rate was shown in Table 33. In these experiments, we also shown the training loss and training accuracy at the end of each epoch are presented in the table below.

Table 33: Accuracy of Experiments Using Various Optimizers with different learning rate for IMDB

| Optimizer | 5e-4 | 5e-5 | 1e-5 |
|---|---|---|---|
| Adam | 50.00 | 92.97 | 92.43 |
| AdamW | 50.00 | 92.98 | 91.79 |
| AdaBelief | 91.75 | 92.82 | 91.75 |
| Lamb | 92.19 | 92.07 | - |
| ALTO | 93.10 | 92.19 | - |

In the natural language processing experiments, the batch size had the most significant impact on the MRPC task, primarily because of its small training dataset size, consisting of only 3,668 samples. For this task, the hyperparameters were set as follows: a batch size of 1024 and a $\beta_1$ value of 0.999. The learning rate was selected in {5e-4, 5e-5, 1e-5}. The performance effects of different learning rates corresponding to various optimizers are presented in Table 34. Each experiment was fine-tuned over the course of 10 epochs.

Table 34: Accuracy of Experiments Using Various Optimizers with different learning rate for MRPC

| Optimizer | 5e-4 | 5e-5 | 1e-5 |
|---|---|---|---|
| Adam | 68.75 | 79.76 | 72.69 |
| AdamW | 66.49 | 80.86 | 66.49 |
| AdaBelief | 79.53 | 70.08 | 66.49 |
| Lamb | 80.98 | 70.60 | - |
| ALTO | 81.39 | 69.56 | - |

In our pre-training setup for the GPT model, we employed a deep network architecture consisting of 24 Transformer layers, with each layer configured to have a hidden size of 1024 and 16 attention

heads. This configuration supports the processing of sequences up to 1024 tokens in length, allowing the model to capture long-range dependencies within the data. The training utilized a micro-batch size of 4, with an effective global batch size of 4096. This large-scale training was facilitated by MPI and NCCL to optimize multi-GPU communication. Furthermore, gradient clipping was applied at a threshold of 1.0 to prevent exploding gradients, a common issue in training such deep networks. Additionally, mixed precision training was leveraged to enhance training speed and reduce memory consumption without compromising model accuracy.Our pre-training experiments utilized the dataset from OpenAI's open-source dataset: the gpt-2-output-dataset, which includes a total of 1 million data entries.

All the optimizers compared in our study were experimented with different learning rates to determine the optimal results. The training used a random seed of 1234. The best learning rates for each optimization algorithm are shown in Table 35. See Figure 17 for the PPL-iter graph of the first 1.5k iterations in our 5k iterations of training.

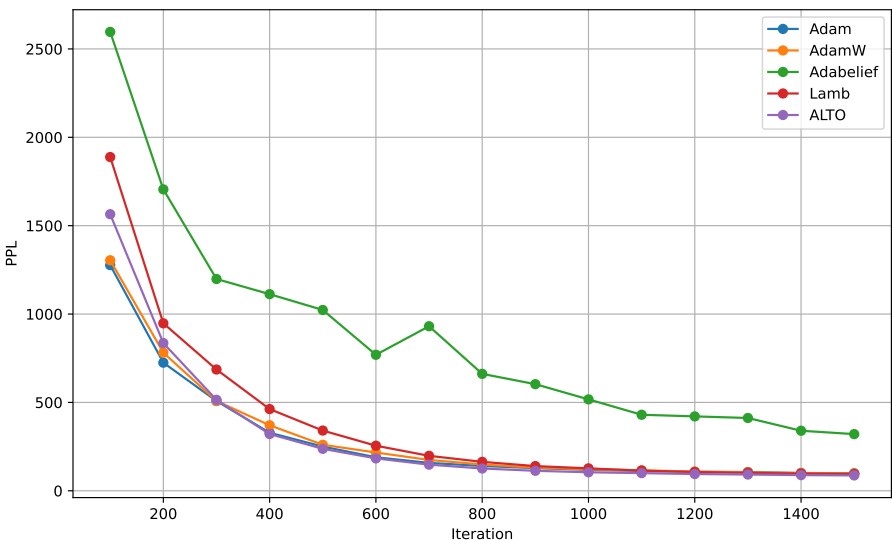

Figure 17: PPL Variation over First 1500 Iterations on 345M GPT with GPT-2-Output-Dataset

Table 35: Best Learning Rates for Various Optimizers on

| Optimizer | Best Learning Rate |
|-----------|--------------------|
| Adam | $6e-4$ |
| AdamW | $8e-4$ |
| AdaBelief | $5e-2$ |
| LAMB | $1e-2$ |
| ALTO | $1e-2$ |

### D.3 Hyperparameter Tuning

In our extended experiments, we present box plots of the training results for ResNet-18 on CIFAR-100 under various hyperparameters, with the training graphs displayed here. In the experiments focusing on $\beta_1$, we observe that the value of $\beta_1$ is generally positively correlated with the size of the batch size. In the comparative experiments for different $\alpha$, due to the closeness of the curves, we extract the graphs of the last 20 epochs. The specific display diagrams are as shown in Figures 18 to 20.

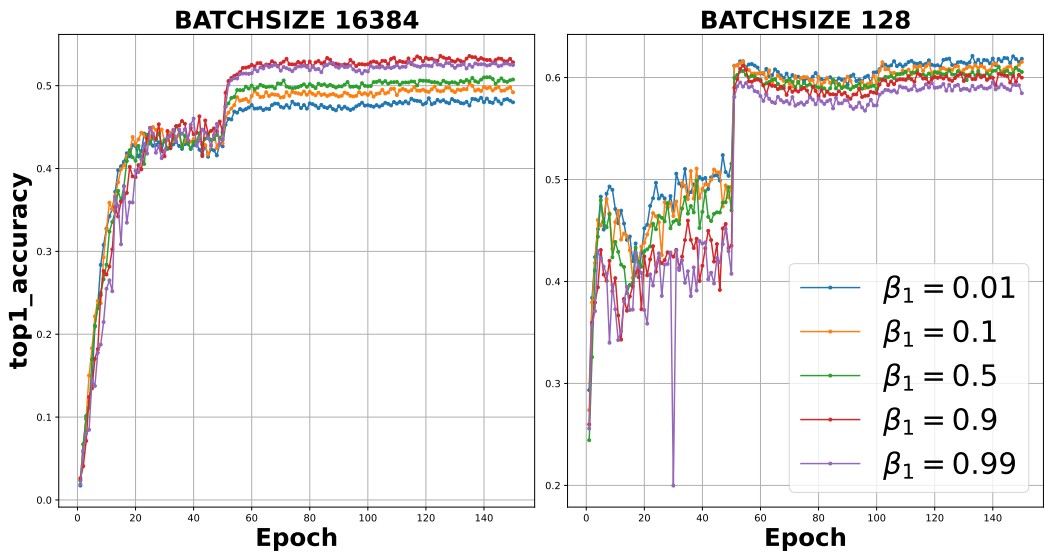

Figure 18: acc-epoch for different $\beta$ on batch size 16384 and 128.

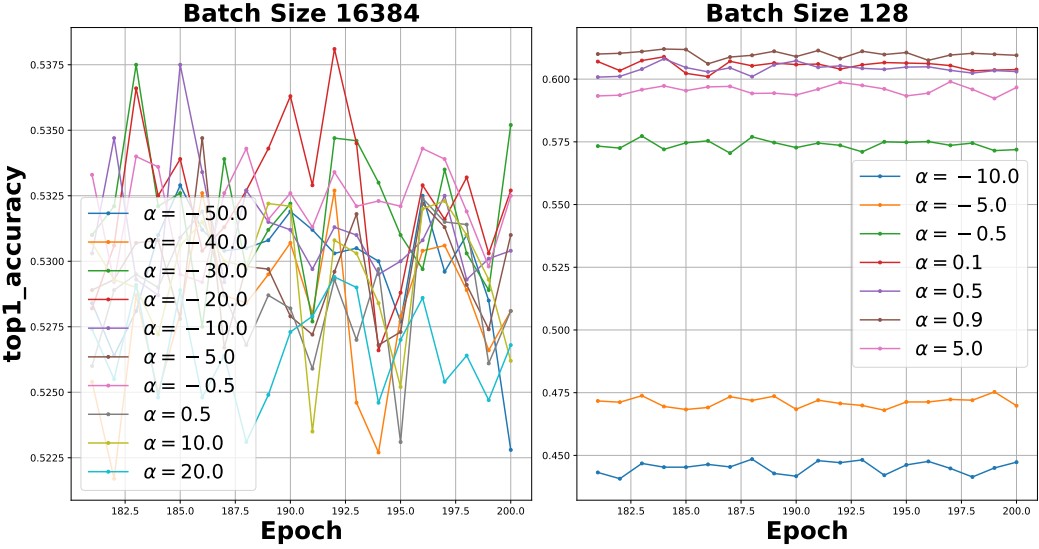

Figure 19: acc-epoch for different $\alpha$ on batch size 16384 and 128.

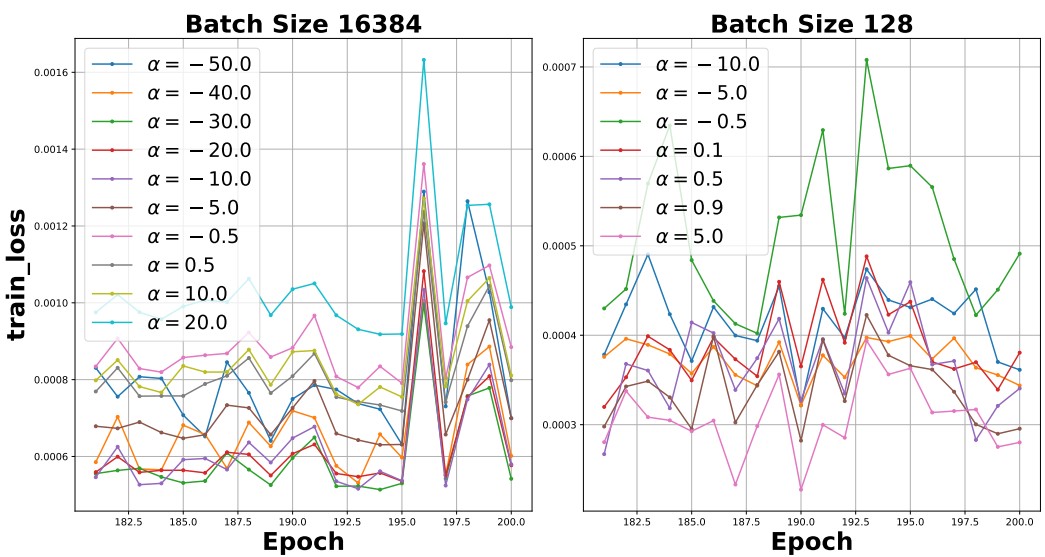

Figure 20: training loss-epoch for different $\alpha$ on batch size 16384 and 128.

