# OpenReview forum: "Exploring Landscapes for Better Minima along Valleys"
_NeurIPS.cc/2025/Conference — NeurIPS 2025 poster_

### Official Review · Reviewer_uGBV · 2025-06-26

**Clarity:** 3
**Significance:** 3
**Originality:** 3
**Rating:** 4
**Confidence:** 3

**Summary:**

The paper aims to develop an algorithm that converges to local minima that have better generalization performance, compared to traditional gradient-based algorithms such as Adam and SGD. Starting from the intuition that flatter minima may have better generalization, the authors propose ALTO – an algorithm that has the usual (stochastic) gradient component in its update for being captured by wide local minima but also a second component that is for escaping narrow local minima. Theoretical analysis of ALTO’s convergence is provided under certain conditions, but there is no proof for the claimed better generalization or flatter minima.

**Questions:**

N/A

**Ethical Concerns:**

["NO or VERY MINOR ethics concerns only"]

**Final Justification:**

[+] The algorithm design is theoretical motivated, and intuively makes sense.

[+] Not much additional computation is needed, although implicitly using second order information.

[+] Good performance in experiments.

[-] No theoretical or experimental evidence of the major claim.

[-] Introduces two hyer-parameters, which introduces additional effort to tune. Rebuttal partially addressed this concern.

**Limitations:**

yes

**Quality:**

2

**Strengths And Weaknesses:**

**Strengths**:

The paper conceptually utilizes second order information of the loss landscape, namely the Hessian matrix, for escaping sharp minima and finding flat ones. Generally, it is computationally expensive to directly use Hessian, this paper smartly avoided such computation by using gradient difference $g_k-g_{k-1}$ as an approximation. Such a design intuitively makes sense.

The proposed method achieves good test performance in various tasks, better than existing algorithms in many such tasks.

**Weaknesses**:

The paper provides no evidence, neither theoretically nor numerically, showing the proposed methods indeed find flat minima, which is the major claim of the paper though. Theoretical analysis of the paper only shows convergence of the algorithm, it would be better if the paper analyzes/discusses where it converges to or whether it converges to flatter minima. It would also enhance the paper if there is a comparison of the sharpness/flatness of the convergence point of different algorithms.

The proposed algorithm introduces two hyper-parameters. On one hand, this makes the algorithm less usable in practice; on the other hand, it is unclear whether the improved test accuracy reported in the paper is because of the flexibility that comes with the additional hyper-parameters or it is from the claimed flatter minima.

The paper claims “accelerated convergence” and “preferential convergence” (Line 44), however, there is no evidence of the claims.

Minor:

Line 116: missing reference

---

> ### Author Rebuttal · Authors · 2025-07-31
>
> Thanks for your carefully reading. I will summarize your questions and answer them one by one.
>
> **Q1: The paper provides no evidence, neither theoretically nor numerically, showing the proposed methods indeed find flat minima, which is the major claim of the paper though.**
>
> "It would also enhance the paper if there is a comparison of the sharpness/flatness of the convergence point of different algorithms" is a good advice. We will add a  such comparison. In fact, our optimizer try to reduce $\\|\nabla f\\|$, which is directly related to the sharpness of a local minimum and Hessian matrix.  SInce the optimization problem is a stochastic optimzation problem, we can not even garantee the final minimum is a global minimum, let alone the flatest one. We only need to let the optimizer incline to flatter minima, so we require the optimizer to reduce $\\|\nabla f\\|$. This is what "preferential convergence" refers to. As for “accelerated convergence”, many expriments show this (Table 17, Figure 14, and etc.).
>
> **Q2: The proposed algorithm introduces two hyper-parameters. On one hand, this makes the algorithm less usable in practice; on the other hand, it is unclear whether the improved test accuracy reported in the paper is because of the flexibility that comes with the additional hyper-parameters or it is from the claimed flatter minima.**
>
> Indeed, we did not fine-tuned the hyperparameter. In small batch training case, the performance is even little worse than baseline. And we said in Line 216, we set $\alpha$ and $\beta_1$ as fixed default value in all our experiments, we do not introduce the flexilibty.
>
> **Q3:The paper claims “accelerated convergence” and “preferential convergence” (Line 44), however, there is no evidence of the claims.**
>
> Please see answer of **Q1**.

---

> > ### Comment · Reviewer_uGBV · 2025-08-06
> >
> > Thanks for the reply. It partially addresses my concerns. Rating score remains.

---

### Official Review · Reviewer_LVE5 · 2025-06-30

**Clarity:** 2
**Significance:** 3
**Originality:** 3
**Rating:** 4
**Confidence:** 3

**Summary:**

This paper introduces ALTO, an adaptation of gradient-based optimizers, aimed at improving convergence to flatter and better-generalizing minima by continuing exploration along low-loss valleys even after encountering a local minimum. The key idea is to combine gradient momentum with an estimated acceleration term (based on gradient differences) to help escape sharp minima and encourage further traversal through flat regions of the loss landscape. The authors provide convergence proofs for both convex and non-convex cases, and extensive experiments show that ALTO improves performance in large-batch training tasks.

**Questions:**

- How does ALTO compare to recent optimizers such as Muon, or second-order optimizers such as Shampoo?
- Given a model and a dataset, is there a principled way to select $\alpha$ and $\beta_1$?

**Ethical Concerns:**

["NO or VERY MINOR ethics concerns only"]

**Final Justification:**

The author's response has addressed my concerns regarding comparison to other optimizers.

**Limitations:**

yes

**Quality:**

3

**Strengths And Weaknesses:**

**Strengths**
- The ALTO method elegantly integrates an EMA-based acceleration term into existing first-order optimizers and is backed by rigorous convergence guarantees in both convex and non-convex settings.
- The proposed optimizer performs well across various tasks in computer vision, natural language processing, and reinforcement learning, which suggests its versatility.
- The authors conduct thorough ablation studies that validate the necessity of each design component, including acceleration in the first- and second-moment estimates, bias correction, and layer-wise scaling.

**Weaknesses**
- While the method excels in large-batch settings, its performance in small-batch training is either comparable or worse than baseline optimizers (e.g., Table 4, where SGD outperforms ALTO on CIFAR-10 with batch size 128). This suggests ALTO may not guarantee good convergence rate unless the batch size is very large.
- There could be more discussion on how the convergence results compared to other optimizers.

---

> ### Author Rebuttal · Authors · 2025-07-31
>
> Thanks for your carefully reading. I will summarize your questions and answer them one by one.
>
> **Q1: How does ALTO compare to recent optimizers such as Muon, or second-order optimizers such as Shampoo?**
>
> We have compared our method with some recent similar optimizers in section 5. Due to the length limitations of Neurips paper, we may not compared our method with some recent works. I think the biggest different between our method and Muon or Shampoo is the computational complexity. Large computational complexity means long training time of an epoch. If the number of parameter is $N$, the computational complexity of our method is $\mathcal{O}(N)$. If the weight matrix is square matrix, the computational complexity of Muon and Shampoo is $\mathcal{O}(N^{\frac{3}{2}})$ at least. As for performance comparison, we will conduct a experiment on this. In fact, we design a optimizer adaptor, not an optimizer, so we can apply this adaptor to Muon and Shampoo.
>
> **Q2: Given a model and a dataset, is there a principled way to select $\alpha$ and $\beta_1$?**
>
> A general but simple principled way is given in section 4.5. Indeed, the $\alpha$ and $\beta_1$ are determinded by the geometry of the landscape of a certain task, with is determinded by models and datasets by part. If the model without skip connection, usually the landscape is relatively rough, we should choose large $\alpha$ and small $\beta_1$. If the dataset is large, usually the landscape is relatively smooth, we should choose small $\alpha$ and large $\beta_1$.

---

> > ### Comment · Reviewer_LVE5 · 2025-08-05
> >
> > The author's response has addressed my concerns. I would like to maintain my positive rating.

---

### Official Review · Reviewer_vVfW · 2025-07-01

**Clarity:** 3
**Significance:** 3
**Originality:** 3
**Rating:** 4
**Confidence:** 3

**Summary:**

This paper introduces a new optimizer method, ALTO which not only can have valley-following behavior, but also be able to escape from small-scale local minima. The author further provides the convergence analysis of algorithm and demonstrate its effectiveness across different experiments.

**Questions:**

Please check the weakness part. My main concern is about format and presentation issue, and limited experimental improvements.
If the author can:
1. largely improve the format and presentation of paper;
2. provide the standard deviation of the experimental results.
3. explain clearly why ALTO mainly works in large batch size while the improvement is still limited. Meanwhile, give some practical suggestions when implementing ALTO.

**Ethical Concerns:**

["NO or VERY MINOR ethics concerns only"]

**Final Justification:**

The method is great and the explanations help to address my concerns.

**Limitations:**

Not applicable.

**Quality:**

2

**Strengths And Weaknesses:**

Strengths:
1. The intuition behind algorithm is well explained.
2. The experimental settings are quite comprehensive.

Weakness:
1. The paper format is quite bad, especially mixing one-column and two-column formats. For example, line 76-88, line 121, line 150-151, etc. These are quite unfriendly for reader to read.
2. For the experimental results. The performance improvement is limited, and only observed in large batch size. However, large batch size is not widely used in practice and from Figure 3, we can see that large batch size suffer worse performance.
3. It will be helpful to report the standard deviation for the experimental results.

---

> ### Author Rebuttal · Authors · 2025-07-30
>
> Thanks for your comments and questions. We will improve the paper format and report the standard deviation for the experimental results.
>
> **Q: Why ALTO mainly works in large batch size while the improvement is still limited?**
>
> In fact, the optimizer adaptor also works for small batch training since it is just an optimizer itself when $\alpha$ degenerates to $0$. Our experiment mainly focus on large batch training and we want to use it to show the ability of our optimizer adaptor, since large batch training is very important and difficult in industry field and our improvement (2.5% averagely) is not very limited compared with similar works in this field. If we pretraing a large model with large dataset, using large batch training can much increase the training efficiengy. Only using small batch to training can not fully utilize GPUs to accelerate the training process (data parallelism) . Large batch training is usually required to use the same number of epoches to training as small batch training, which means the number of parameter iteration in large batch training is much less than that in small batch training. Traditional method is to enlarge the learning rate directly, but the learning rate can not be too large (bounded by the geometry of the landscape of certain task). Otherwise, the training will explode. In order to show the abilibty of the adaptor, we do not spend much time on choosing a hyperparameter. This is a possible reason for worse result than SOTA in small bach training.  We will systematically add more experiments of small batch training in our paper.
>
> **Some suggestions for using our optimizer adaptor (for large batch training, ALTO).**
>
> Usually, the larger the batch size is , the larger the $-\alpha$, $\beta_1$, and learning rate $\eta$ are. $\alpha$ should be postive for small batch training and negative for large batch training. As we said in Line 216, we default to $\alpha=0.5, \beta_1=0.01$ for small batch and $\alpha=-5, \beta_1=0.99$ for large batch in all our experiments (of course, it can be better).  As we show in Figure 4, there are more better results can be achieved with different $\alpha$ and $\beta_1$. We just want to show the method works. Only choosing the best result in experiment may be thought to be cherry-picky.

---

> > ### Comment · Reviewer_vVfW · 2025-08-05
> >
> > Thank you for your reply. Can you help to explain why $\alpha$ should be positive for the small batch while negative for the large batch? Meanwhile, the selection of $\beta_1$ which is either 0.1 or 0.99 makes it hard to choose if I have a moderate batch size. Is there any further suggestions for practical use? and intuition behind such choices.

---

> > > ### Author Response · Authors · 2025-08-06
> > >
> > > Thanks for your questions.
> > >
> > > **A1:** During training process, the graph of $\ell_k$ shakes around the landscape the graph of $f:=\mathbb{E}\ell$. The smaller the batch size is, the more violently it shakes. Large $\alpha$ (positive) stablizes the small batch training process, which forces the optimizer to converge to the local minimum it finds at present (exploitation).  In contrast, large batch training process is stable. Especially, loss of full batch is fixed. Therefore, we should let the optimizer explore more possible local minima (larger parameter space along valley), which corresponds to small $\alpha$ (negative).
> > >
> > > **A2:** We just intuitively set $\beta_1=0.01$ for small batch case  (small than 1k) and $\beta_1=0.99$ for large batch case  (small than 1k) in our experiments and did not finely tune $\beta_1$. The threshold "1k" is not sharp, there would be a better choice. The intuition behind the choice is: large $\beta_1$ means remember more history information of $\mathbf{g}_k-\mathbf{g}_k$. As we stated above, the graph of $\ell_k$ in small batch case shakes violently, which means more noise in history information. Therefore, we should let the optimizer remember less noise-flooded history information (small $\beta_1$) and mainly focus on information of current parameter upate iteration. When the batch size is large, the graph (loss) becomes stable, we should remeber more history information (large $\beta_1$) for guiding optimization in later training stage, since the gradient at early stage usually contains more informaion than noise (high signal-to-noise ratio).

---

> > > > ### Comment · Reviewer_vVfW · 2025-08-08
> > > >
> > > > Thank you for your explanations. The authors are encouraged to include the explanations in the revision. Concern addressed.

---

### Official Review · Reviewer_9xUK · 2025-07-03

**Clarity:** 3
**Significance:** 3
**Originality:** 3
**Rating:** 5
**Confidence:** 5

**Summary:**

The paper proposes an adaptor to optimization algorithms by introducing a term that minimizes the norm of the gradient, thereby, leading the parameters to a flatter minima. The authors show across multiple experiments that their method improves performance, especially for large mini-batch training.

**Questions:**

Please see Weaknesses for actionable items that will improve the paper. I am willing to update the score based on authors addressing some or all of these.

**Ethical Concerns:**

["NO or VERY MINOR ethics concerns only"]

**Final Justification:**

The authors have addressed most of my concerns relating to the new hyperparameters introduced and the weak small-batch training results. In the light of the detailed rebuttal and suggestions to include a brief version of them in the final copy of the paper, I'm increasing my final rating to Accept.

**Limitations:**

There is no limitations section and I highly suggest one especially contrasting large vs small batch training & failure modes where alpha is highly sensitive (e.g. cases where every minima is quite sharp etc).

**Paper Formatting Concerns:**

1) Fig 2 caption: Typo: (e) does not exist
2) Typo on line 70: "activate"
3) Line 116: Empty citation []
4) Fonts on Figures need to be legible on printed paper. Example see y-axis for Fig 4

**Quality:**

3

**Strengths And Weaknesses:**

Strengths:
1) The paper is well written and mathematically rigourous
2) The authors motivate the intuition well and describe the design choices systematically, to motivate their contribution
3) The results are very convincing for large batch training and show good improvement

Weaknesses/Questions:
1) Negative Examples: Can the authors find any negative examples/tasks for training with their new optimizer? For instance, take the case of a novel task where inherently the loss function turns out to have minimas that are only sharp and not much of flatter minimas. At the very least, can a toy loss landscape be designed that specifically has only sharp minimas with a large value of Hessian at the minimas? How does the norm of Hessian correlate with the optimal alpha parameter in such cases for convergence?
2) The method introduces a new alpha parameter that is akin to a learn rate but for the grad(||grad f||^2) term. How hard or easy is this to tune? How does this correlate with the learning rate for the term grad(f)? It is not clear if the new parameter is quite sensitive from Figure 4 where, especially for small-batch training, the top 1 accuracy seems to be varying significantly with the alpha parameter value.
3) The results are very convincing for large batch training but not so significant for small-batch training. Do the authors suggest users resorting to their method only for large batch training?
4) An arxiv preprint (https://arxiv.org/abs/1905.13200) looks into similar issues and uses a biased estimator to reduce the variance in the loss landscape, which effectively leads it to a flatter minima. (See AdamUCB). What is the relationship of such methods where the variance of the loss landscape is reduced via a separate loss term Versus reducing the norm of the gradient with a Hessian term as in the current paper?

---

> ### Author Rebuttal · Authors · 2025-07-30
>
> Thanks for your carefully reading and expertise. I will summarize your questions and answer them one by one.
>
> **Q1: How does the norm of Hessian correlate with $\alpha$?**
>
> Although I am not very sure about the target of the question, I try my best to give a tentative answer. If we want to find the correlation between $\alpha$ and sharpness of a minimum, let us set the toy landscape $f(\theta)=\frac{\sigma\theta^2}{2}$. The sharper the landscape is, the greater the $\sigma$ is. As we discussed in Remark 2.2, the convergence process is governed by the following equation:
> $$
> \eta^2\dddot{\theta}+\eta\ddot{\theta}(2-\beta_1-\beta_2)+(1-\beta_1)(1-\beta_2)\dot{\theta}+(1-\beta_1)(1-\beta_2) g+(1-\beta_2)\eta\dot{g}(1+(1-\beta_1)\alpha)=0,
> $$
> where $g=\nabla f$. If $f(\theta)=\frac{\sigma\theta^2}{2}$, we have
> $$
> \eta^2\dddot{\theta}+\eta(2-\beta_1-\beta_2)\ddot{\theta}+(1-\beta_2)(1-\beta_1+\eta\sigma(1+(1-\beta_1)\alpha))\dot{\theta}+(1-\beta_1)(1-\beta_2)\sigma\theta=0.
> $$
> For fast convergence, large $\sigma$ (stiffness) usually requires large $\alpha$ (damping). If the damping is too large, the optimizer converges very slowly. If the damping is too small, the optimizer will oscillate. For the optimal $\alpha$, we should use Routh-Hurwitz Criterion. The calculation is very long, we only give a detail idea here. Considering equation
> $$
> r(x)=\eta^2x^3+\eta(2-\beta_1-\beta_2)x^2+(1-\beta_2)(1-\beta_1+\eta\sigma(1+(1-\beta_1)\alpha))x+(1-\beta_1)(1-\beta_2)\sigma=0,
> $$
> we believe the optimal $\alpha$ should be $$\alpha^{\star}=argmin_{\alpha}(\max\\{Re(x_1),Re(x_2),Re(x_3)\\}).$$
> Considering
> $$
> g(x)=r(x-h)=\eta^2(x-h)^3+\eta(2-\beta_1-\beta_2)(x-h)^2+(1-\beta_2)(1-\beta_1+\eta\sigma(1+(1-\beta_1)\alpha))(x-h)+(1-\beta_1)(1-\beta_2)\sigma=a_3(h)x^3+a_2(h)x^2+a_1(h,\alpha)x+a_0(h,\alpha)=0,
> $$ where $h>0$, we let the real part of all roots of $g$ be negative. Using Routh-Hurwitz Criterion, problem convert to an optimization problem $$\max_{\alpha} h$$ with constraints $a_3(h)>0, a_2(h)>0, a_1(h,\alpha)>0, a_0(h,\alpha)>0, a_2(h)a_1(h,\alpha)>a_3(h)a_0(h,\alpha)$. Solving the optimization problem, we obtain the optimal $\alpha$.
>
> As we said in Remark 2.1, $\alpha$ could be negative and positive. In detail, if we want the optimizer to explore more possible minima (to escape local minima), $\alpha$ should be negative. The sharper the local minima are (greater the norm of Hessian), the smaller the $\alpha$ should be (greater $|\alpha|$). In contrast, if we do not want the optimizer to explore more possible minima but want it to exploit the possible global minimum found at first (to converge to the possible global minimum), $\alpha$ should be positive. The faster the convergence we want, the greater the $\alpha$ should be. Carefully, as we said in Remark 2.2, $|\alpha|$ is bounded by $\frac{1}{1-\beta_1}$ for convergence stability. When $\alpha$ is negetive and very smalll (near $-\frac{1}{1-\beta_1}$), the optimizer can escape nearly every minimum (no matter how sharp it is), but it need very long time to converge.
>
> **Q2: How to tune  $\alpha$ and $\beta_1$?**
>
> In fact, it is easy to tune. Usually, the larger the batch size is , the larger the $-\alpha$, $\beta_1$, and learning rate $\eta$ are. For large batch training, the landscape is almost unchanged from batch to batch, especially full-batch (totally unchanged). Meanwhile, there is a efficiency problem in large batch training. We hope the optimizer converges to the same accuracy level with the same the number of epochs as it does in small batch training. This means large batch training needs large learning rate $\eta$ (for fewer iterations), but the learning rate is bounded by geometry of the landscape. However, the landscape of large batch is more stable, which is friendly to convergence. Therefore, we should let the optimizer to explore the landscape in large batch case (negative $\alpha$) and to exploit the landscape in small batch case (positive $\alpha$). As we said in Line 216, we default to $\alpha=0.5, \beta_1=0.01$ for small batch and $\alpha=-5, \beta_1=0.99$ for large batch in all our experiments (of course, it can be better).  As we show in Figure 4, there are more better results can be achieved with different $\alpha$ and $\beta_1$. We just want to show the method works. Only choosing the best result in experiment may be thought to be cherry-picky.
>
> **Q3: The optimizer is for only large batch training or both large and small batch training?**
>
> Since large batch training problem is very difficult in industry field, we just thoroughly focus on large batch training problem to show the ability of our optimizer for large batch training. Of course, the adaptor also theoretically works for small batch, since it degenerates to an optimizer itself when $\alpha=0$. For small batch case, we should set $\alpha$ positive and $\beta_1$ small (defalut to $\alpha=0.5$ and $\beta_1=0.01$). The not so significant result for small batch case just be due to the unified default hyperparameter setting is not very suitable for certain small batch task. We do not spend much time on choose a suitable hyperparameter. Slightly tuning the $\alpha$ and $\beta_1$ will improve the result. We will systematically add more experiments of small batch training in our paper.
>
> **Q4: What is the relationship between reducing the variance of the loss landscape via a separate loss term and reducing the norm of the gradient with a Hessian term as in the current paper?**
>
> AdamUCB is an adapted Adam optimizer by adding $\sigma_k:=\sqrt{\mathbb{E}\left\|\ell_k(\boldsymbol{\theta}, \zeta)-\mathbb{E}\ell(\boldsymbol{\theta},\zeta)\right\|^2}$ to loss function $\ell_k$ and tries to reduce both $\sigma_k$ and $\ell_k$. Our method adds $\\|\nabla\ell_k(\boldsymbol{\theta})\\|^2$ to loss function $\ell_k$ and tries to reduce both $\\|\nabla\ell_k(\boldsymbol{\theta})\\|$ and $\ell_k$. The relationship between $\sigma_k(\boldsymbol{\theta})$ and $\\|\nabla\ell_k(\boldsymbol{\theta})\\|$ is
> $$
> \\|\nabla\ell_k(\boldsymbol{\theta})\\|\sqrt{\mathbb{E}\delta_k^2(\boldsymbol{\theta})}=\sigma_k(\boldsymbol{\theta}), \tag{1}
> $$
> if we name $\delta_k(\boldsymbol{\theta}):=\frac{\mathbb{E}\ell(\boldsymbol{\theta})-\ell_k(\boldsymbol{\theta})}{\\|\nabla\ell_k(\boldsymbol{\theta})\\|}$ horizontal amplitude at $\boldsymbol{\theta}$ and assume $\\|\nabla\ell_k(\boldsymbol{\theta})\\|\neq 0$ at $\boldsymbol{\theta}$. Substituting the definition of $\delta_k$ into the definition of $\sigma_k$ yeild the equation \eqref{eq:gauss}. We find the difference is $\sqrt{\mathbb{E}\delta_k^2(\boldsymbol{\theta})}$. From the definition of the horizontal amplitude $\delta_k$, we find $\mathbb{E}\ell(\boldsymbol{\theta})=\ell_k(\boldsymbol{\theta})+\delta_k(\boldsymbol{\theta})\\|\nabla\ell_k(\boldsymbol{\theta})\\|\approx\ell_k(\boldsymbol{\theta}+\delta_k(\boldsymbol{\theta})\frac{\nabla\ell_k(\boldsymbol{\theta})}{\\|\nabla\ell_k(\boldsymbol{\theta})\\|})$. This means if we regard the graph (landscape) of $\mathbb{E}\ell(\boldsymbol{\theta})$ as a translation of the graph of $\ell_k(\boldsymbol{\theta})$, the smallest translation length should be $\delta_k$. The $\delta_k$ measures how much the the graph of $\ell_k(\boldsymbol{\theta})$ shakes around the graph of  $\mathbb{E}\ell(\boldsymbol{\theta})$, and the difference between the two methods is the standard deviation of $\delta_k$.
>
> **Thank you again for finding paper formatting problem, we will revise our paper soon.**

---

> > ### Comment · Reviewer_9xUK · 2025-08-06
> > **Thanks to the authors for their rebuttal**
> >
> > Thank you for the detailed rebuttal to the concerns raised, and for clarifications on my questions on alpha, beta_1.
> >
> > To significantly improve the paper, I suggest the authors include the following changes in the final copy based on the rebuttal:
> > 1. A brief sentence about empirically tuning alpha and beta_1
> > 2. A brief note about the particular focus of this method to large-batch training so the reader primarily judges the method by those results over small-batch results
> > 3. Appropriate citation to the arXiv paper on reducing the variance of the loss function and its connection to the current method in the Related Works section.
> >
> > In the light of the authors' detailed responses, I am inclined to update my rating of the paper.
> > Thank you.

---

### Note · Authors · 2025-08-12

Thanks for responses from reviewers!!!

We guessed we had $\textbf{addressed almost all of your concerns as you said in these responses.}$ If we just partially addressed your concerns, please tell us what the specific concern is for more explanation.

Thanks for all suggestions from reviewers. $\textbf{We will follow your suggestions in revision.}$

---

### Decision · Program_Chairs · 2025-09-17

**Decision:**

Accept (poster)

**Comment:**

This paper introduces ALTO, an adaptor for gradient-based optimizers designed to find better minima by continuing to explore the loss landscape along low-loss "valleys" rather than halting at the first local minimum it encounters. The core mechanism involves augmenting the standard gradient update with an acceleration term, approximated by the difference between consecutive gradients (g(k) - g(k-1)), which helps the optimizer escape sharp minima and traverse flat regions more effectively. The reviewers (9xUK, LVE5) found this approach to be well-motivated, elegant, and supported by a rigorous theoretical analysis that provides convergence guarantees in both convex and non-convex settings. The paper's primary strength, noted by all reviewers, lies in its extensive and convincing empirical results, which demonstrate significant performance improvements over state-of-the-art optimizers, particularly in the challenging and practically relevant domain of large-batch training.

The initial reviews raised several important concerns. A key point of discussion, highlighted by reviewers 9xUK, vVfW, and LVE5, was that the method's strong performance was primarily demonstrated in large-batch settings, with less significant or even inferior results in small-batch scenarios. Reviewers 9xUK and uGBV also questioned the introduction of two new hyperparameters (alpha, beta_1) and their tuning sensitivity. Finally, reviewer uGBV noted a lack of direct experimental evidence confirming that ALTO indeed finds flatter minima. The authors provided a comprehensive rebuttal that effectively addressed most of these points. They clarified that their focus was intentionally on the difficult large-batch training problem and provided clear, practical intuition for setting the new hyperparameters based on batch size—linking them to a trade-off between exploration (large-batch) and exploitation (small-batch). This explanation satisfied reviewers 9xUK, vVfW, and LVE5. The authors also agreed to add further discussion, improve formatting, and include numerical comparisons of minima sharpness in the final version to address reviewer uGBV's valid point.

This paper presents a novel and well-supported contribution to the field of optimization. The proposed method of using an acceleration term to facilitate valley exploration is a valuable idea that demonstrably improves upon strong baselines in large-batch training across various domains, including CV, NLP, and RL. The discussion period was constructive, and the authors showed a clear commitment to improving the paper based on reviewer feedback. Although the final scores are borderline, I believe the paper has significant enough contributions to be of interest to the NeurIPS community. I therefore recommend that this paper be accepted. I strongly encourage the authors to incorporate the detailed feedback from all reviewers and the clarifications from the rebuttal period into the final version of their paper.